# Posterior Sampling with Delayed Feedback for Reinforcement Learning with Linear Function Approximation

**Nikki Lijing Kuang** *
University of California, San Diego
l1kuang@ucsd.edu

**Ming Yin** *
Princeton University
my0049@princeton.edu

**Mengdi Wang**
Princeton University
mengdiw@princeton.edu

**Yu-Xiang Wang**
University of California, Santa Barbara
yuxiangw@cs.ucsb.edu

**Yi-An Ma**
University of California, San Diego
yianma@ucsd.edu

## Abstract

Recent studies in reinforcement learning (RL) have made significant progress by leveraging function approximation to alleviate the sample complexity hurdle for better performance. Despite the success, existing provably efficient algorithms typically rely on the accessibility of immediate feedback upon taking actions. The failure to account for the impact of delay in observations can significantly degrade the performance of real-world systems due to the regret blow-up. In this work, we tackle the challenge of delayed feedback in RL with linear function approximation by employing posterior sampling, which has been shown to empirically outperform the popular UCB algorithms in a wide range of regimes. We first introduce *Delayed-PSVI*, an optimistic value-based algorithm that effectively explores the value function space via noise perturbation with posterior sampling. We provide the first analysis for posterior sampling algorithms with delayed feedback in RL and show our algorithm achieves $\widetilde{O}(\sqrt{d^3 H^3 T} + d^2 H^2 \mathbb{E}[\tau])$ worst-case regret in the presence of unknown stochastic delays. Here $\mathbb{E}[\tau]$ is the expected delay. To further improve its computational efficiency and to expand its applicability in high-dimensional RL problems, we incorporate a gradient-based approximate sampling scheme via Langevin dynamics for *Delayed-LPSVI*, which maintains the same order-optimal regret guarantee with $\widetilde{O}(dHK)$ computational cost. Empirical evaluations are performed to demonstrate the statistical and computational efficacy of our algorithms.

## 1 Introduction

Reinforcement Learning (RL) is the main workhorse for sequential decision-making problems where an agent needs to balance the trade-off between exploitation and exploration in the unknown environment. The flexible and powerful function approximation endowed by deep neural networks greatly contributes to the empirical success of RL in domains such as Large Language Models (LLMs) [50, 59], robotics [51], and AI for Science [37]. In general, collecting real-world training data from such practical systems can be expensive, which requires algorithms to be both sample efficient and computationally efficient. Recently, there have been growing efforts towards studying provably efficient RL algorithms in settings ranging from tabular Markov Decision Processes (MDPs) [29, 45, 69] to large-scale RL with function approximation [13, 35]. However, these algorithms typically rely

---

*Equal contribution.

37th Conference on Neural Information Processing Systems (NeurIPS 2023).

on the availability of immediate observations of states, actions and rewards in learning no-regret policies. Unfortunately, such an assumption is rarely satisfied in real-world domains, where delayed feedback is ubiquitous and fundamental. In recommender systems and online advertisement, for instance, responses from users (e.g. click, purchase) may not be immediately observable, which can take hours or days. In healthcare and clinical trials, medical feedback from patients on the effectiveness of treatments can only be determined at a deferred time frame. More examples exist in platforms that involve human interaction and evaluation, including human-robot collaboration in teleoperating systems and multi-agent systems [15, 39], aligning LLMs with human values [50, 63], and fine-tuning generative AI models using RL with human feedback (RLHF) [11, 41].

Despite the practical importance of addressing delays in decision-making problems, theoretical understanding of delayed feedback in RL remains limited. Recent parallel works study exploration under delayed feedback via upper confidence bound (UCB) algorithms [8] in tabular RL [29, 45], adversarial MDPs [36, 40], and RL with low policy-switching scheme [68] (see Table 1). Nevertheless, posterior sampling (PS) analysis that handles delayed feedback remains untackled in both bandit and RL literature. We aim to bridge the gap in this work.

PS is a randomized Bayesian algorithm that extends Thompson sampling (TS) [57] to RL, which selects an action according to its posterior probability of being the best. This philosophy inspires a number of promising exploration strategies that explicitly or implicitly adopt PS to explore [52], including bootstrapped DQN [42, 47] and RLSVI [49]. Compared to the popular UCB algorithms, it bears greater robustness in the presence of delays [14], and provides exceptional computational efficiency with competitive empirical performance [14, 65]. The fact that posteriors are often intractable in practice necessitates the use of approximate Bayesian inference such as ensemble sampling, variational inference (VI) and Markov Chain Monte Carlo (MCMC) [20, 38, 47].

In this paper, we provide the first analysis for the class of PS algorithms that handles delayed feedback in RL frameworks, in which the trajectory information is randomly delayed according to some unknown distribution. We highlight that delayed feedback model imposes new challenges that do not arise in standard RL settings. Algorithmically, it requires the computation of new posterior variance due to the weaker concentration arising from delays. Theoretically, it complicates the frequentist analysis of PS algorithms in several ways: (a) the lack of timely update in posterior learning can cause distribution shift, especially in the case of approximate sampling; (b) delays need to be carefully disentangled to quantify the penalty in regret decomposition and it prohibits the direct application of previous analysis; (c) balance between concentration and anti-concentration needs to be handled deliberately to achieve sub-linear regret.

To tackle these challenges, we introduce two novel value-based algorithms for *linear MDPs* under unknown stochastic delayed feedback. Developed upon Bayesian linear modeling with a multi-round ensembling mechanism ($M \approx \text{Polylog}(H, K, d, \delta)$ round), our algorithms achieve a sub-linear worst-case regret without requiring the knowledge of delay, thereby addressing the question raised in [60] that "No frequentist analysis exists for posterior sampling with delayed feedback". Empirical studies show that our algorithms outperform UCB-based methods in terms of both statistical accuracy and computational efficiency when delays are well-behaved or even long-tailed. We summarize our main contributions as follows.

- We propose the *Delayed Posterior Sampling Value Iteration* (Delayed-PSVI, Algorithm 1) for linear MDPs. It achieves a high-probability worst-case regret of $\widetilde{O}(\sqrt{d^3 H^3 T} + d^2 H^2 \mathbb{E}[\tau])^2$, where $\mathbb{E}[\tau]$ is the expected delay.

- We leverage *Langevin Monte Carlo (LMC)* for approximate inference and introduce *Delayed Langevin Posterior Sampling Value Iteration* (Delayed-LPSVI, Algorithm 2), which maintains the same order-optimal worst-case regret of $\widetilde{O}(\sqrt{d^3 H^3 T} + d^2 H^2 \mathbb{E}[\tau])$. To the best of our knowledge, this is the first analysis that provably incorporates LMC in linear MDPs and jointly considers the impact of delays.

- Both algorithms achieve the optimal dependence on the parameters $d$ and $T$ in leading terms under the class of PS algorithms, and recover the best-available frequentist regret of $\widetilde{O}(\sqrt{d^3 H^3 T})$ [31, 72] as in non-delayed linear MDPs when $\mathbb{E}[\tau] = 0$. In particular, Delayed-LPSVI reduces the computational complexity of Delayed-PSVI from $\widetilde{O}(d^3 H K)$

---

[2]It provides a stronger guarantee as opposed to the weaker worst-case expected regret and Bayesian regret.

to $\widetilde{O}(dHK)$, expanding the applicability in complex high-dimensional RL tasks while potentially providing a more flexible form of approximation.

| Algorithms | Setting | Exploration | Worst-case Regret | Computation |
|---|---|---|---|---|
| [28] | Linear Bandits | UCB | $\widetilde{O}(d\sqrt{T} + d^{3/2}\mathbb{E}[\tau])$ | Confidence set optimization |
| [29] | Tabular MDPs | UCB | $\widetilde{O}(\sqrt{SAH^3T} + S^2AH^3\mathbb{E}[\tau])$ | Active update |
| [68] | Linear MDPs | UCB | $\widetilde{O}(\sqrt{d^3H^3T} + dH^2\mathbb{E}[\tau])$ | Multi-batch reduction |
| [40] | Adversarial MDPs | UCB | $\widetilde{O}(H^2S\sqrt{AK} + H^{3/2}\sqrt{S\sum_{k=1}^{K}\tau_k})$ | Confidence set optimization |
| Delayed-PSVI (Thm 1) | Linear MDPs | PS | $\widetilde{O}(\sqrt{d^3H^3T} + d^2H^2\mathbb{E}[\tau])$ | $O((d^3 + Md)HK)$ |
| Delayed-LPSVI (Thm 2) | Linear MDPs | PS | $\widetilde{O}(\sqrt{d^3H^3T} + d^2H^2\mathbb{E}[\tau])$ | $O((N + d)MHK)$ |
| Delayed-PSLB (Cor 2) | Linear Bandits | PS | $\widetilde{O}(\sqrt{d^3T} + d^2\mathbb{E}[\tau])$ | $O((N + d)MK)$ |
| UCB Lower bound [27] | Linear MDPs | UCB | $\Omega(dH\sqrt{T})$ | —— |
| PS Lower bound [24] | Linear Bandits | PS | $\Omega(\sqrt{d^3T})$ | —— |

Table 1: Summary of regret bounds in linear bandits and episodic MDPs under stochastic delay. We denote by $T$ the time horizon, $K$ the number of episodes, $H$ the episode length, $d$ the dimension of feature space, $M$ the number of sampling rounds, and $N$ the total iterations in running LMC. Our choice of $M$ and $N$ has order of Polylog($H, K, d, \delta$), ensuring both Delayed-PSVI and Delayed-LPSVI are computationally efficient and statistically sample-efficient. We remark that the gap in the frequentist regret between PS and best UCB-based methods is unavoidable by a factor of $\sqrt{d}$ [24]. Thus, our dependencies on $d$ and $T$ are optimal for the class of PS algorithms. Our results fulfill the caveat [60] that no worst-case analysis exists for PS with delay.

## 1.1 Related Work.

**Delayed feedback.** In bandit literature, delay is extensively studied in both stochastic [22, 56, 60, 77] and adversarial settings [32, 58, 78] for UCB-based methods. In comparison, while delay draws much attention in empirical RL studies [12, 17, 18], there is a lack of theoretical understanding until very recently. Parallel works focus on UCB-based methods in various RL settings [16, 28, 36, 40, 45, 68]. To provide the first analysis for PS algorithms in this context, we consider stochastic delays under linear function approximation without requiring any policy-switch scheme as in [68].

**Posterior sampling.** To encourage efficient exploration, PS is adopted in value-based methods to inject randomness in empirical Bellman update via Gaussian noise. From the Bayesian perspective, it is equivalent to maintaining an approximate Gaussian posterior for parameterized value function. Its sample complexity is studied in tabular settings [48, 49, 53], with the sharp worst-case regret of $\widetilde{O}(H^2S\sqrt{AT})$ [5]. Under linear function approximation, frequentist regret of $\widetilde{O}(\sqrt{d^3H^3T})$ [31, 72] and Bayesian regret of $\widetilde{O}(d\sqrt{H^3T})$ [19] are established. However, in complex problem domains that require higher computational efficiency and more refined surrogates, approximate inference is the remedy. Toward this end, we resort to a gradient-based MCMC method.

**Langevin Monte Carlo.** LMC is a class of MCMC methods tailored for large-scale online learning with strong convergence guarantee by utilizing the first-order gradient information [64]. It has been successfully applied to stochastic bandits [43], linear bandits [65] and tabular RL [38], In this work, we extend its usage in linear MDPs and demonstrate its convergent property under delay.

**RL with Function Approximation.** Function approximation is widely adopted to empower RL for large-scale applications. Fruitful results have been established for regret minimization in two types of MDPs under linear function approximation: linear mixture MDPs [9, 67], and linear MDPs [35, 66]. In linear mixture MDPs where transition kernel is parameterized as a linear combination of base models, provably efficient algorithms are discussed [13, 75, 76] and [75] provides the corresponding lower bound of $\Omega(dH\sqrt{T})$. In contrast, linear MDPs enjoy a linear structure in value functions by assuming a low-rank representation for both transitions and reward function, where algorithms are shown to enjoy polynomial sample complexity [27, 35, 62, 73]. When it comes to general function approximation, theoretical guarantees are developed based on measures of eluder dimension [54, 61] and Bellman rank [33]. In this work, we focus on delayed feedback in linear MDPs.

## 2 Preliminaries

We study the finite-horizon episodic MDP $(\mathcal{S}, \mathcal{A}, H, \mathbb{P}, r)$, which is time-inhomogeneous, and denote by $\mathcal{S}, \mathcal{A}$ the state and action spaces respectively, $H$ the episode length, $\mathbb{P} = \{\mathbb{P}_h\}_{h=1}^{H}$ the

transition dynamics, and $r = \{r_h\}_{h=1}^{H}$ reward function. At each step $h \in [H]$, $\mathbb{P}_h : \mathcal{S} \times \mathcal{A} \to \Delta_{\mathcal{S}}$ specifies the probabilities of transitioning from the current state-action pair into the next state, and $r_h : \mathcal{S} \times \mathcal{A} \to [0,1]$ emits a bounded reward. We adopt the prior protocol of linear MDPs as follows.

**Definition 1** (Linear MDPs [35, 66]). *Suppose there exists a known feature map $\phi : \mathcal{S} \times \mathcal{A} \to \mathbb{R}^d$ that encodes each state-action pair into a $d$-dimensional feature vector. An MDP is a linear MDP[3] if for any time step $h \in [H]$, $\forall (s,a) \in \mathcal{S} \times \mathcal{A}$, both the transition dynamics $\mathbb{P}$ and reward function $r$ are linear in $\phi$:*

$$\mathbb{P}_h(\cdot|s,a) = \phi(s,a)^{\mathrm{T}} \mu_h(\cdot), \qquad r_h(s,a) = \phi(s,a)^{\mathrm{T}} \theta_h, \qquad (1)$$

*where $\mu_h : \mathcal{S} \to \mathbb{R}^d$ contains $d$ unknown probability measures over $\mathcal{S}$, and $\theta_h \in \mathbb{R}^d$. Furthermore, we assume that $\forall (s,a) \in \mathcal{S} \times \mathcal{A}, \|\phi(s,a)\| \le 1$, and $\forall h \in [H], \|\theta_h\| \le \sqrt{d}, \left\| \int_{\mathcal{S}} \mathrm{d}\mu_h(s') \right\| \le \sqrt{d}$, where $\|\cdot\|$ denotes the Euclidean norm.*

A non-stationary policy $\pi = \{\pi_h\}_{h=1}^{H}$ assigns the action to take at step $h$ in state $s_h \in \mathcal{S}$. Accordingly, we define the value functions of a policy $\pi$ as the expected rewards received under $\pi$:

$$Q_h^{\pi}(s,a) = \mathbb{E}_{\pi} \left[ \sum_{h'=h}^{H} r_{h'} | s_h = s, a_h = a \right], \quad V_h^{\pi}(s) = \mathbb{E}_{\pi} \left[ \sum_{h'=h}^{H} r_{h'} | s_h = s \right].$$

We further denote by $\pi^*$ the optimal policy whose value functions are defined as $V_h^*(s) := V_h^{\pi^*}(s) = \sup_{\pi} V_h^{\pi}(s)$ and $Q_h^*(s,a) := Q_h^{\pi^*}(s,a) = \sup_{\pi} Q_h^{\pi}(s,a)$. Under Definition 1, the action-value functions are always linear in the feature map, and there exists some $w_h^*$ such that $Q_h^* = \phi^{\mathrm{T}} w_h^*$ (Lemma A.1). For ease of notation, , $\forall (s,a)$, denote $[\mathbb{P}_h V_{h+1}^{\pi}](s,a) = \mathbb{E}_{s' \sim \mathbb{P}_h(\cdot|s,a)}[V(s')]$. By Bellman equation and Bellman optimality equation,

$$Q_h^{\pi}(s,a) = (r_h + \mathbb{P}_h V_{h+1}^{\pi})(s,a), \quad V_h^{\pi}(s) = Q_h^{\pi}(s, \pi_h(s)),$$
$$Q_h^*(s,a) = (r_h + \mathbb{P}_h V_{h+1}^*)(s,a), \quad V_h^*(s) = \max_a (r_h + \mathbb{P}_h V_{h+1}^*)(s,a).$$

The goal of the agent is to maximize the cumulative episodic rewards or equivalently, minimize the regret that quantifies the difference between the value of the optimal policy $\pi^*$ and that of the executed policies. Formally, the *worse-case regret* over $K$ episodes is given as:

$$R(T) = \sum_{k=1}^{K} V_1^*(s_1^k) - V_1^{\pi_k}(s_1^k). \qquad (2)$$

**Remark 1.** *Different types of regret are used in literature to measure the performance of PS algorithms. Bayesian regret $\mathbb{E}_{w^* \sim p_0(\cdot)} \mathbb{E}[R(T)|w^*]$ is often considered when assuming a prior $p_0(w)$ over the true parameter $w^*$. Frequentist regret $\mathbb{E}[R(T)]$ is considered when $w^*$ is fixed, where the expectation is taken over all the randomness over data and algorithm. As explained in Appendix A.2, the worst-case regret that we study is stronger than the frequentist regret.*

## 2.1 Delayed Feedback Model

In this work, we consider stochastic delays across episodes. More specifically, the trajectory (i.e., sequence of states, actions and rewards) generated in each episode is not immediately observable in the presence of delay. The formal definition is given as follows.

**Definition 2** (Episodic Delayed Feedback). *In each episode $k \in [K]$, the execution of a fixed policy $\pi^k$ generates a trajectory $\{s_h^k, a_h^k, r_h^k, s_{h+1}^k\}_{h \in [H]}$. Such trajectory information is called the feedback of episode $k$. Let $\tau_k$ represent the random delay between the rollout completion of episode $k$ and the time point at which its feedback becomes observable.*

**Remark 2.** *Various types of delays have been independently studied in the literature, including delays in states [4, 12, 16], delays in rewards [25, 45, 60], delays in actions[56], and delays in trajectories [29, 68]. We focus on the last scheme which facilitates the delayed analysis of value-based methods in episodic linear MDPs.*

---

[3]Linear MDPs recover tabular MDPs by taking $d = |\mathcal{S}||\mathcal{A}|$, where feature map is a one-one mapping for each state-action pair.

---

**Algorithm 1:** Delayed Posterior Sampling Value Iteration (Delayed-PSVI)

**Input:** priors $p_0(w_h^k) \leftarrow \mathcal{N}(0, \lambda I)$, scaling factor $\nu$, multi-round paramter $M$, hyper parameters $\lambda$ and $\sigma^2$.

1 **Initialization**: $\forall k, h, \widetilde{Q}_{H+1}^k(\cdot, \cdot), \widetilde{V}_{H+1}(\cdot, \cdot), \widetilde{V}_h(\cdot, \cdot) \leftarrow 0, \mathcal{D}_h \leftarrow \emptyset$.

2 **for** *episode* $k = 1, \ldots, K$ **do**

3      Sample initial state $s_1^k$

4      **for** *time step* $h = H, \ldots, 1$ **do**

5          $\boldsymbol{y_h} \leftarrow [y_h^1, \ldots, y_h^{k-1}]$, with $y_h^\tau \leftarrow \mathbb{1}_{\tau, k-1} \cdot [r_h^\tau + \widetilde{V}_{h+1}(s_{h+1}^\tau)]$

6          $\Phi_h \leftarrow [\phi^1, \phi^2, \ldots, \phi^{k-1}]$ with $\phi^\tau = \mathbb{1}_{\tau, k-1} \cdot \phi(s_h^\tau, a_h^\tau)$

7          $\Omega_h^k \leftarrow \sigma^{-2} \Phi_h \Phi_h^{\mathrm{T}} + \lambda I, \widehat{w}_h^k \leftarrow \sigma^{-2} (\Omega_h^k)^{-1} \Phi_h \boldsymbol{y_h}^{\mathrm{T}}$

8          $p(w_h^k \mid \mathcal{D}_h, \boldsymbol{y_h}) \leftarrow \mathcal{N}(\widehat{w}_h^k, \nu^2 \cdot (\Omega_h^k)^{-1})$

9          **for** *m = 1, ..., M* **do**

10              Sample $\widetilde{w}_h^{k,m} \sim p(w_h^k \mid \mathcal{D}_h, \boldsymbol{y_h})$

11              $\widetilde{Q}_h^{k,m}(\cdot, \cdot) \leftarrow \phi(\cdot, \cdot)^{\mathrm{T}} \widetilde{w}_h^{k,m}$

12          Update $\widetilde{Q}_h^k(\cdot, \cdot) \leftarrow \max_m \widetilde{Q}_h^{k,m}$

13          $\widetilde{V}_h(\cdot, \cdot) \leftarrow \max_a \min\{\widetilde{Q}_h^k(\cdot, a), H - h + 1\}$

14          Update $\pi_h^k(\cdot) \leftarrow \mathrm{argmax}_{a \in \mathcal{A}} \min\{\widetilde{Q}_h^k(\cdot, a), H - h + 1\}$

15      **for** *time step* $h = 1, \ldots, H$ **do**

16          Choose action $a_h^k = \pi_h^k(s_h^k)$

17          Collect trajectory observations $\mathcal{D}_h \leftarrow \mathcal{D}_h \cup \{(s_h^k, a_h^k, r_h^k, s_{h+1}^k)\}$

         /* Feedback generated in episode $k$ cannot be immediately observed in the presence of delay    */

---

Episodic delays do not disrupt the policy rollout within an episode, but alter the utilization of information in subsequent episodes. More precisely, the feedback of episode $k$ remains inaccessible for the following $\tau_k - 1$ episodes, becoming observable only at the onset of the $(k + \tau_k)$-th episode. To track whether the feedback generated at episode $k$ is revealed at episode $k'$, we utilize the indicator $\mathbb{1}_{k,k'} := \mathbb{1}\{k + \tau_k \leq k'\}$ (where 1 denotes "yes" and 0 denotes "no"). We follow the standard assumption in literature in [28, 68] to assume delays are sub-exponential. It is crucial to note that this assumption primarily serves the purpose of theoretical analysis and is not a prerequisite for the effective functioning of our algorithms in practical settings. Without loss of generality, we discuss the performance bound under general random delays in Section 4 and empirically study the performance against different types of delays in Section 5.

**Assumption 1** (Sub-exponential Episodic Delay). *The episodic delays $\{\tau_k\}_{k=1}^K$ are non-negative, integer-valued, independent and identically distributed $(v, b)$-subexponential random variables: $\tau_k \overset{i.i.d.}{\sim} f_\tau(\cdot)$ with $f_\tau(\cdot)$ being the probability mass function, and $\mathbb{E}[\tau]$ being the expected value. For all $k \in [K]$, the moment generating function of $\tau_k$ satisfies:*

$$\mathbb{E}\left[\exp\left(\gamma\left(\tau_k - \mathbb{E}\left[\tau\right]\right)\right)\right] \leq \exp\left(\frac{1}{2} v^2 \gamma^2\right),$$

*where $v$ and $b$ are non-negative, and $|\gamma| \leq 1/b$.*

## 3 Delayed Posterior Sampling Value Iteration

In this section, we introduce a novel optimistic value-based algorithm, namely, *Delayed Posterior Sampling Value Iteration* (Delayed-PSVI), which efficiently explores the value function space in linear MDPs by embracing several critical components: posterior sampling that injects random noise when performing the least-square value iteration, optimism via multi-round sampling to achieve the optimal worst-case regret and delayed feedback model that encodes episodic trajectory delays.

**Noisy value iteration via posterior sampling.** At the beginning of each episode, we apply PS to sample an estimated value function from the posterior, which is maintained using the observed feedback $\mathcal{D}$ over the previous episodes. Specifically, at each time step, the $Q$-function is parameterized by some $w \in \Gamma$ such that $\widetilde{Q}(s, a) = \phi(s, a)^{\mathrm{T}} w$ is an approximation of the corresponding true optimal $Q$-function $Q^*(s, a)$. Let $p_0(w)$ be the prior of $w$, and $p(\boldsymbol{y}|w, \mathcal{D})$ be the likelihood of the observation $\boldsymbol{y}$, then the posterior of $w$ satisfies:

$$p(w|\mathcal{D}, \boldsymbol{y}) \propto \exp(-L(w, \boldsymbol{y}, \mathcal{D}))p_0(w),$$

where $L(\cdot)$ is the log-likelihood. Unlike the case of model-based RL (MBRL), where PS is utilized to maintain an exact posterior over the environment model, we aim to adopt PS to perform noisy value-iteration by injecting randomness for efficient exploration of the value function space. Specifically, at each step $h \in [H]$, we consider Gaussian-noise perturbation in Delayed-PSVI by setting prior as $p_0(w_h) = \mathcal{N}(0, \lambda I_d)$, and log-likelihood (with $\mathcal{D}_h = \{s_h^\tau, a_h^\tau, r_h^\tau, s_{h+1}^\tau\}_{h \in [H]}^{\tau \in [k-1]}$) as

$$L(w_h, \boldsymbol{y}_h, \mathcal{D}_h) = \sum_{\tau=1}^{k-1} (\phi(s_h^\tau, a_h^\tau)^\mathrm{T} w_h - y_h^\tau)^2, \tag{3}$$

where $\boldsymbol{y}_h = [y_h^1, \ldots, y_h^{k-1}]$ with $y_h^\tau = r_h^\tau(s_h^\tau, a_h^\tau) + \widetilde{V}_{h+1}(s_{h+1}^\tau)$. Then for all step $h \in [H]$ of episode $k$, the posterior of $w_h^k$ follows a Gaussian distribution,

$$p(w_h^k | \mathcal{D}_h, \boldsymbol{y}_h) \propto \mathcal{N}\Big((\Omega_h^k)^{-1}\Phi_h \boldsymbol{y}_h^\mathrm{T}, (\Omega_h^k)^{-1}\Big),$$

where $\Omega_h^k := \Phi_h\Phi_h^\mathrm{T} + \lambda I_d$ and $\Phi_h = [\phi(s_h^1, a_h^1), \phi(s_h^2, a_h^2), \ldots, \phi(s_h^{k-1}, a_h^{k-1})]$. Adding the scaling factors $\sigma^2$ and $\nu^2$ yields the Line 10 of Algorithm 1. It is important to note that while the induced likelihood $\exp(-L(w_h^k, \boldsymbol{y}_h^k, \mathcal{D}_h^k))$ from (3) is Gaussian, we do not assume $y_h^\tau = r_h^\tau(s_h^\tau, a_h^\tau) + \widetilde{V}_{h+1}(s_{h+1}^\tau)$ follows a Gaussian distribution. Instead, the above likelihood model can be used for non-Gaussian problems as we need not sample from the exact Bayesian posterior model [2, 74].

On the other hand, the $\widehat{w}_h^k$ computed in Line 9 of Algorithm 1 together with the greedy choice $\widetilde{V}(\cdot) \approx \max_a \widetilde{Q}(\cdot, a)$ (Line 15) approximates the solution of Bellman optimality equation via the least-square ridge regression: $\widehat{w}_h^k = \mathrm{argmin}_w \sum_{\tau=1}^{k-1} (\phi(s_h^\tau, a_h^\tau)^\mathrm{T} w - (r + \max \bar{Q}_h^k))^2 + \lambda I_d$.[4] Consequently, Line 5-10 essentially performs the Posterior Sampling Value Iteration.

**Optimism via multi-round sampling scheme.** Unlike the Bayesian regret or the worst-case expected regret, the high-probability worst-case regret in (2) needs to control the sub-optimal gap with arbitrarily high probability of at least $1 - \delta$. However, sampling once at each time step only provides a constant-probability optimistic estimation, which breaks the high probability requirement. In addition, the estimation error incurred by sampling (i.e. constant-probability pessimistic estimation) at each timestep will propagate to the previous time steps during the backward posterior sampling value iteration. This phenomenon does not appear in the 1-horizon bandit problem due to a saturated-arm analysis [2, 6]. To remedy this issue, we design a multi-round sampling scheme that generates $M$ estimates $\{\widetilde{Q}^m\}_{m \in [M]}$ for $Q$-fuction through $M$ i.i.d. sampling procedures, and constructs an optimistic estimate by setting $\widetilde{Q} = \max_m \widetilde{Q}^m$. Notably, our choice of $M$ has order Polylog$(H, K, d, \delta)$, and thus makes our algorithm sample-efficient without increasing the overall complexity dependence. As shown in Line 11-14 of Algorithm 1, this scheme guarantees the optimistic estimates $\widetilde{Q} \geq Q^*$ can be achieved as desired. Lastly, ensemble sampling methods enjoy empirical success and popularity in RL [21, 23, 31], including double q-learning [26] and bootstrapped DQN [42, 47]. We are among the first few works to explain its theoretical effectiveness.

**Episodic delayed feedback model.** Recall that by Definition 2, when delay $\tau_k$ takes place, the feedback $\{s_h^t, a_h^t, r_h^t, s_{h+1}^t\}_{h \in [H]}$ of episode $k$ cannot be observed until the beginning of the $k + \tau_k$-th episode. Accordingly, the delayed version of the fully observed $y^\tau, \Omega_h^k$ now becomes,

$$y_h^\tau \leftarrow \mathbb{1}_{\tau, k-1} \cdot [r_h^\tau(s_h^\tau, a_h^\tau) + \widetilde{V}_{h+1}(s_{h+1}^\tau)], \ \Phi_h \leftarrow [\mathbb{1}_{1, k-1} \cdot \phi(s_h^1, a_h^1), \ldots, \mathbb{1}_{k-1, k-1} \cdot \phi(s_h^{k-1}, a_h^{k-1})].$$

As a result, episodic delays are considered during the posterior updates in subsequent episodes. This completes the design of Delayed-PSVI as presented in Algorithm 1. In the remainder of this section, we present the main theoretical guarantees of Delayed-PSVI and the proof sketch of Theorem 1.

**Theorem 1.** *Suppose delays satisfy Assumption 1. In any episodic linear MDP with time horizon $T = KH$, where $K$ is the total number of episodes, for any $0 < \delta < 1$, let $\lambda = 1$, $\sigma^2 = 1$, $M = \log(4HK/\delta)/\log(64/63)$ and $\nu = C_{\delta/4} \approx \widetilde{O}(\sqrt{dMH^2})$ ($C_{\delta/4}$ in Lemma B.10). Then with probability at least $1 - \delta$, there exists some absolute constants $c, c', c'' > 0$ such that the regret of Delayed-PSVI (Algorithm 1) satisfies:*

$$R(T) \leq c\sqrt{d^3 H^3 T \iota} + c' d^2 H^2 \mathbb{E}[\tau]\iota + c'' \iota.$$

*Here $\iota$ is a Polylog term of $H, d, K, \delta$.*

---

[4] Here $\bar{Q} := \min\{\widetilde{Q}(\cdot, a), H - h + 1\}$ is the truncated version.

**On the complexity bound.** Theorem 1 provides the first analysis for PS algorithms under delay and answers the conjecture from [60]. Our result recovers the best-available frequentist regret of $\widetilde{O}(\sqrt{d^3 H^3 T})$ for PS algorithms when there is no delay ($\mathbb{E}[\tau] = 0$). According to [24], the worst-case regret of linear Thompson sampling is lower bounded by $\Omega(\sqrt{d^3 T})$, and this implies our regret dependencies on parameter $d$ and $T$ are optimal under the class of PS algorithms.[5] The order $\sqrt{H^3}$ in our regret is $\sqrt{H}$-suboptimal to the optimal dependence in [27]. As an initial study for posterior sampling with delayed feedback, improving the horizon dependence is beyond our pursuit and we leave it for future work. Moreover, the presence of delay incurs an additive regret term $\widetilde{O}(d^2 H^2 \mathbb{E}[\tau])$. As $T$ grows, the impact of delay will not dominate the overall regret. Furthermore, our high-probability regret bound directly implies the following worst-case expected regret.

**Corollary 1.** *Under the setting of Theorem 1, the expected regret of Delayed-PSVI is bounded by*

$$\mathbb{E}[R(T)] \leq O(\sqrt{d^3 H^3 T \iota}) + O(d^2 H^2 \mathbb{E}[\tau] \iota) + O(\iota)$$

*Here $\iota$ is a Polylog of $H, d, K$. The expectation is taken over the randomness in data and algorithm.*

Proof of Corollary 1 is included in Appendix A.2. Additionally, we present the following corollary in linear bandits, whose main regret $\widetilde{O}\sqrt{d^3 T}$ is optimal for PS algorithms.

**Corollary 2** (Delayed Posterior Sampling for Linear Bandits)**.** *For the linear bandit with $y_t = x_t^{\mathrm{T}} \theta_* + \eta_t$, where $x_t \in D_t \subseteq \mathbb{R}^d$ and $\eta_t$ be a mean-zero noise with $B$-subgaussian. Let $T$ be the total number of steps. Under Assumption 1, for any $0 < \delta < 1$, with probability at least $1 - \delta$, the regret of Delayed-PSLB satisfies:*

$$R(T) \leq O(\sqrt{d^3 T \iota}) + O(d^2 \mathbb{E}[\tau] \iota) + O(\iota).$$

*Here $\iota$ is a Polylog term of $d, K, \delta$.*

### 3.1 Sketch of the analysis

Due to the space limit, we outline the key steps in our analysis and defer the complete proof of Theorem 1 in Appendix B. To bound the worst-case regret in (2), first note that

$$R(T) = \sum_{k=1}^{K} \underbrace{V_1^*(s_1^k) - \widetilde{V}_1^k(s_1^k)}_{\Delta_{opt}^k} + \underbrace{\widetilde{V}_1^k(s_1^k) - V_1^{\pi_k}(s_1^k)}_{\Delta_{est}^k}.$$

Our goal is to attain an optimistic estimation so that $\Delta_{opt}^k \leq 0$ while controlling the estimation error $\Delta_{est}^k$. For optimistic PS algorithms, Gaussian anti-concentration is the main tool [6, 7, 65] to achieve optimism with constant probability. However, the probability of optimism will diminish as the algorithm back-propagates with respect to time. In contrast, we maintain $m \in [M]$ independent ensembles $Q^m$ so that roughly speaking, $\mathbb{P}(Q^m \geq Q^*) \geq \frac{1}{64}$ for all valid $m$. For any $0 < \delta < 1$, with the choice $M = \log(1/m)/\log(64/63)$, the optimistic estimator $Q = \max_m Q^m$ satisfies $\mathbb{P}(Q \geq Q^*) \geq 1 - \delta$ (Lemma B.6). We can then proceed to prove $\Delta_{opt}^k \leq 0$.

To control $\Delta_{est}^k$, one key challenge is to bound the error term $\sum_{k=1}^{K} \left\| \phi\left(s^k, a^k\right) \right\|_{(\Omega^k)^{-1}}$. Due to the presence of delays, we cannot directly apply the Elliptical Potential Lemma as in the non-delayed settings. Therefore, we decompose $(\Omega^k)^{-1}$ into $(\Sigma^k)^{-1} + M_k$, where $\Sigma^k := \sum_{\tau=1}^{k-1} \phi\left(s_h^\tau, a_h^\tau\right) \phi\left(s_h^\tau, a_h^\tau\right)^{\mathrm{T}} + \lambda I$ is the full information matrix, and show

$$\sum_{k=1}^{K} \left\| \phi(s^k, a^k) \right\|_{M_k} \lesssim \max_{k \in [K]} \tau_k \sum_{k=1}^{K} \left\| \phi(s^k, a^k) \right\|_{(\Sigma^k)^{-1}}^2.$$

By doing so, $\sum_{k=1}^{K} \left\| \phi(s^k, a^k) \right\|_{(\Sigma^k)^{-1}}^2$ can be upper bounded by $\widetilde{O}(d \log(K))$ via the Elliptical Potential Lemma and $\max_{k \in [K]} \tau_k$ can be upper bounded by $\widetilde{O}(\mathbb{E}[\tau])$ via the sub-exponential tail bound. Combing all these steps completes the proof.

---

[5]Note for non-sampling based on algorithms, e.g. UCB, the regret can attain $\widetilde{O}(\sqrt{d^2 T})$ [1].

**Algorithm 2:** Delayed Langevin Posterior Sampling Value Iteration (Delayed-LPSVI)

**Input:** $w_0, \eta_k, N_k, \gamma$ and rounds $M, \lambda$. Delayed loss $L_h^k$ as (5).

**1 Initialization:** $\forall k \in [K], h \in [H], \widetilde{Q}_{H+1}^k(\cdot, \cdot) \leftarrow 0, \widetilde{V}_{H+1}^k(\cdot, \cdot) \leftarrow 0, \widetilde{V}_h^0(\cdot, \cdot) \leftarrow 0$

**2 for** *episode* $k = 1, \ldots, K$ **do**

**3**      Sample initial state $s_1^k$

**4**      **for** *time step* $h = H, \ldots, 1$ **do**

**5**          **for** *m = 1, ..., M* **do**

**6**              $\widetilde{w}_h^{k,m} \leftarrow LMC(L_h^k, w_0, \eta_k, N_k, \gamma)$             //*LMC* is given by Algorithm 3

**7**              $\widetilde{Q}_h^{k,m}(\cdot, \cdot) \leftarrow \phi(\cdot)^{\mathrm{T}} \widetilde{w}_h^{k,m}$

**8**          Update $\widetilde{Q}_h^k(\cdot, \cdot) \leftarrow \max_m \widetilde{Q}_h^{k,m}$

**9**          $\widetilde{V}_h^k(\cdot, \cdot) \leftarrow \max_a \min\{\widetilde{Q}_h^k(\cdot, a), H - h + 1\}$

**10**          Update policy $\pi_h^k(\cdot) \leftarrow \operatorname{argmax}_{a \in \mathcal{A}} \min\{\widetilde{Q}_h^k(\cdot, a), H - h + 1\}$

**11**      **for** *time step* $h = 1, \ldots, H$ **do**

**12**          Choose action $a_h^k = \pi_h^k(s_h^k)$

**13**          Collect trajectory observations $\mathcal{D}_h \leftarrow \mathcal{D}_h \cup \{(s_h^k, a_h^k, r_h^k, s_{h+1}^k)\}$

         /* Feedback generated in episode $k$ cannot be immediately observed in the presence of delay      */

## 4 Delayed Posterior Sampling via Langevin Dynamics

Delayed-PSVI performs noisy value iteration for linear MDPs by injecting randomness for exploration via Gaussian noise. From the Bayesian perspective, it constructs a Laplace approximation to obtain a Gaussian posterior given the observed data. However, sampling from a Gaussian distribution with a general covariance matrix $\Omega_h^k$ can be computationally expensive in high-dimensional RL tasks. Specifically, Line 10 of Algorithm 1 is conducted via $\widetilde{w} := \widehat{w} + \nu \cdot \Omega^{-1/2}\zeta$, where $\zeta \sim \mathcal{N}(0, I_d)$. The complexity of computing the matrix inverse involved (*e.g.* via Cholesky decomposition) is at least $O(d^3)$, which is prohibitively high for large $d$. More importantly, in complex problem domains, a flexible form of non-Gaussian noise perturbation may be desirable.

To tackle these challenges, we incorporate a gradient-based approximate sampling scheme via Langevin dynamics for PS algorithms, namely, LMC, and introduce the *Delayed-Langevin Posterior Sampling Value Iteration* (Delayed-LPSVI) in Algorithm 2. The update rule of LMC essentially performs the following noisy gradient update:

$$w_t \leftarrow w_{t-1} - \eta \nabla \mathcal{L}(w_{t-1}) + \sqrt{2\eta\gamma}\epsilon_t,$$

where $\epsilon_t \overset{\text{i.i.d.}}{\sim} \mathcal{N}(0, I_d)$. It is based on the Euler-Murayama discretization of the Langevin stochastic differential equation (SDE):

$$d\boldsymbol{w}(t) = -\nabla L(\boldsymbol{w}(t))dt + \sqrt{2\beta^{-1}} \, d\boldsymbol{B}(t), \tag{4}$$

where $\boldsymbol{B}(t) \in \mathbb{R}^d$ is a Brownian motion, $\beta > 0$ and $t > 0$. Under certain regularity conditions on the drift term $\nabla L(\boldsymbol{w}(t))$ in (4), it can be shown that the Langevin dynamics converges to a unique stationary distribution $\pi(d\boldsymbol{w}) \propto \exp(-\beta L(\mathbf{w}))d\mathbf{w}$. As a result, LMC is capable of generating samples from arbitrarily complex distributions which can be intractble without closed form. With sufficient number of iterations, the posterior of $w_t$ is in proportional to $\exp(-\sqrt{1/\gamma}\mathcal{L}(w))$.

In our problem, we specify $\mathcal{L}$ to be the following delayed loss function

$$L_h^k(w) := \sum_{\tau=1}^{k-1} \mathbb{1}_{\tau,k-1} \left( \langle \phi(s_h^\tau, a_h^\tau), w \rangle - \bar{y}_h^\tau \right)^2 + \lambda \|w\|_2^2, \tag{5}$$

where $\bar{y}_h^\tau := r_h^\tau + \widetilde{V}_{h+1}^k(s_{h+1}^\tau)$. Compared to Delayed-PSVI, Algorithm 2 does not require the matrix inversion computation. Below we present the worst-case regret of Delayed-LPSVI and discuss the key insights in our analysis. The full proof is deferred to Appendix C.

**Theorem 2.** *Suppose delays satisfy Assumption 1. In any episodic linear MDP with time horizon $T = KH$, where $K$ is the total number of episodes and $H$ is the fixed episode length, for any $0 < \delta < 1$, let $\lambda = 1$, $N_k = \max\{\log(\frac{32H^2(K+\lambda)dk}{\gamma\lambda} + 1)/[2\log(1/(1 - \frac{1}{2\kappa_h}))],$ $\frac{\log 2}{2\log(1/(1-\frac{1}{2\kappa_h}))}, \log(\frac{4HK^3}{\sqrt{\lambda/dK}})/\log(1/(1 - \frac{1}{2\kappa_h}))\}, \eta_k = \frac{1}{4\lambda_{\max}(\Omega_h^k)}, \gamma = 16C_{\delta/4}^2 \approx \widetilde{O}(dMH^2),$*

$w_0 = \mathbf{0}$ and $M = \log(4HK/\delta)/\log(64/63)$. Then with probability at least $1 - \delta$, there exists some absolute constants $c, c', c'' > 0$ such that the regret of Algorithm 2 satisfies:

$$R(T) \leq c\sqrt{d^3 H^3 T \iota} + c'd^2 H^2 \mathbb{E}[\tau]\iota + c''\iota.$$

Here $\iota$ is a Polylog term of $H, d, K, \delta$ and $C_\delta$ is defined in Lemma C.9.

Neglecting the constants and Polylog factors, Delayed-LPSVI maintains the same order regret of $\widetilde{O}(\sqrt{d^3 H^3 T} + d^2 H^2 \mathbb{E}[\tau])$ as Delayed-PSVI while significantly improving the computational efficiency. Precisely, LMC requires $O(N)$ complexity to perform gradient steps in Line 6 of Algorithm 2 and an extra $O(d)$ operations to compute $\widetilde{Q}_h^{k,m}$ in Line

---

**Algorithm 3:** Langevin Monte Carlo $LMC(\mathcal{L}, w_0, \eta, N, \gamma)$

---
**1** **for** $t = 1 \ldots N - 1$ **do**
**2** $\quad$ Draw $\epsilon_t \sim \mathcal{N}(0, I_d)$
**3** $\quad$ $w_t \leftarrow w_{t-1} - \eta \nabla \mathcal{L}(w_{t-1}) + \sqrt{2\eta\gamma}\epsilon_t$
**4** **Output:** $w_N$

---

7. Thus, the total computation complexity of LMC is $O((N + d)MHK)$. On the other hand, sampling without LMC (Line5-8 in Algorithm 1) requires $O(d^3)$ operations, and the multi-round sampling (Line9-11) incurs $O(dM)$ additional operations, which implies for a total computation complexity of $O((d^3 + dM)HK)$. As the choice of $N$ in Algorithm 2 has logarithmic order, and $M = \log(4HK/\delta)/\log(64/63)$, the overall complexity of Delayed-LPSVI is $\widetilde{O}(dHK)$, whereas the overall computational complexity of Delayed-PSVI is $\widetilde{O}(d^3 HK)$. Notably, Delayed-LPSVI reduces the computational overhead of Delayed-PSVI by $\widetilde{O}(d^2)$.

**On the analysis.** The key step in the proof of Theorem 2 is to show the convergence guarantee of LMC. Indeed, by recursion, one can show

$$w_N = A_{h,k}^N w_0 + \left(I - A_{h,k}^N\right) \widehat{w}_h^k + \sqrt{2\eta\gamma} \sum_{l=0}^{N-1} A_{h,k}^l \epsilon_{N-l},$$

where $A_{h,k} := I - 2\eta_k \Omega_h^k$. For any $w_0$, it implies $w_N$ follows the Gaussian distribution $\mathcal{N}\left(A_{h,k}^{N_k} w_0 + \left(I - A_{h,k}^{N_k}\right) \widehat{w}_h^k, \Theta_h^k\right)$. With the choice of $\eta_k = \frac{1}{4\lambda_{\max}(\Omega_h^k)}$, $A_{h,k} \prec I_d$ and $\frac{\gamma}{2}(1 - (1 - \frac{1}{2\kappa_h})^{2N_k})\left(\Omega_h^k\right)^{-1} \prec \Theta_h^k \prec \gamma\left(\Omega_h^k\right)^{-1}$, which is the key to connect $\Theta_h^k$ with $\left(\Omega_h^k\right)^{-1}$ (Lemma C.2), the main analysis for Delayed-PSVI goes through by utilizing this connection.

**On arbitrary delayed feedback.** The current study considers the stochastic delays that are sub-exponential Assumption 1. What if delay has an arbitrary distribution (*e.g.* Cauchy distribution has unbounded mean)? Indeed, the regret can be (roughly) bounded by $\widetilde{O}(\frac{1}{q}\sqrt{d^3 H^3 T} + dH^2 d_\tau(q))$ for $d_\tau(q)$ to be the $q$-th quantile of delay $\tau$. We do not focus on this setting since there is a $1/q$ blow-up in the main regret that many distributions (*e.g.* sub-exponential) do not need to sacrifice. We include the discussion in Appendix A.3.

## 5 Experiments

To validate whether our posterior sampling algorithms are competitive or outperform the non-sampling-based algorithms in the delayed setting, in this section, we examine their empirical performance in two simulated RL environments with different delayed feedback distributions. In particular, we consider a linear MDP environment following [44, 46], and a variant of the popular River-Swim [55]. In both environments, we benchmark Delayed-PSVI (Algorithm 1), Delayed-LPSVI (Algorithm 2) against LSVI-UCB [35] with delayed feedback, namely, Delayed-UCBVI. In this section, we discuss results in the first setting and defer the discussion of RiverSwim in Appendix E.

### 5.1 Synthetic Linear MDP

We construct a synthetic linear MDP instance with $|\mathcal{S}| = 2$, $|\mathcal{A}| = 50$, $d = 10$, and $H = 20$. The linear feature mapping embeds each state-action pair with its binary representation and induces the following reward function: $r(s, a) = 0.99$ if $s = 0, a = 0$; $r(s, a) = 0.01$ otherwise. The design of the environment results in the same optimal value $V_1^*(s_1)$ when $d$ and $H$ are fixed. Algorithms are examined under three types of delays that are commonly encountered in real-world phenomena, including sub-exponential delays and long-tail delays:

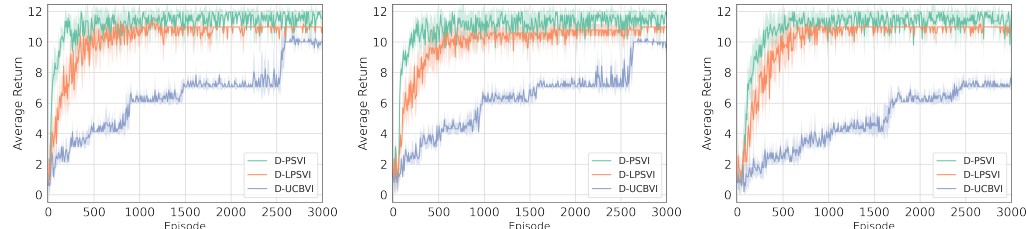

Figure 1: Left:(a) Multinomial delay with delay categories $\{10, 20, 30\}$. (b) Poisson delay with rate $\mathbb{E}[\tau] = 50$. (c) Long-tail Pareto delay with shape 1.0, scale 500. Results are reported over 10 experiments. Delayed-PSVI and Delayed-LPSVI demonstrate robust performance under both well-behaved and long-tail delays.

- **Multinomial delay.** Delays follow a Multinomial distribution with three categories $\{10, 20, 30\}$, with the corresponding probabilities as $\{0.5, 0.3, 0.2\}$.

- **Poisson delay.** Delays follow a Poisson distribution with the expected delay as $\mathbb{E}[\tau] = 50$.

- **Long-tail delay.** Delays are discretized from a Pareto distribution [6] with the shape parameter as 1.0 and the scale parameter as 500.

To run Delayed-LPSVI, we warm start LMC by initializing $w_0$ at each time step with the previous sample, and let $M = 2$, $N = 40$, $\eta = c_\eta / \lambda_{\max}(\Omega_h^k)$. For Delayed-PSVI, we set parameters $M = 2$, $\nu = \sqrt{d}H$. In the case of Delayed-UCBVI, we set the bonus coefficient as $\beta = c_\beta / 2 \cdot dH \sqrt{\log(dH)}$. To make a fair comparison, we perform a grid search to determine the optimal hyperparameter values and fix $c_\beta = 0.1$, $c_\eta = 0.5$, $\gamma = 0.02$. Experiments are repeated with 10 different random seeds, and the returns are averaged over episodes in Figure 1. Further elaboration on additional metrics is available in Appendix E.2.

**Results and Discussions.** Both Delayed-PSVI and Delayed-LPSVI exhibit consistent and robust performance with resilience, not only under the well-behaved delays that decay exponentially fast, as assumed in Assumption 1, but also under the heavy-tailed delays, such as those following Pareto distributions. Notably, when confronted with the challenge of long-tail delays, our algorithms excel Delayed-UCBVI in terms of statistical accuracy (yielding higher return) and convergence rate. Specifically, the performance of Delayed-UCBVI degrades under long-tail delays, resulting from its computational inefficiency in iteratively constructing confidence intervals. In contrast, PS methods offer a higher degree of flexibility to adjust the range of exploration, owing to the inherent randomized algorithmic nature. To assess the computational advantages facilitated by LMC, we consider additional synthetic environments with varied dimensions for a more comprehensive analysis. For detailed statistics and further discussions, please refer to Appendix E.2. It is noteworthy that in practical high-dimensional RL tasks, the computational savings achieved by Delayed-LPSVI, in comparison to Delayed-PSVI, are considerably more significant.

## 6 Conclusion

In this paper, we study posterior sampling with episodic delayed feedback in linear MDPs. We introduce two novel value-based algorithms: Delayed-PSVI and Delayed-LPSVI. Both algorithms are proved to achieve $\widetilde{O}(\sqrt{d^3 H^3 T} + d^2 H^2 \mathbb{E}[\tau])$ worst-case regret. Notably, by incorporating LMC for approximate sampling, Delayed-LPSVI reduces the computational cost by $\widetilde{O}(d^2)$ while maintaining the same order of regret. Our empirical experiments further validate the effectiveness of our algorithms by demonstrating their superiority over the UCB-based methods.

This work provides the first delayed-feedback analysis for posterior sampling algorithms in RL, paving the way to several promising avenues for future research. Firstly, it is interesting to extend the current results to settings with general function approximation [34, 71]. Additionally, leveraging the sharp analysis outlined in [27] to improve the suboptimal dependence on $H$ for posterior sampling algorithms presents an intriguing avenue for exploration. Furthermore, addressing other types of delay (e.g. adversarial delay) that differ from stochastic one will contribute to the ongoing field of delayed feedback studies in online learning, and we leave the investigation in future works.

---

[6]Pareto distribution with shape parameter less than 5.0 are known to have hevy right tails.

## Acknowledgements

Ming Yin and Yu-xiang Wang are gratefully supported by National Science Foundation (NSF) Awards #2007117 and #2003257. Nikki Kuang and Yi-An Ma are supported by the NSF SCALE MoDL-2134209 and the CCF-2112665 (TILOS) awards, as well as the U.S. Department of Energy, Office of Science, and the Facebook Research award. Mengdi Wang gratefully acknowledges funding from Office of Naval Research (ONR) N00014-21-1-2288, Air Force Office of Scientific Research (AFOSR) FA9550-19-1-0203, and NSF 19-589, CMMI-1653435.

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

# Appendices

## A   Some Properties

### A.1   Properties of Linear MDPs

**Lemma A.1.** *In linear MDPs, the action-value function is also linear in feature map.* $\forall (s,a) \in \mathcal{S} \times \mathcal{A}$, $h \in [H]$ *and* $\phi \in \mathbb{R}^d$, *under any fixed policy* $\pi$,

$$Q_h^\pi(s,a) = \phi(s,a)^{\mathrm{T}} w_h^\pi,$$

*where* $w_h^\pi := \theta_h + \mathbb{E}_\mu[V_{h+1}^\pi(s')]$ *and* $w_h \in \mathbb{R}^d$. *As a corollary, there exists* $w_h^*$ *such that* $Q_h^* = \phi^{\mathrm{T}} w_h^*$.

*Proof of Lemma A.1.* By Bellman equation,

$$
\begin{aligned}
Q_h^\pi(s,a) &= r_h(s,a) + \mathbb{E}_{s' \sim \mathbb{P}_h(\cdot|s,a)}[V_{t+1}^\pi(s')] \\
&= \phi(s,a)^{\mathrm{T}} \theta_h + \int V_{h+1}^\pi(s') \, d(\phi(s,a)^{\mathrm{T}} \mu_h(s')) \\
&= \phi(s,a)^{\mathrm{T}} w_h^\pi,
\end{aligned}
$$

where $w_h^\pi := \theta_h + \mathbb{E}_{\mu_h}[V_{h+1}^\pi(s')]$. $\qquad \square$

### A.2   Worst-case regret as a stronger criterion

We use Theorem 1 as an example. Using the worst-case result, *i.e.* with probability $1 - \delta$,

$$R(T) \leq c\sqrt{d^3 H^3 T \iota} + c' d^2 H^2 \mathbb{E}[\tau]\iota + c'' \iota.$$

Here $\iota$ has the functional form $\iota = \mathrm{Polylog}(d, K, H, \delta)$. Then choosing $\delta = 1/(HK)$ to obtain with probability $1 - 1/(HK)$,

$$R(T) \leq c\sqrt{d^3 H^3 T \iota} + c' d^2 H^2 \mathbb{E}[\tau]\iota + c'' \iota := A$$

for $\iota = \mathrm{Polylog}(d, K, H)$. Therefore,

$$
\begin{aligned}
\mathbb{E}[R(T)] &\leq \mathbb{E}[R(T)\mathbb{1}_{\{R(T) \leq A\}}] + \mathbb{E}[R(T)\mathbb{1}_{\{R(T) \geq A\}}] \\
&\leq A \cdot 1 + HK \cdot \mathbb{P}(R(T) \geq A) \leq A + 1.
\end{aligned}
$$

This completes Corollary 1.

### A.3   Discussion on the arbitrary delay

For completeness of our study, we also briefly discuss the case when delay is arbitrary. In genreal, the regret can be (roughly) bounded by $\widetilde{O}(\frac{1}{q}\sqrt{d^3 H^3 T} + dH^2 d_\tau(q))$ for $d_\tau(q)$ to be the $q$-th quantile of delay $\tau$. This could be achieved by creating a low-switching variant of our Theorem 1/Theorem 2 and applying the reduction of the concurrent work [68]. We do not focus on this setting since there is a $1/q$ blow-up in the main regret that many distributions (*e.g.* sub-exponential) do not need to sacrifice.

## B   Regret Analysis for Delayed-PSVI

To proceed with the regret analysis, we introduce some helpful notations. Besides $\widetilde{Q}_h^k(s,a) = \max_m \phi(s,a)^{\mathrm{T}}, \widetilde{w}_h^{k,m}, \widetilde{V}_h^k(s) = \max_a \widetilde{Q}_h^k(s,a)$ in Algorithm 1, we define

$$\widehat{Q}_h^k(s,a) = \phi(s,a)^{\mathrm{T}} \widehat{w}_h^k, \quad \widehat{V}_h^k(s) = \max_a \widehat{Q}_h^k(s,a), \quad \bar{Q}_h^k = \min\{\widetilde{Q}_h^k, H - h + 1\};$$

$$(r_h^k + \mathbb{P}_h \widetilde{V}_{h+1}^k)(s,a) := \phi(s,a)^{\mathrm{T}} w_h^k, \text{ with } w_h^k := \theta_h + \int_{\mathcal{S}} \widetilde{V}_{h+1}^k(s') \mathrm{d}\mu_h(s').$$

**Regret decomposition:** We start by rewriting regret in terms of value-function error decomposition following the standard analysis of optimistic algorithms [10]:

$$R(T) = \sum_{k=1}^{K} \underbrace{V_1^*(s_1^k) - \widetilde{V}_1^k(s_1^k)}_{\Delta_{opt}^k} + \underbrace{\widetilde{V}_1^k(s_1^k) - V_1^{\pi_k}(s_1^k)}_{\Delta_{est}^k},$$

where at each episode $k$, $\Delta_{opt}^k$ corresponds to the regret resulting from optimism, and $\Delta_{est}^k$ tracks down the regret incurred from estimation error. Efficient RL algorithms thus need to strike a balance between both terms. More specifically, it is desirable to generate optimistic estimations over the true value function, while keeping estimation error relatively small. By cautious design of noise perturbation, we show in Theorem 1 that Algorithm 1 effectively achieves $\sqrt{T}$ order regret in episodic MDPs with linear function approximation.

*Proof of Theorem 1.* The proof proceeds by bounding $\Delta_{opt}^k$ and $\Delta_{est}^k$ respectively.

**Step 1: bound regret from optimism.**

By Lemma B.7, the optimism provided by our algorithm guarantees with probability at least $1 - \delta/2$, for all $k \in [K]$, $\Delta_{opt}^k := V_1^*(s_1^k) - \widetilde{V}_1^k(s_1^k) \leq 0$.

**Step 2: bound regret from estimation error.** To bound the estimation error, we first condition on the following event

$$\mathcal{E} := \{ \| \min\{\widetilde{Q}_h^k(s,a), H-h+1\} - (r_h^k + \mathbb{P}_h \widetilde{V}_{h+1}^k)(s,a)| \leq \beta \, \|\phi(s,a)\|_{(\Omega_h^k)^{-1}} + \frac{1}{K^3}, \, \forall s,a,h,k \},$$

with $\beta := \sqrt{2\nu^2 \log(16 C_d H M K/\delta)} + \sqrt{8H^2 \left[ \frac{d}{2} \log\left(\frac{k+\lambda}{\lambda}\right) + dM \log(1 + \frac{2\sqrt{8k^3} C_{H,d,k,M,\delta/8}}{H\sqrt{\lambda}}) + \log \frac{16}{\delta} \right]} + 2\sqrt{\lambda}\sqrt{d}H.$[7] Here $C_d$ and $C_{H,d,k,M,\delta}$ are defined in Lemma B.8.

Recall that $\Delta_{est}^k := \widetilde{V}_1^k(s_1^k) - V_1^{\pi_k}(s_1^k)$ and define $\zeta_h^k = \mathbb{E}[\widetilde{V}_{h+1}^k(s_{h+1}^k) - V_{h+1}^{\pi_k}(s_{h+1}^k) | s_h^k, a_h^k] - \widetilde{V}_{h+1}^k(s_{h+1}^k) + V_{h+1}^{\pi_k}(s_{h+1}^k)$. Then by applying Lemma B.1 recursively, the total estimation error $\sum_{k=1}^{K} \Delta_{est}^k$ can be decomposed as:

$$\sum_{k=1}^{K} \Delta_{est}^k = \sum_{k=1}^{K} \widetilde{V}_1^k(s_1^k) - V_1^{\pi_k}(s_1^k)$$

$$\leq \sum_{k=1}^{K} \left( \widetilde{V}_2^k(s_2^k) - V_2^{\pi_k}(s_2^k) + \zeta_1^k + \beta \left\| \phi(s_1^k, a_1^k) \right\|_{(\Omega_1^k)^{-1}} + \frac{1}{K^3} \right) \qquad (6)$$

$$\leq \dots$$

$$\leq \sum_{k=1}^{K} \sum_{h=1}^{H} \zeta_h^k + \beta \sum_{k=1}^{K} \sum_{h=1}^{H} \left\| \phi(s_h^k, a_h^k) \right\|_{(\Omega_h^k)^{-1}} + \frac{H}{K^2}.$$

On one hand, by definition, $|\zeta_h^k| \leq 2H$ for all $h \in [H], k \in [K]$. Therefore, $\{\zeta_h^k\}$ is a martingale difference sequence (since the computation of $\widetilde{V}_h^k$ is independent of the new observation at episode $k$). By Azuma-Hoeffding's inequality (for $t > 0$),

$$\mathbb{P} \left( \sum_{k=1}^{K} \sum_{h=1}^{H} \zeta_h^k > t \right) \geq \exp \left( \frac{-t^2}{2K \cdot H^3} \right) := \delta/8,$$

which implies with probability $1 - \delta/8$,

$$\sum_{k=1}^{K} \sum_{h=1}^{H} \zeta_h^k \leq \sqrt{2KH^3 \cdot \log(8/\delta)} = \sqrt{2H^2 T \cdot \log(8/\delta)}. \qquad (7)$$

---

[7] Note here the $\delta$ equals $\delta/4$ as of Lemma B.8. Therefore, by Lemma B.8, $\mathbb{P}(\mathcal{E}) \geq 1 - \delta/4$.

**Step 3: bounding the delayed error.** By Lemma B.4, with probability $1 - \delta/8$,

$$\sum_{h-1}^{H}\sum_{k=1}^{K}\left\|\phi(s_h^k, a_h^k)\right\|_{(\Omega_h^k)^{-1}} \leq H\sqrt{2dK\log((d+K)/d)} + dHD_{\tau,\delta,H,K}\log((d+K)/d).$$

Here $D_{\tau,\delta,H,K} := 1 + 2\mathbb{E}[\tau] + 2\sqrt{2\mathbb{E}[\tau]\log(\frac{24KH}{\delta})} + \frac{4}{3}\log(\frac{24KH}{\delta}) + D_{\tau,K,\frac{\delta}{16H}}$ and $D_{\tau,K,\delta}$ is defined in Lemma D.6. Consequently,

$$\beta\sum_{k=1}^{K}\sum_{h=1}^{H}\left\|\phi(s_h^k, a_h^k)\right\|_{(\Omega_h^k)^{-1}} \leq \beta H\sqrt{2dK\log((d+K)/d)} + \beta dHD_{\tau,\delta,H,K}\log((d+K)/d). \quad (8)$$

Note that by Lemma B.8, event $\mathcal{E}$ holds with probability $1 - \delta/4$, and by a union bound with (7) and (8), we have with probability $1 - \delta/2$,

$$\sum_{k=1}^{K}\Delta_{est}^k \leq \sqrt{2H^2T \cdot \log(8/\delta)} + \beta H\sqrt{2dK\log((d+K)/d)} + \beta dHD_{\tau,\delta,H,K}\log((d+K)/d) + \frac{H}{K^2}.$$

Finally, by a union bound over Step1, Step2 and Step3, we obtain with probability $1 - \delta$,

$$R(T) = \sum_{k=1}^{K}\Delta_{opt}^k + \sum_{k=1}^{K}\Delta_{est}^k \leq \sum_{k=1}^{K}\Delta_{est}^k$$

$$\leq \sqrt{2H^2T \cdot \log(8/\delta)} + \beta H\sqrt{2dK\log((d+K)/d)} + \beta dHD_{\tau,\delta,H,K}\log((d+K)/d) + \frac{H}{K^2}$$

$$\leq c\sqrt{d^3H^3T\iota} + c'd^2H^2\mathbb{E}[\tau]\iota + O(\iota)$$

where $c > 0$ is some universal constant and $\iota$ is a Polylog term of $H, d, K, \delta$. The last step is due to: by the choice of $\lambda = 1$, $\sigma^2 = 1$, $\nu = C_{\delta/4}$ and $M = \log(4HK/\delta)/\log(64/63)$, we can bound $C_\delta$ (in Lemma B.10) by $C_\delta \leq c_0 H\sqrt{dM\iota_\delta}$ with $c_0$ a universal constant and $\iota_\delta$ contains only the Polylog terms. This implies $\nu^2 \leq c_1 H^2 dM\iota_\delta$. Note $C_d \leq d\iota_\delta$, therefore $\beta$ is dominated by the first term $\beta \leq C_2\sqrt{2\nu^2\log(16C_dHMK/\delta)} \leq C_3 dH\iota_\delta$ for some universal constants $C_2, C_3$. Since $R(T)$ is dominated by the second term in the second to last inequality, plug back the upper bound for $\beta$ gets the result. Finally, it is readily to verify $D_{\tau,\delta,H,K}$ is bounded by $c'\mathbb{E}[\tau]\iota + O(\iota)$. $\square$

**Lemma B.1.** *Define* $\zeta_h^k = \mathbb{E}[\widetilde{V}_{h+1}^k(s_{h+1}^k) - V_{h+1}^{\pi_k}(s_{h+1}^k)|s_h^k, a_h^k] - \widetilde{V}_{h+1}^k(s_{h+1}^k) + V_{h+1}^{\pi_k}(s_{h+1}^k)$ *and condition on the event* (10) *in Lemma B.8. Then for all* $k \in [K]$, $h \in [H]$, *the following holds,*

$$\widetilde{V}_h^k(s_h^k) - V_h^{\pi_k}(s_h^k) \leq \widetilde{V}_{h+1}^k(s_{h+1}^k) - V_{h+1}^{\pi_k}(s_{h+1}^k) + \zeta_{h+1}^k + \beta\left\|\phi(s_h^k, a_h^k)\right\|_{(\Omega_h^k)^{-1}} + \frac{1}{K^3}.$$

*Proof of Lemma B.1.* Since $a_h^k = \pi_k(s_h^k)$, it implies $V_h^k(x_h^k) = \bar{Q}_h^k(s_h^k, a_h^k)$ (recall $\bar{Q}_h^k := \min\{\widetilde{Q}_h^k, H - h + 1\}$) and $V_h^{\pi_k}(x_h^k) = Q_h^{\pi_k}(s_h^k, a_h^k)$. Hence,

$$\left|(\widetilde{V}_h^k(s_h^k) - V_h^{\pi_k}(s_h^k)) - (\widetilde{V}_{h+1}^k(s_{h+1}^k) - V_{h+1}^{\pi_k}(s_{h+1}^k)) - \zeta_h^k\right|$$

$$= \left|(\widetilde{V}_h^k(s_h^k) - V_h^{\pi_k}(s_h^k)) - \mathbb{E}[\widetilde{V}_{h+1}^k(s_{h+1}^k) - V_{h+1}^{\pi_k}(s_{h+1}^k)|s_h^k, a_h^k]\right|$$

$$= \left|\widetilde{V}_h^k(s_h^k) - r_h^k - (\mathbb{P}_h\widetilde{V}_{h+1}^k)(s_h^k, a_h^k)\right|$$

$$= \left|\bar{Q}_h^k(s_h^k, a_h^k) - r_h^k - (\mathbb{P}_h\widetilde{V}_{h+1}^k)(s_h^k, a_h^k)\right|$$

$$\leq \beta\left\|\phi(s_h^k, a_h^k)\right\|_{(\Omega_h^k)^{-1}} + \frac{1}{K^3},$$

where the last step is by the event defined in (10). $\square$

## B.1  Bounding the delayed error term $\sum_{k=1}^{K}\left\|\phi(s_h^k, a_h^k)\right\|_{(\Omega_h^k)^{-1}}$.

Recall the delayed covariance matrix $\Omega_h^k = \sum_{\tau=1}^{k-1}\mathbb{1}_{\tau,k-1}\phi(s_h^\tau, a_h^\tau)\phi(s_h^\tau, a_h^\tau)^{\mathrm{T}} + \lambda I$ with $\mathbb{1}_{s,t} := \mathbb{1}[s + \tau_s \leq t]$, then we can define the full design matrix $\Sigma_h^k$ and the complement matrix $\Lambda_h^k$ as

$$\Sigma_h^k := \sum_{\tau=1}^{k-1}\phi(s_h^\tau, a_h^\tau)\phi(s_h^\tau, a_h^\tau)^{\mathrm{T}} + \lambda I, \quad \Lambda_h^k := \sum_{\tau=1}^{k-1}\mathbb{1}[s + \tau_s > t]\phi(s_h^\tau, a_h^\tau)\phi(s_h^\tau, a_h^\tau)^{\mathrm{T}}, \quad (9)$$

then $\Sigma_h^k = \Omega_h^k + \Lambda_h^k$. Also, denote the number of missing episodes as: $U_k = \sum_{s=1}^{k} \mathbb{1}[s + \tau_s > k]$. Then we have the following Lemmas.

**Lemma B.2.** *For $\lambda > 0$, $(\Omega_h^k)^{-1} = (\Sigma_h^k)^{-1} + (\Sigma_h^k)^{-1}\Lambda_h^k(\Omega_h^k)^{-1}$.*

*Proof of Lemma B.2.* Since $\lambda > 0$, both $\Omega_h^k$ and $\Sigma_h^k$ are invertible with:
$$
\begin{aligned}
(\Omega_h^k)^{-1} &= (\Sigma_h^k)^{-1} + (\Omega_h^k)^{-1} - (\Sigma_h^k)^{-1} \\
&= (\Sigma_h^k)^{-1} + (\Sigma_h^k)^{-1}\Sigma_h^k(\Omega_h^k)^{-1} - (\Sigma_h^k)^{-1}\Omega_h^k(\Omega_h^k)^{-1} \\
&= (\Sigma_h^k)^{-1} + (\Sigma_h^k)^{-1}\Lambda_h^k(\Omega_h^k)^{-1}
\end{aligned}
$$
$\square$

**Lemma B.3.** *Denote $\phi_h^k := \phi(s_h^k, a_h^k)$. Let $\lambda > 0$, then*
$$
\sum_{k=1}^{K} \left\|\phi_h^k\right\|_{(\Sigma_h^k)^{-1}\Lambda_h^k(\Omega_h^k)^{-1}} \leq \frac{1}{2}\sum_{k=1}^{K}(1 + \max_{k\in[K]} U_k + \tau_k)\left\|\phi_h^k\right\|_{(\Sigma_h^k)^{-1}}^2
$$

*Proof of Lemma B.3.* By definition and Trace of matrix, we have
$$
\left\|\phi_h^k\right\|_{(\Sigma_h^k)^{-1}\Lambda_h^k(\Omega_h^k)^{-1}} = \sqrt{(\phi_h^k)^{\mathrm{T}}(\Sigma_h^k)^{-1}\Lambda_h^k(\Omega_h^k)^{-1}\phi_h^k}
$$
$$
= \sqrt{\mathrm{Tr}[(\phi_h^k)^{\mathrm{T}}(\Sigma_h^k)^{-1}\Lambda_h^k(\Omega_h^k)^{-1}\phi_h^k]} = \sqrt{\mathrm{Tr}[(\Sigma_h^k)^{-1}\Lambda_h^k(\Omega_h^k)^{-1}\phi_h^k(\phi_h^k)^{\mathrm{T}}]}
$$
Denote $A = (\Sigma_h^k)^{-1}\Lambda_h^k$ and $B = (\Omega_h^k)^{-1}\phi_h^k(\phi_h^k)^{\mathrm{T}}$, then $A, B$ both have non-negative eigenvalues (by Lemma D.14) and this implies
$$
\mathrm{Tr}(AB) = \mathrm{Tr}\left(AB^{1/2}B^{1/2}\right) = \mathrm{Tr}\left(B^{1/2}AB^{1/2}\right) \leq \mathrm{Tr}\left(B^{1/2}(\mathrm{Tr}(A))IB^{1/2}\right) = \mathrm{Tr}(A)\,\mathrm{Tr}(B)
$$
and this implies
$$
\begin{aligned}
\left\|\phi_h^k\right\|_{(\Sigma_h^k)^{-1}\Lambda_h^k(\Omega_h^k)^{-1}} &= \sqrt{\mathrm{Tr}[AB]} \leq \sqrt{\mathrm{Tr}[A]\mathrm{Tr}[B]} \leq \frac{1}{2}\mathrm{Tr}(A) + \frac{1}{2}\mathrm{Tr}(B) \\
&= \frac{1}{2}\left\|\phi_h^k\right\|_{(\Omega_h^k)^{-1}}^2 + \frac{1}{2}\sum_{t=1}^{k-1}\mathbb{1}[t + \tau_t > k - 1]\left\|\phi_h^t\right\|_{(\Sigma_h^k)^{-1}}^2 \\
&\leq \frac{1 + \max_{k\in[K]} U_k}{2}\left\|\phi_h^k\right\|_{(\Sigma_h^k)^{-1}}^2 + \frac{1}{2}\sum_{t=1}^{k-1}\mathbb{1}[t + \tau_t > k - 1]\left\|\phi_h^t\right\|_{(\Sigma_h^k)^{-1}}^2
\end{aligned}
$$

where the last inequality uses Lemma D.15. Next, by changing the order summation, we have
$$
\sum_{k=1}^{K}\sum_{t=1}^{k-1}\mathbb{1}[t + \tau_t > k - 1]\left\|\phi_h^t\right\|_{(\Sigma_h^t)^{-1}}^2 = \sum_{t=1}^{K-1}\sum_{k=t+1}^{K}\mathbb{1}[t + \tau_t > k - 1]\left\|\phi_h^t\right\|_{(\Sigma_h^t)^{-1}}^2
$$
$$
= \sum_{t=1}^{K-1}\sum_{s=0}^{K-t-1}\mathbb{1}[\tau_t > s]\left\|\phi_h^t\right\|_{(\Sigma_h^t)^{-1}}^2 \leq \sum_{t=1}^{K-1}\sum_{s=0}^{\infty}\mathbb{1}[\tau_t > s]\left\|\phi_h^t\right\|_{(\Sigma_h^t)^{-1}}^2 = \sum_{t=1}^{K-1}\tau_t\left\|\phi_h^t\right\|_{(\Sigma_h^t)^{-1}}^2,
$$
which implies
$$
\begin{aligned}
&\sum_{k=1}^{K}\left\|\phi_h^k\right\|_{(\Sigma_h^k)^{-1}\Lambda_h^k(\Omega_h^k)^{-1}} \\
&\leq \frac{1 + \max_{k\in[K]} U_k}{2}\sum_{k=1}^{K}\left\|\phi_h^k\right\|_{(\Sigma_h^k)^{-1}}^2 + \frac{1}{2}\sum_{k=1}^{K}\sum_{t=1}^{k-1}\mathbb{1}[t + \tau_t > k - 1]\left\|\phi_h^t\right\|_{(\Sigma_h^k)^{-1}}^2 \\
&\leq \frac{1 + \max_{k\in[K]} U_k}{2}\sum_{k=1}^{K}\left\|\phi_h^k\right\|_{(\Sigma_h^k)^{-1}}^2 + \frac{1}{2}\sum_{k=1}^{K}\sum_{t=1}^{k-1}\mathbb{1}[t + \tau_t > k - 1]\left\|\phi_h^t\right\|_{(\Sigma_h^t)^{-1}}^2 \\
&\leq \frac{1 + \max_{k\in[K]} U_k}{2}\sum_{k=1}^{K}\left\|\phi_h^k\right\|_{(\Sigma_h^k)^{-1}}^2 + \sum_{t=1}^{K-1}\tau_t\left\|\phi_h^t\right\|_{(\Sigma_h^t)^{-1}}^2,
\end{aligned}
$$

where the second step uses $(\Sigma_h^k)^{-1} \succeq (\Sigma_h^t)^{-1}$ for $k \geq t$. $\qquad\square$

**Lemma B.4** (Bounding the delayed error)**.** *With probability* $1 - \delta/8$,

$$\sum_{h-1}^{H}\sum_{k=1}^{K}\left\|\phi(s_h^k, a_h^k)\right\|_{(\Omega_h^k)^{-1}} \leq H\sqrt{2dK\log((d+K)/d)} + dHD_{\tau,\delta,H,K}\log((d+K)/d).$$

*Here* $D_{\tau,\delta,H,K} := 1 + 2\mathbb{E}[\tau] + 2\sqrt{2\mathbb{E}[\tau]\log(\frac{24KH}{\delta})} + \frac{4}{3}\log(\frac{24KH}{\delta}) + D_{\tau,K,\frac{\delta}{16H}}$ *and* $D_{\tau,K,\delta}$ *is defined in [Lemma D.6](#).*

*Proof of [Lemma B.4](#).* Now Combine [Lemma B.2](#) and [Lemma B.3](#), we obtain

$$\sum_{k=1}^{K}\left\|\phi_h^k\right\|_{(\Omega_h^k)^{-1}} \leq \sum_{k=1}^{K}\left\|\phi_h^k\right\|_{(\Sigma_h^k)^{-1}} + \sum_{k=1}^{K}\left\|\phi_h^k\right\|_{(\Sigma_h^k)^{-1}\Lambda_h^k(\Omega_h^k)^{-1}}$$

$$\leq \underbrace{\sum_{k=1}^{K}\left\|\phi_h^k\right\|_{(\Sigma_h^k)^{-1}}}_{(*)} + \underbrace{\frac{1}{2}\sum_{k=1}^{K}(1 + \max_{k\in[K]}U_k + \tau_k)\left\|\phi_h^k\right\|_{(\Sigma_h^k)^{-1}}^2}_{(**)}.$$

For term $(*)$, since $\lambda = 1$, by Cauchy-Schwartz inequality and Elliptical Potential Lemma [D.8](#),

$$\sum_{k=1}^{K}\left\|\phi_h^k\right\|_{(\Sigma_h^k)^{-1}} \leq \sqrt{K\sum_{k=1}^{K}\left\|\phi_h^k\right\|_{(\Sigma_h^k)^{-1}}^2} \leq \sqrt{2K\log\left(\frac{\det(\Sigma_h^{K+1})}{\det(\Sigma_h^1)}\right)} \leq \sqrt{2dK\log((d+K)/d)}$$

For term $(**)$, by [Lemma D.15](#) and [Lemma D.6](#) and a union bound, with probability $1 - \delta/8$,

$$\frac{1}{2}\sum_{k=1}^{K}(1 + \max_{k\in[K]}U_k + \tau_k)\left\|\phi_h^k\right\|_{(\Sigma_h^k)^{-1}}^2$$

$$\leq \frac{1}{2}(1 + \max_{k\in[K]}U_k + \max_{k\in[K]}\tau_k)\sum_{k=1}^{K}\left\|\phi_h^k\right\|_{(\Sigma_h^k)^{-1}}^2$$

$$\leq \frac{1}{2}(1 + \max_{k\in[K]}U_k + \max_{k\in[K]}\tau_k)2d\log(1+K)$$

$$\leq d(1 + \mathbb{E}[\tau] + 2\sqrt{2\mathbb{E}[\tau]\log(\frac{24K}{\delta})} + \frac{4}{3}\log(\frac{24K}{\delta}) + \max_{k\in[K]}\tau_k)\log((d+K)/d)$$

$$\leq d(1 + \mathbb{E}[\tau] + 2\sqrt{2\mathbb{E}[\tau]\log(\frac{24K}{\delta})} + \frac{4}{3}\log(\frac{24K}{\delta}) + \mathbb{E}[\tau] + D_{\tau,\frac{\delta}{16}})\log((d+K)/d)$$

Denote $D_{\tau,\delta,K} := 1 + \mathbb{E}[\tau] + 2\sqrt{2\mathbb{E}[\tau]\log(\frac{24K}{\delta})} + \frac{4}{3}\log(\frac{24K}{\delta}) + \mathbb{E}[\tau] + D_{\tau,K,\frac{\delta}{16}}$, then we have with probability $1 - \delta/8$,

$$\sum_{k=1}^{K}\left\|\phi(s_h^k, a_h^k)\right\|_{(\Omega_h^k)^{-1}} \leq \sqrt{2dK\log((d+K)/d)} + dD_{\tau,\delta,K}\log((d+K)/d),$$

then apply a union bound over $h \in [H]$ to obtained the stated result. $\qquad\square$

## B.2 Proofs of Anti-concentration for Delayed-PSVI

In this section, we prove the optimism via anti-concentration for Delayed-PSVI. We first present two assisting lemmas.

**Lemma B.5** (Anti-concentration for Optimism)**.** *Suppose the event*

$$E = \{\left|\widehat{Q}_h^k(s,a) - (r_h^k + \mathbb{P}_h\widetilde{V}_{h+1}^k)(s,a)\right| \leq C_{\delta'}\left\|\phi(s,a)\right\|_{(\Omega_h^k)^{-1}}, \forall s, a, h, k\}$$

*holds. Choose* $\nu = C_{\delta'}$ *and* $M_\delta = \log(HK/\delta)/\log(64/63)$. *Then we have with probability* $1 - \delta$,

$$\widetilde{Q}_h^k(s,a) \geq (r_h + \mathbb{P}_h\widetilde{V}_{h+1}^k)(s,a), \ \forall (s,a) \in \mathcal{S} \times \mathcal{A}, h \in [H], k \in [K].$$

*Proof of Lemma B.5.* For the rest of the proof, we condition on the event

$$E = \left\{ \left| \widehat{Q}_h^k(s,a) - (r_h^k + \mathbb{P}_h \widetilde{V}_{h+1}^k)(s,a) \right| \leq C_{\delta'} \left\| \phi(s,a) \right\|_{(\Omega_h^k)^{-1}}, \; \forall s, a, h, k \right\}$$

where $\delta'$ will be specified later and $C_\delta$ is defined in the Lemma B.10. Also note

$$\widetilde{Q}_h^{k,m}(s,a) - \widehat{Q}_h^k(s,a) = \phi(s,a)^{\mathrm{T}}(\widetilde{w}_h^k - \widehat{w}_h^k) \sim \mathcal{N}(0, \nu^2 \phi(s,a)^{\mathrm{T}}(\Omega_h^k)^{-1}\phi(s,a))$$

$$\Leftrightarrow \frac{\widetilde{Q}_h^{k,m}(s,a) - \widehat{Q}_h^k(s,a)}{\sqrt{\nu^2 \phi(s,a)^{\mathrm{T}}(\Omega_h^k)^{-1}\phi(s,a)}} \sim \mathcal{N}(0,1).$$

Therefore,

$$\mathbb{P}\left( \widetilde{Q}_h^{k,m}(s,a) \geq (r_h + \mathbb{P}_h \widetilde{V}_{h+1}^k)(s,a), \forall s, a \, \Big| \, \widehat{Q}_h^k \right)$$

$$= \mathbb{P}\left( \frac{\widetilde{Q}_h^{k,m}(s,a) - \widehat{Q}_H^k(s,a)}{\sqrt{\nu^2 \phi(s,a)^{\mathrm{T}}(\Omega_h^k)^{-1}\phi(s,a)}} \geq \frac{(r_h + \mathbb{P}_h \widetilde{V}_{h+1})(s,a) - \widehat{Q}_h^k(s,a)}{\sqrt{\nu^2 \phi(s,a)^{\mathrm{T}}(\Omega_h^k)^{-1}\phi(s,a)}}, \; \forall s, a \, \Big| \, \widehat{Q}_h^k \right)$$

$$= \mathbb{P}\left( \mathcal{N}(0,1) \geq \frac{(r_h + \mathbb{P}_h \widetilde{V}_{h+1})(s,a) - \widehat{Q}_h^k(s,a)}{\sqrt{\nu^2 \phi(s,a)^{\mathrm{T}}(\Omega_h^k)^{-1}\phi(s,a)}}, \; \forall s, a \, \Big| \, \widehat{Q}_h^k \right)$$

$$\geq \mathbb{P}\left( \mathcal{N}(0,1) \geq C_{\delta'}/\nu \right)$$

$$\geq \frac{1}{2\sqrt{8\pi}} e^{-1/2} \geq \frac{1}{64},$$

where the first event uses the condition on $E$ and the second inequality chooses $\nu = C_{\delta'}$ and uses Lemma D.5. Apply Lemma B.6 with $f = r_h + \mathbb{P}_h \widetilde{V}_{h+1}^k$, for $M_\delta = \log(1/\delta)/\log(64/63)$,

$$\mathbb{P}\left( \widetilde{Q}_h^k(s,a) \geq (r_h + \mathbb{P}_h \widetilde{V}_{h+1}^k)(s,a), \forall s, a \, \Big| \, \widehat{Q}_h^k \right) \geq 1 - \delta.$$

By law of total expectation $\mathbb{E}[\mathbb{E}[\mathbf{1}_A | X]] = \mathbb{E}[\mathbf{1}_A] = \mathbb{P}[A]$, it implies

$$\mathbb{P}\left( \widetilde{Q}_h^k(s,a) \geq (r_h + \mathbb{P}_h \widetilde{V}_{h+1}^k)(s,a), \forall s, a \right) \geq 1 - \delta.$$

Apply a union bound for $h, k$, we have for $M_\delta = \log(HK/\delta)/\log(64/63)$, with probability $1 - \delta$,

$$\mathbb{P}\left( \widetilde{Q}_h^k(s,a) \geq (r_h + \mathbb{P}_h \widetilde{V}_{h+1}^k)(s,a), \forall s, a, h, k \right) \geq 1 - \delta.$$

$\square$

The following lemma is used to prove Lemma B.5.

**Lemma B.6.** *For any function $f : \mathcal{S} \times \mathcal{A} \mapsto \mathbb{R}$. For any $0 < \delta < 1$. Suppose for any $(k, h, m) \in [K] \times [H] \times [M]$, $\mathbb{P}\left( \widetilde{Q}_h^{k,m}(s,a) \geq f(s,a), \forall s, a \mid \widehat{Q}_h^k \right) \geq c$ for some constant $c > 0$. Let $M = \log(1/\delta)/\log(1/(1-c))$. Then*

$$\mathbb{P}\left( \widetilde{Q}_h^k(s,a) \geq f(s,a), \forall s, a \mid \widehat{Q}_h^k \right) \geq 1 - \delta.$$

*Proof of Lemma B.6.* For any fixed $(k, h) \in [K] \times [H]$, we have

$$\mathbb{P}\left( \exists (s,a) \; s.t. \; \max_{m \in [M]} \widetilde{Q}_h^{k,m}(s,a) \leq f(s,a) \mid \widehat{Q}_h^k \right)$$

$$= \mathbb{P}\left( \exists (s,a) \; s.t. \; \forall m \in [M], \; \widetilde{Q}_h^{k,m}(s,a) \leq f(s,a) \mid \widehat{Q}_h^k \right)$$

$$\leq \mathbb{P}\left( \forall m \in [M], \exists (s_m, a_m) \; s.t. \; \widetilde{Q}_h^{k,m}(s_m, a_m) \leq f(s_m, a_m) \mid \widehat{Q}_h^k \right)$$

$$= \prod_{m=1}^{M} \mathbb{P}\left( \exists (s,a) \; s.t. \; \widetilde{Q}_h^{k,m}(s,a) \leq f(s,a) \mid \widehat{Q}_h^k \right)$$

$$= \prod_{m=1}^{M} \left[ 1 - \mathbb{P}\left( \widetilde{Q}_h^{k,m}(s,a) \geq f(s,a), \forall s, a \mid \widehat{Q}_h^k \right) \right] \leq (1-c)^M = \delta,$$

then this implies

$$\mathbb{P}\left(\widetilde{Q}_h^k(s,a) \geq f(s,a), \forall s, a \mid \widehat{Q}_h^k\right) \geq 1 - \delta.$$

$\square$

With the above two lemmas, we are ready to prove the optimism achieved by Delayed-PSVI with respect to $\widetilde{Q}_h^k$.

**Lemma B.7** (Optimism). *For any $0 \leq \delta < 1$, we set the input in Algorithm 1 as $\nu = C_{\delta/4}$ and $M_\delta = \log(4HK/\delta)/\log(64/63)$, then with probability $1 - \delta/2$, we have*

$$\widetilde{Q}_h^k(s,a) \geq Q_h^*(s,a), \ \widetilde{V}_h^k(s) \geq V_h^*(s) \quad \forall s, a \in \mathcal{S} \times \mathcal{A}, \forall h \in [H], k \in [K].$$

*Here $C_\delta$ is defined in Lemma B.10.*

*Proof of Lemma B.7.* **Step1:** Suppose the event

$$E = \{\left|\widehat{Q}_h^k(s,a) - (r_h^k + \mathbb{P}_h\widetilde{V}_{h+1}^k)(s,a)\right| \leq C_{\delta'} \left\|\phi(s,a)\right\|_{(\Omega_h^k)^{-1}}, \ \forall s, a, h, k\}$$

holds. Choose $\nu = C_{\delta'}$ and $M_\delta = \log(4HK/\delta)/\log(64/63)$. Then we show, for any $h \in [H]$, with probability $1 - \delta/4$, $\widetilde{Q}_h^k(s,a) \geq Q_h^*(s,a)$, $\widetilde{V}_h^k(s) \geq V_h^*(s)$ for all $(s,a) \in \mathcal{S} \times \mathcal{A}$, $h \in [H]$, $k \in [K]$.

First, due to our choice of $M_\delta = \log(4HK/\delta)/\log(64/63)$, by Lemma B.5, with probability $1 - \delta/4$,

$$\widetilde{Q}_h^k(s,a) \geq (r_h + \mathbb{P}_h\widetilde{V}_{h+1}^k)(s,a), \ \forall(s,a) \in \mathcal{S} \times \mathcal{A}, h \in [H], k \in [K],$$

which we condition on.

Next, we finish the proof by backward induction. Base case: for $h = H + 1$, the value functions are zero, and thus $\widetilde{Q}_{H+1}^k \geq Q_{H+1}^*$ holds trivially, which also implies $\widetilde{V}_{H+1}^k \geq V_{H+1}^*$. Suppose the conclusion holds true for $h + 1$. Then for time step $h$ and any $k \in [K]$,

$$\begin{aligned}
\widetilde{Q}_h^k - Q_h^* &= \widetilde{Q}_h^k - (r_h + \mathbb{P}_h\widetilde{V}_{h+1}^k) + (r_h + \mathbb{P}_h\widetilde{V}_{h+1}^k) - Q_h^* \\
&\geq \widetilde{Q}_h^k - (r_h + \mathbb{P}_h\widetilde{V}_{h+1}^k) + (r_h + \mathbb{P}_H V_{h+1}^*) - Q_h^* \\
&= \widetilde{Q}_h^k - (r_h + \mathbb{P}_h\widetilde{V}_{h+1}^k) \geq 0
\end{aligned}$$

where the first inequality uses the induction hypothesis and the second inequality uses the condition. Lastly, $\widetilde{V}_h^k(\cdot) = \max_a \min\{\widetilde{Q}_h^k(\cdot,a), H - h + 1\} \leq \max_a \min\{Q_h^*(\cdot,a), H - h + 1\} = \max_a Q_h^*(\cdot,a) = V_h^*(\cdot)$. By induction, this finishes the Step1.

**Step2:** By Lemma B.10, with probability $1 - \delta/4$, for all $k \in [K], h \in [H], s \in \mathcal{S}, a \in \mathcal{A}$, it holds

$$\left|\widehat{Q}_h^k(s,a) - (r_h^k + \mathbb{P}_h\widetilde{V}_{h+1}^k)(s,a)\right| \leq C_{\delta/4} \left\|\phi(s,a)\right\|_{(\Omega_h^k)^{-1}}.$$

Therefore, in Step1, choose $\delta' = \delta/4$, and a union bound we obtain: for the choice $\nu = C_{\delta/4}$ and $M_\delta = \log(4HK/\delta)/\log(64/63)$, then with probability $1 - \delta/2$, we have

$$\widetilde{Q}_h^k(s,a) \geq Q_h^*(s,a), \ \widetilde{V}_h^k(s) \geq V_h^*(s) \ \forall(s,a) \in \mathcal{S} \times \mathcal{A}, \ h \in [H], \ k \in [K].$$

$\square$

## B.3 Proofs of Concentration for Delayed-PSVI

**Lemma B.8** (Pointwise Concentration). *Algorithm 1 guarantees that with probability $1 - \delta$, $\forall k \in [K], h \in [H], s \in \mathcal{S}, a \in \mathcal{A},$, it holds:*

$$\left|\min\{\widetilde{Q}_h^k(s,a), H - h + 1\} - (r_h^k + \mathbb{P}_h\widetilde{V}_{h+1}^k)(s,a)\right| \leq \beta \left\|\phi(s,a)\right\|_{(\Omega_h^k)^{-1}} + \frac{1}{K^3} \qquad (10)$$

*where $\beta := \sqrt{2\nu^2\log(4C_d HMK/\delta)} + \sqrt{8H^2\left[\frac{d}{2}\log\left(\frac{k+\lambda}{\lambda}\right) + dM\log(1 + \frac{2\sqrt{8k^3}C_{H,d,k,M,\delta/2}}{H\sqrt{\lambda}}) + \log\frac{4}{\delta}\right]} + 2\sqrt{\lambda}\sqrt{d}H$. In particular, here $\log C_d = d\log(1 + (8\sqrt{2\nu^2\log(4/\delta)/\lambda} + 8H\sqrt{d})K^3)$ and $C_{H,d,k,M,\delta} = 2H\sqrt{\frac{dk}{\lambda}} + \frac{\nu\sqrt{2d} + \nu\sqrt{2\log(M/\delta)}}{\sqrt{\lambda}}$.*

*Proof of Lemma B.8.* Recall that $|r_h^k + \mathbb{P}_h \widetilde{V}_{h+1}^k| \leq H - h + 1$, therefore $r_h^k + \mathbb{P}_h \widetilde{V}_{h+1}^k = \min\{r_h^k + \mathbb{P}_h \widetilde{V}_{h+1}^k, H - h + 1\}$. This implies $|\min\{\widehat{Q}_h^k(s,a), H - h + 1\} - r_h^k - [\mathbb{P}_h \widetilde{V}_{h+1}^k](s,a)| = |\min\{\widehat{Q}_h^k(s,a), H - h + 1\} - \min\{r_h^k + [\mathbb{P}_h \widetilde{V}_{h+1}^k](s,a), H - h + 1\}| \leq |\widehat{Q}_h^k(s,a) - r_h^k - [\mathbb{P}_h \widetilde{V}_{h+1}^k](s,a)|$. Hence

$$\left| \min\{\widehat{Q}_h^k(s,a), H - h + 1\} - r_h^k - [\mathbb{P}_h \widetilde{V}_{h+1}^k](s,a) \right| \leq \left| \widehat{Q}_h^k(s,a) - r_h^k - [\mathbb{P}_h \widetilde{V}_{h+1}^k](s,a) \right|$$

$$= \left| \widehat{Q}_h^k(s,a) - \widehat{Q}_h^k(s,a) + \widehat{Q}_h^k(s,a) - r_h^k - [\mathbb{P}_h \widetilde{V}_{h+1}^k](s,a) \right|$$

$$\leq \underbrace{\left| \widehat{Q}_h^k(s,a) - \widehat{Q}_h^k(s,a) \right|}_{R_1} + \underbrace{\left| \widehat{Q}_h^k(s,a) - r_h^k - [\mathbb{P}_h \widetilde{V}_{h+1}^k](s,a) \right|}_{R_2}.$$

The proof then directly follows Lemma B.9 and Lemma B.10 to bound $R_1$ and $R_2$ respectively (together with a union bound). $\square$

**Lemma B.9** (Concentration of $R_1$). *For any $0 < \delta < 1$, define the event $\widetilde{E}$ as*

$$\widetilde{E} = \left\{ \left| \widetilde{Q}_h^k(s,a) - \phi(s,a)^{\mathrm{T}} \widehat{w}_h^k \right| \leq \sqrt{2\nu^2 \log(2C_d HMK/\delta)} \, \|\phi(s,a)\|_{(\Omega_h^k)^{-1}} + \frac{1}{K^3}, \right.$$

$$\left. \forall k \in [K], h \in [H], s \in \mathcal{S}, a \in \mathcal{A} \right\}, \tag{11}$$

*then $\widetilde{E}$ happens with probability $1 - \delta$. Here $\log C_d = d \log(1 + (8\sqrt{2\nu^2 \log(2/\delta)/\lambda} + 8H\sqrt{d})K^3)$.*

*Proof of Lemma B.9.* In the Step1 and Step2, we abuse $\widetilde{w}_h^k$ to denote $\widetilde{w}_h^{k,m}$ for arbitrary $m$ to avoid notation redundancy.

In **Step1**: We first show for any $k \in [K], h \in [H], (s,a) \in \mathcal{S} \times \mathcal{A}$, with probability $1 - \delta$,

$$\left| \phi(s,a)^{\mathrm{T}} (\widetilde{w}_h^k - \widehat{w}_h^k) \right| \leq \sqrt{2\nu^2 \log(2/\delta)} \, \|\phi(s,a)\|_{(\Omega_h^k)^{-1}} \, .$$

Indeed, by design of Algorithm 1, $\widetilde{w}_h^k \sim \mathcal{N}(\widehat{w}_h^k, \nu^2 (\Omega_h^k)^{-1})$, which gives,

$$\phi(s,a)^{\mathrm{T}} (\widetilde{w}_h^k - \widehat{w}_h^k) \sim \mathcal{N}(0, \nu^2 \phi(s,a)^{\mathrm{T}} (\Omega_h^k)^{-1} \phi(s,a)).$$

Therefore, $\phi(s,a)^{\mathrm{T}} (\widetilde{w}_h^k - \widehat{w}_h^k)$ is $\nu^2 \phi(s,a)^{\mathrm{T}} (\Omega_h^k)^{-1} \phi(s,a)$-sub-Gaussian. By concentration of sub-Gaussian random variables, we have

$$\mathbb{P}\left( \left| \phi(s,a)^{\mathrm{T}} (\widetilde{w}_h^k - \widehat{w}_h^k) \right| \geq t \right) \leq 2 \exp\left( -\frac{t^2}{2\nu^2 \phi(s,a)^{\mathrm{T}} (\Omega_h^k)^{-1} \phi(s,a)} \right) := \delta$$

Solving for $\delta$ gives with probability $1 - \delta$,

$$\left| \phi(s,a)^{\mathrm{T}} (\widetilde{w}_h^k - \widehat{w}_h^k) \right| \leq \sqrt{2\nu^2 \phi(s,a)^{\top} (\Omega_h^k)^{-1} \phi(s,a) \log(2/\delta)} = \sqrt{2\nu^2 \log(2/\delta)} \, \|\phi(s,a)\|_{(\Omega_h^k)^{-1}}$$

**Step2:** For any $0 < \delta < 1$, define the event $\widetilde{E}$ as

$$\widetilde{E} = \left\{ \left| \phi(s,a)^{\mathrm{T}} \widetilde{w}_h^k - \phi(s,a)^{\mathrm{T}} \widehat{w}_h^k \right| \leq \sqrt{2\nu^2 \log(2C_d HK/\delta)} \, \|\phi(s,a)\|_{(\Omega_h^k)^{-1}} + \frac{1}{K^3}, \right.$$

$$\left. \forall k \in [K], h \in [H], s \in \mathcal{S}, a \in \mathcal{A} \right\}, \tag{12}$$

then $\widetilde{E}$ happens with probability $1 - \delta$. Here $\log C_d = d \log(1 + (8\sqrt{2\nu^2 \log(2/\delta)/\lambda} + 8H\sqrt{d})K^3)$.

In Lemma D.12, set $\theta = \widetilde{w}_h^k - \widehat{w}_h^k$ and $A = (\Omega_h^k)^{-1}$ and $B = 1/\lambda$, and let $\mathcal{V}$ be the $\frac{1}{2K^3}$-epsilon net for the class of values $\{|\langle \phi, \widetilde{w}_h^k - \widehat{w}_h^k \rangle| - C\sqrt{\phi^{\top} (\Omega_h^k)^{-1} \phi} : \|\phi\| \leq 1\}$ (where $C = \sqrt{2\nu^2 \log(2/\delta)}$), then it must also be the $\frac{1}{2K^3}$-epsilon net for the class of values $\mathcal{F} = \{|\langle \phi(s,a), \widetilde{w}_h^k - \widehat{w}_h^k \rangle| - C\sqrt{\phi(s,a)^{\top} (\Omega_h^k)^{-1} \phi(s,a)} : (s,a) \in \mathcal{S} \times \mathcal{A}\}$, let $\bar{\mathcal{V}}$ is the smallest subset of $\mathcal{V}$ such that it is

$\frac{1}{2K^3}$-epsilon net for the class of values $\mathcal{F}$. Then we can select $\mathcal{V}_{\mathcal{S}\times\mathcal{A}}$ to be the set of state-action pairs such that for any $f_\phi := |\langle\phi, \widetilde{w}_h^k - \widehat{w}_h^k\rangle| - C\sqrt{\phi^\top (\Omega_h^k)^{-1}\phi} \in \bar{\mathcal{V}}$, there exists $(s,a) \in \mathcal{V}_{\mathcal{S}\times\mathcal{A}}$ satisfies $|\langle\phi(s,a), \widetilde{w}_h^k - \widehat{w}_h^k\rangle| C\sqrt{\phi(s,a)^\top (\Omega_h^k)^{-1}\phi(s,a)|} - f_\phi \leq 1/2K^3$, then we have $\mathcal{V}_{\mathcal{S}\times\mathcal{A}}$ is a $1/K^3$-epsilon net of $\mathcal{F}$ and $|\mathcal{V}_{\mathcal{S}\times\mathcal{A}}| \leq |\bar{\mathcal{V}}| \leq |\mathcal{V}|$. Therefore,

$$\sup_{s,a} |\langle\phi(s,a), \widetilde{w}_h^k - \widehat{w}_h^k\rangle| - C\sqrt{\phi(s,a)^\top (\Omega_h^k)^{-1}\phi(s,a)}$$

$$\leq \sup_{(s,a)\in\mathcal{V}_{\mathcal{S}\times\mathcal{A}}} |\langle\phi(s,a), \widetilde{w}_h^k - \widehat{w}_h^k\rangle| - C\sqrt{\phi(s,a)^\top (\Omega_h^k)^{-1}\phi(s,a)} + 1/K^3$$

Then by a union bound over $(1 + (8\sqrt{2\nu^2\log(2/\delta)/\lambda} + 8H\sqrt{d})K^3)^d$, $H$ and $K$, we have the stated the result.

**Step3:** Note $\widetilde{Q}_h^k = \max_m \phi^{\mathrm{T}}\widetilde{w}_h^{k,m}$, hence by a union bound over $M$, we have

$$\left|\widetilde{Q}_h^k(s,a) - \phi(s,a)^{\mathrm{T}}\widehat{w}_h^k\right| = |\max_m \phi(s,a)^{\mathrm{T}}\widetilde{w}_h^{k,m} - \phi(s,a)^{\mathrm{T}}\widehat{w}_h^k|$$

$$\leq \max_m |\phi(s,a)^{\mathrm{T}}\widetilde{w}_h^{k,m} - \phi(s,a)^{\mathrm{T}}\widehat{w}_h^k|$$

$$\leq \sqrt{2\nu^2\log(2C_dHMK/\delta)}\,\|\phi(s,a)\|_{(\Omega_h^k)^{-1}} + \frac{1}{K^3}$$

for all $k, h, s, a$ with probability $1 - \delta$. Here the last inequality follows Step2, which completes the proof. $\square$

**Lemma B.10** (Concentration of $R_2$). *For any* $0 < \delta < 1$, *with probability* $1-\delta$, *for all* $k \in [K], h \in [H], s \in \mathcal{S}, a \in \mathcal{A}$, *it holds*

$$\left|\widehat{Q}_h^k(s,a) - (r_h^k + \mathbb{P}_h\widetilde{V}_{h+1}^k)(s,a)\right| \leq C_\delta \,\|\phi(s,a)\|_{(\Omega_h^k)^{-1}}$$

*where* $C_\delta = \sqrt{8H^2\left[\frac{d}{2}\log\left(\frac{k+\lambda}{\lambda}\right) + dM\log(1 + \frac{2\sqrt{8k^3}C_{H,d,k,M,\delta}}{H\sqrt{\lambda}}) + \log\frac{2}{\delta}\right]} + 2\sqrt{\lambda}\sqrt{d}H$ *and the quantity* $C_{H,d,k,M,\delta} = 2H\sqrt{\frac{dk}{\lambda}} + \frac{\nu\sqrt{2d}+\nu\sqrt{2\log(M/\delta)}}{\sqrt{\lambda}}$.

*Proof of Lemma B.10.* For any $(k,h) \in [K] \times [H]$ and $(s,a) \in \mathcal{S} \times \mathcal{A}$, denote

$$\phi(s,a)^{\mathrm{T}}w_h^k := (r_h^k + \mathbb{P}_h\widetilde{V}_{h+1}^k)(s,a), \text{where } w_h^k := \theta_h + \int_{\mathcal{S}}\widetilde{V}_{h+1}^k(s')\mathrm{d}\mu_h(s').$$

Recall $y_h^\tau = \mathbb{1}_{\tau,k-1} \cdot [r_h^\tau(s_h^\tau, a_h^\tau) + \widetilde{V}_{h+1}^k(s_{h+1}^\tau)]$ from Algorithm 1 and denote $\bar{y}_h^\tau := r_h^\tau(s_h^\tau, a_h^\tau) + \widetilde{V}_{h+1}^k(s_{h+1}^\tau)$. Then by definition,

$$\widehat{w}_h^k = (\Omega_h^k)^{-1}\sum_{\tau=1}^{k-1}\mathbb{1}_{\tau,k-1} \cdot \phi(s_h^\tau, a_h^\tau)y_h^\tau = (\Omega_h^k)^{-1}\sum_{\tau=1}^{k-1}\mathbb{1}_{\tau,k-1} \cdot \phi(s_h^\tau, a_h^\tau)\bar{y}_h^\tau.$$

From $\Omega_h^k$ defined in line 7 of Algorithm 1, we have $\Phi_h\Phi_h^{\mathrm{T}} = \Omega_h^k - \lambda I$. Plug it into the definition of $\widehat{w}_h^k$, we have

$$\widehat{w}_h^k = (\Omega_h^k)^{-1}\sum_{\tau=1}^{k-1}\mathbb{1}_{\tau,k-1} \cdot \phi(s_h^\tau, a_h^\tau)\left(\bar{y}_h^\tau - \phi(s_h^\tau, a_h^\tau)^{\mathrm{T}}w_h^k + \phi(s_h^\tau, a_h^\tau)^{\mathrm{T}}w_h^k\right)$$

$$= (\Omega_h^k)^{-1}\sum_{\tau=1}^{k-1}\mathbb{1}_{\tau,k-1} \cdot \phi(s_h^\tau, a_h^\tau)\left(\bar{y}_h^\tau - \phi(s_h^\tau, a_h^\tau)^{\mathrm{T}}w_h^k\right) + (\Omega_h^k)^{-1}\left(\Omega_h^k - \lambda I\right)w_h^k.$$

We then proceed to bound $\widehat{w}_h^k - w_h^k$, which gives

$$\widehat{w}_h^k - w_h^k = (\Omega_h^k)^{-1} \sum_{\tau=1}^{k-1} \mathbb{1}_{\tau,k-1} \cdot \phi(s_h^\tau, a_h^\tau) \left(\bar{y}_h^\tau - \phi(s_h^\tau, a_h^\tau)^{\mathrm{T}} w_h^k\right) - \lambda(\Omega_h^k)^{-1} w_h^k$$

$$= \underbrace{(\Omega_h^k)^{-1} \sum_{\tau=1}^{k-1} \mathbb{1}_{\tau,k-1} \cdot \phi(s_h^\tau, a_h^\tau) \left(\widetilde{V}_{h+1}^k(s_{h+1}^\tau) - \mathbb{P}_h \widetilde{V}_{h+1}^k(s_h^\tau, a_h^\tau)\right)}_{(i)} - \underbrace{\lambda(\Omega_h^k)^{-1} w_h^k}_{(ii)}.$$

**Term (i).** Since $\Omega_h^k$ is positive definite, multiplying the first term $(i)$ with $\phi(s,a)$ and by Cauchy-Schwartz inequality, we obtain,

$$\left|\phi(s,a)^{\mathrm{T}}(i)\right| \leq \|\phi(s,a)\|_{(\Omega_h^k)^{-1}} \left\|\sum_{\tau=1}^{k-1} \mathbb{1}_{\tau,k-1} \cdot \phi(s_h^\tau, a_h^\tau) \left(\widetilde{V}_{h+1}^k(s_{h+1}^\tau) - \mathbb{P}_h \widetilde{V}_{h+1}^k(s_h^\tau, a_h^\tau)\right)\right\|_{(\Omega_h^k)^{-1}}.$$

Apply Lemma B.11, we have with probability at least $1 - \delta$, for any $(k,h) \in [K] \times [H]$, and $(s,a) \in \mathcal{S} \times \mathcal{A}$,

$$\left|\phi(s,a)^{\mathrm{T}}(i)\right| \leq C_1 \|\phi(s,a)\|_{(\Omega_h^k)^{-1}}, \tag{13}$$

where $C_1 = \sqrt{8H^2 \left[\frac{d}{2} \log\left(\frac{k+\lambda}{\lambda}\right) + dM \log(1 + \frac{2\sqrt{8k^3}C_{H,d,k,M,\delta}}{H\sqrt{\lambda}}) + \log \frac{2}{\delta}\right]}$.

**Term (ii).** By Lemma B.12, $\forall (s,a) \in \mathcal{S} \times \mathcal{A}$, and $(k,h) \in [K] \times [H]$, $\left|\phi(s,a)^{\mathrm{T}}(ii)\right|$ can be bounded as

$$\left|\phi(s,a)^{\mathrm{T}}(ii)\right| = \lambda \left|\phi(s,a)^{\mathrm{T}}(\Omega_h^k)^{-1} w_h^k\right| \leq 2\sqrt{\lambda}\sqrt{d}H \|\phi(s,a)\|_{(\Omega_h^k)^{-1}}. \tag{14}$$

Combining (13), (14), we have with probability $1 - \delta$, for any $(k,h) \in [K] \times [H]$ and $(s,a) \in \mathcal{S} \times \mathcal{A}$,

$$\left|\widehat{Q}_h^k(s,a) - (r_h^k + \mathbb{P}_h \widetilde{V}_{h+1}^k)(s,a)\right| = \left|\phi(s,a)^{\mathrm{T}}(\widehat{w}_h^k - w_h^k)\right| \leq \left|\phi(s,a)^{\mathrm{T}}(i)\right| + \left|\phi(s,a)^{\mathrm{T}}(ii)\right|$$

$$\leq (C_1 + 2\sqrt{\lambda}\sqrt{d}H) \|\phi(s,a)\|_{(\Omega_h^k)^{-1}},$$

This concludes the proof. $\qquad\square$

**Lemma B.11.** *For any $0 < \delta < 1$, with probability $1 - \delta$, we have $\forall (k,h) \in [K] \times [H]$,*

$$\left\|\sum_{\tau=1}^{k-1} \mathbb{1}_{\tau,k-1} \cdot \phi(s_h^\tau, a_h^\tau) \left(\widetilde{V}_{h+1}^k(s_{h+1}^\tau) - \mathbb{P}_h \widetilde{V}_{h+1}^k(s_h^\tau, a_h^\tau)\right)\right\|_{(\Omega_h^k)^{-1}}^2$$

$$\leq 8H^2 \left[\frac{d}{2} \log\left(\frac{k+\lambda}{\lambda}\right) + dM \log(1 + \frac{2\sqrt{8k^3}C_{H,d,k,M,\delta}}{H\sqrt{\lambda}}) + \log \frac{2}{\delta}\right],$$

*here $C_{H,d,k,M,\delta} = 2H\sqrt{\frac{dk}{\lambda}} + \frac{\nu\sqrt{2d} + \nu\sqrt{2\log(M/\delta)}}{\sqrt{\lambda}}$.* [8]

*Proof of Lemma B.11.* First note that

$$\widetilde{V}_h^k(\cdot) := \max_a \min\{\widehat{Q}_h^k(\cdot,a), (H-h+1)\} = \max_a \min \max_m \{\widetilde{Q}_h^{k,m}, (H-h+1)\}$$

$$= \max_a \min\{\max_m \phi(\cdot,a)^{\mathrm{T}} \widetilde{w}_h^{k,m}, (H-h+1)\}.$$

Recall that $(\Omega_h^k)^{1/2}(\widetilde{w}_h^{k,m} - \widehat{w}_h^k)/\nu \sim \mathcal{N}(0, I_d)$, then by Lemma D.7, with probability $1 - \delta/2$, we have

$$\frac{\sqrt{\lambda}}{\nu} \left\|\widetilde{w}_h^{k,m} - \widehat{w}_h^k\right\| \leq \frac{1}{\nu} \left\|(\Omega_h^k)^{1/2}(\widetilde{w}_h^{k,m} - \widehat{w}_h^k)\right\| \leq \sqrt{2d} + \sqrt{2\log(1/\delta)}.$$

---

[8]Note here $\nu$ is in the line 10 of Algorithm 1. At the end we will choose $\nu$ to be $\mathrm{Poly}(H,d,K)$ and this will not affect the overall dependence of the guarantee since $C_{H,d,k,M,\delta}$ is inside the log term.

Apply the union bound over all $m$, then above implies with probability $1 - \delta/2$, $\forall m \in [M]$

$$\left\|\widetilde{w}_h^{k,m}\right\| \le \left\|\widehat{w}_h^k\right\| + \frac{\nu\sqrt{2d} + \nu\sqrt{2\log(M/\delta)}}{\sqrt{\lambda}} \le 2H\sqrt{\frac{dk}{\lambda}} + \frac{\nu\sqrt{2d} + \nu\sqrt{2\log(M/\delta)}}{\sqrt{\lambda}} := C_{H,d,k,M,\delta}. \quad (15)$$

Now consider the function class $\bar{\mathcal{V}} := \{\max_a \max_m \phi(\cdot, a)^{\mathrm{T}} w^m : \|w^m\| \le C_{H,d,k,M,\delta}\}$, so by Lemma D.13 the $\epsilon$-log covering number for $\bar{\mathcal{V}}$ is $dM\log(1 + \frac{2C_{H,d,k,M,\delta}}{\epsilon})$. Since $\min\{\cdot, \cdot\}$ is a non-expansive operator, the $\epsilon$-log covering number for the function class $\mathcal{V} := \{\max_a \min\{\max_m \phi(\cdot, a)^{\mathrm{T}} w^m, (H - h + 1)\} : \|w^m\| \le C_{H,d,k,M,\delta}\}$, is at most $dM\log(1 + \frac{2C_{H,d,k,M,\delta}}{\epsilon})$. Hence, for any $V \in \mathcal{V}$, there exists $V'$ in the $\epsilon$-covering such that $V = V' + \Delta_V$ with $\|\Delta_V\|_\infty \le \epsilon$. Then with probability $1 - \delta/2$,

$$\left\|\sum_{\tau=1}^{k-1} \mathbb{1}_{\tau,k-1} \phi(s_h^\tau, a_h^\tau) \left(V(s_{h+1}^\tau) - \mathbb{P}_h V(s_h^\tau, a_h^\tau)\right)\right\|_{(\Omega_h^k)^{-1}}^2$$

$$\le 2 \left\|\sum_{\tau=1}^{k-1} \mathbb{1}_{\tau,k-1} \phi(s_h^\tau, a_h^\tau) \left(V'(s_{h+1}^\tau) - \mathbb{P}_h V'(s_h^\tau, a_h^\tau)\right)\right\|_{(\Omega_h^k)^{-1}}^2$$

$$+ 2 \left\|\sum_{\tau=1}^{k-1} \mathbb{1}_{\tau,k-1} \phi(s_h^\tau, a_h^\tau) \left(\Delta_V(s_{h+1}^\tau) - \mathbb{P}_h \Delta_V(s_h^\tau, a_h^\tau)\right)\right\|_{(\Omega_h^k)^{-1}}^2 \quad (16)$$

$$\le 2 \left\|\sum_{\tau=1}^{k-1} \mathbb{1}_{\tau,k-1} \phi(s_h^\tau, a_h^\tau) \left(V'(s_{h+1}^\tau) - \mathbb{P}_h V'(s_h^\tau, a_h^\tau)\right)\right\|_{(\Omega_h^k)^{-1}}^2 + \frac{8k^2\epsilon^2}{\lambda}$$

$$\le 4H^2 \left[\frac{d}{2}\log\left(\frac{k+\lambda}{\lambda}\right) + dM\log(1 + \frac{2C_{H,d,k,M,\delta}}{\epsilon}) + \log\frac{2}{\delta}\right] + \frac{8k^2\epsilon^2}{\lambda}$$

where the second inequality can be conducted using a direct calculation and the third inequality uses Lemma D.9 and a union bound over the covering number. Now by (15) and (16) and a union bound, we have for any $\epsilon > 0$, with probability $1 - \delta$,

$$\left\|\sum_{\tau=1}^{k-1} \phi(s_h^\tau, a_h^\tau) \left(\widetilde{V}_{h+1}^k(s_{h+1}^\tau) - \mathbb{P}_h \widetilde{V}_{h+1}^k(s_h^\tau, a_h^\tau)\right)\right\|_{(\Omega_h^k)^{-1}}^2$$

$$\le 4H^2 \left[\frac{d}{2}\log\left(\frac{k+\lambda}{\lambda}\right) + dM\log(1 + \frac{2C_{H,d,k,M,\delta}}{\epsilon}) + \log\frac{2}{\delta}\right] + \frac{8k^2\epsilon^2}{\lambda}$$

$$\le 8H^2 \left[\frac{d}{2}\log\left(\frac{k+\lambda}{\lambda}\right) + dM\log(1 + \frac{2\sqrt{8k^3}C_{H,d,k,M,\delta}}{H\sqrt{\lambda}}) + \log\frac{2}{\delta}\right],$$

where the last step choose $\epsilon^2 = H^2\lambda/8k^2$ so $\frac{8k^2\epsilon^2}{\lambda} \le 4H^2$. Lastly, apply the union bound over $H, K$ to obtain the stated result. $\square$

**Lemma B.12.** $\forall (s, a) \in \mathcal{S} \times \mathcal{A}, h \in [H], k \in [K]$, it holds that

$$\left|\phi(s,a)^{\mathrm{T}}(\Omega_h^k)^{-1} w_h^k\right| \le \frac{2}{\sqrt{\lambda}}\sqrt{d}H \left\|\phi(s,a)\right\|_{(\Omega_h^k)^{-1}}.$$

*Proof of Lemma B.12.* Note that the precision matrix $\Omega_h^k$ for any step $h$ and episode $k$ is always positive definite. By Cauchy-Schwartz inequality and Lemma D.1,

$$\left|\phi(s,a)^{\mathrm{T}}(\Omega_h^k)^{-1} w_h^k\right| = \left|\phi(s,a)^{\mathrm{T}}(\Omega_h^k)^{-1/2}(\Omega_h^k)^{-1/2} w_h^k\right|$$

$$\le \left\|\phi(s,a)\right\|_{(\Omega_h^k)^{-1}} \left\|w_h^k\right\|_{(\Omega_h^k)^{-1}}$$

$$\le \left\|\phi(s,a)\right\|_{(\Omega_h^k)^{-1}} \sqrt{\left\|w_h^k\right\|^2 \left\|(\Omega_h^k)^{-1}\right\|}$$

$$\le \left\|\phi(s,a)\right\|_{(\Omega_h^k)^{-1}} \left\|w_h^k\right\| \frac{1}{\sqrt{\lambda_{\min}(\Omega_h^k)}}$$

Note that $\lambda_{\min}(\Omega_h^k) \ge \lambda$. Applying Lemma D.3 for $\left\|w_h^k\right\|$ completes the proof. $\square$

## C  Regret Analysis for Delayed-LPSVI

*Proof of Theorem 2.* The proof structure is similar to that of Theorem 1. We proceed by bounding $\Delta_{opt}^k$ and $\Delta_{est}^k$ respectively.

**Step 1: bound regret from optimism.** By Lemma C.5, with probability $1 - \delta/2$,
$$\Delta_{opt}^k := V_1^*(s_1^k) - \widetilde{V}_1^k(s_1^k) \leq 0, \quad \forall k \in [K].$$

**Step 2: bound regret from estimation error.** We first condition on the event that

$$\mathcal{E} := \{\|\min\{\widetilde{Q}_h^k(s,a), H-h+1\} - (r_h^k + \mathbb{P}_h \widetilde{V}_{h+1}^k)(s,a)| \leq \beta \|\phi(s,a)\|_{(\Omega_h^k)^{-1}} + \frac{1}{K^3}, \ \forall s,a,h,k\},$$

with $\beta := \sqrt{2\gamma \log(16 C_d HMK/\delta)} + \sqrt{8H^2 \left[\frac{d}{2} \log\left(\frac{k+\lambda}{\lambda}\right) + dM \log(1 + \frac{2\sqrt{8k^3} C_{H,d,k,M,\delta/8}}{H\sqrt{\lambda}}) + \log \frac{16}{\delta}\right]} + 2\sqrt{\lambda}\sqrt{d}H$. Here $C_d$ and $C_{H,d,k,M,\delta}$ are defined in Lemma C.6.

Similarly, define
$$\zeta_h^k = \mathbb{E}[\widetilde{V}_{h+1}^k\left(s_{h+1}^k\right) - V_{h+1}^{\pi_k}\left(s_{h+1}^k\right) | s_h^k, a_h^k] - \widetilde{V}_{h+1}^k\left(s_{h+1}^k\right) + V_{h+1}^{\pi_k}\left(s_{h+1}^k\right).$$

Then by Lemma C.1,

$$
\begin{aligned}
\sum_{k=1}^K \Delta_{est}^k &= \sum_{k=1}^K \widetilde{V}_1^k\left(s_1^k\right) - V_1^{\pi_k}\left(s_1^k\right) \\
&\leq \sum_{k=1}^K \left(\widetilde{V}_2^k\left(s_2^k\right) - V_2^{\pi_k}\left(s_2^k\right) + \zeta_1^k + \beta \left\|\phi(s_1^k, a_1^k)\right\|_{(\Omega_1^k)^{-1}} + \frac{1}{K^3}\right) \qquad (17)\\
&\leq \sum_{k=1}^K \sum_{h=1}^H \zeta_h^k + \beta \sum_{k=1}^K \sum_{h=1}^H \left\|\phi(s_h^k, a_h^k)\right\|_{(\Omega_h^k)^{-1}} + \frac{H}{K^2}.
\end{aligned}
$$

By definition, $|\zeta_h^k| \leq 2H$ for all $h \in [H], k \in [K]$, therefore $\{\zeta_h^k\}$ is a martingale difference sequence. By Azuma-Hoeffding's inequality,

$$\mathbb{P}\left(\sum_{k=1}^K \sum_{h=1}^H \zeta_h^k > t\right) \geq \exp\left(\frac{-t^2}{2K \cdot H^3}\right) := \delta/8, \quad \forall t > 0.$$

Thus, with probability $1 - \delta/8$,
$$\sum_{k=1}^K \sum_{h=1}^H \zeta_h^k \leq \sqrt{2KH^3 \cdot \log(8/\delta)} = \sqrt{2H^2 T \cdot \log(8/\delta)}. \qquad (18)$$

**Step 3: bounding the delayed error.** By Lemma B.4, with probability $1 - \delta/8$,
$$\beta \sum_{k=1}^K \sum_{h=1}^H \left\|\phi(s_h^k, a_h^k)\right\|_{(\Omega_h^k)^{-1}} \leq \beta H \sqrt{2dK \log((d+K)/d)} + \beta dH D_{\tau,\delta,H,K} \log((d+K)/d). \qquad (19)$$

Here $D_{\tau,\delta,H,K} := 1 + 2\mathbb{E}[\tau] + 2\sqrt{2\mathbb{E}[\tau] \log(\frac{24KH}{\delta})} + \frac{4}{3} \log(\frac{24KH}{\delta}) + D_{\tau,K,\frac{\delta}{16H}}$ and $D_{\tau,K,\delta}$ is defined in Lemma D.6. By Lemma C.6, event $\mathcal{E}$ holds with probability $1 - \delta/4$, by a union bound with (18) and (19), we have with probability $1 - \delta/2$,

$$\sum_{k=1}^K \Delta_{est}^k \leq \sqrt{2H^2 T \cdot \log(8/\delta)} + \beta H \sqrt{2dK \log((d+K)/d)} + \beta dH D_{\tau,\delta,H,K} \log((d+K)/d) + \frac{H}{K^2}.$$

Finally, by a union bound over Step1, Step2 and Step3, we obtain with probability $1 - \delta$,

$$
\begin{aligned}
R(T) = \sum_{k=1}^K \Delta_{opt}^k + \sum_{k=1}^K \Delta_{est}^k &\leq \sum_{k=1}^K \Delta_{est}^k \\
&\leq \sqrt{2H^2 T \cdot \log(8/\delta)} + \beta H \sqrt{2dK \log((d+K)/d)} + \beta dH D_{\tau,\delta,H,K} \log((d+K)/d) + \frac{H}{K^2} \\
&\leq c\sqrt{d^3 H^3 T \iota} + c' d^2 H^2 \mathbb{E}[\tau] \iota + O(\iota)
\end{aligned}
$$

where $c > 0$ is some universal constant and $\iota$ is a Polylog term of $H, d, K, \delta$. Similarly, we can bound $\beta \leq CdH\iota_\delta$ for some universal constant $C$, and it is readily to verify $D_{\tau,\delta,H,K}$ is bounded by $c'\mathbb{E}[\tau]\iota + O(\iota)$. □

**Lemma C.1.** *Define* $\zeta_h^k = \mathbb{E}[\widetilde{V}_{h+1}^k\left(s_{h+1}^k\right) - V_{h+1}^{\pi_k}\left(s_{h+1}^k\right)|s_h^k, a_h^k] - \widetilde{V}_{h+1}^k\left(s_{h+1}^k\right) + V_{h+1}^{\pi_k}\left(s_{h+1}^k\right)$ *and condition on the event* (21) *in Lemma C.6. Then for all* $k \in [K]$, $h \in [H]$, *the following holds,*

$$\widetilde{V}_h^k\left(s_h^k\right) - V_h^{\pi_k}\left(s_h^k\right) \leq \widetilde{V}_{h+1}^k\left(s_{h+1}^k\right) - V_{h+1}^{\pi_k}\left(s_{h+1}^k\right) + \zeta_{h+1}^k + \beta\left\|\phi(s_h^k, a_h^k)\right\|_{(\Omega_h^k)^{-1}} + \frac{1}{K^3}.$$

*Proof of Lemma C.1.* By the event defined in (21), the proof follows exactly as in that of Lemma B.1. □

## C.1 Convergence of Langevin Monte Carlo

The following lemma is crucial to prove the optimism and bound the error in Langevin analysis. For ease of notation, within the episode $k$, we simply use $\eta$ to denote $\eta_k$ for conciseness.

**Lemma C.2** (Convergence of LMC). *Denote* $\{\widetilde{w}_h^{k,m}\}_{m\in[M]}$ *to be the weights returned by Line 6 of Algorithm 2. Set* $\eta = \frac{1}{4\lambda_{\max}(\Omega_h^k)}$, *we have*

$$\widetilde{w}_h^{k,m} \sim \mathcal{N}(A_{h,k}^{N_k}w_0 + (I - A_{h,k}^{N_k})\widehat{w}_h^k, \Theta_h^k) \quad \forall m \in [M]$$

*where*

$$A_{h,k} := I - 2\eta\Omega_h^k$$

$$\Omega_h^k := \lambda I + \sum_{k=1}^K \phi_h(s_h^k, a_h^k)\phi_h(s_h^k, a_h^k)^T$$

$$\widehat{w}_h^k := (\Omega_h^k)^{-1}\sum_{\tau=1}^{k-1} \phi_h(s_h^\tau, a_h^\tau)y_h^\tau$$

$$\Theta_h^k := \gamma(I - A_{h,k}^{2N_k})(\Omega_h^k)^{-1}(I + A_{h,k})^{-1}.$$

*Furthermore, we have*

$$\frac{\gamma}{2}\left(1 - (1 - \frac{1}{2\kappa_h})^{2N_k}\right)(\Omega_h^k)^{-1} \prec \Theta_h^k \prec \gamma(\Omega_h^k)^{-1},$$

*where* $\kappa_h := \frac{\lambda_{\max}(\Omega_h^k)}{\lambda_{\min}(\Omega_h^k)}$ *is the condition number.*

*Proof of Lemma C.2.* Let $b_h^k := \sum_{\tau=1}^{k-1} \phi(s_h^\tau, a_h^\tau)y_h^\tau$, then

$$\nabla L_h^k(w) = 2\Omega_h^k w - 2b_h^k.$$

Therefore, fix $h, k, m$, and within the Algorithm 3 we have

$$\begin{aligned}
w_N &= w_{N-1} - 2\eta(\Omega_h^k \cdot w_{N-1} - b_h^k) + \sqrt{2\eta\gamma}\epsilon_N \\
&= (I - 2\eta\Omega_h^k)w_{N-1} + 2\eta b_h^k + \sqrt{2\eta\gamma}\epsilon_N \\
&= A_{h,k}w_{N-1} + 2\eta b_h^k + \sqrt{2\eta\gamma}\epsilon_N \\
&= A_{h,k}^N w_0 + 2\eta\sum_{l=0}^{N-1} A_{h,k}^l b_h^k + \sqrt{2\eta\gamma}\sum_{l=0}^{N-1} A_{h,k}^l\epsilon_{N-l} \\
&= A_{h,k}^N w_0 + (I - A_{h,k}^N)\widehat{w}_h^k + \sqrt{2\eta\gamma}\sum_{l=0}^{N-1} A_{h,k}^l\epsilon_{N-l}
\end{aligned}$$

where the last equality uses $(\Omega_h^k)^{-1}b_h^k = \widehat{w}_h^k$ and $I \succ I - 2\eta\Omega_h^k \succ \mathbf{0}$, so $\sum_{l=0}^{N-1} A^l = (I - A^N)(I - A)^{-1}$. Since $\epsilon_i$ are i.i.d gaussian noise, from the above we directly have

$$w_N \sim \mathcal{N}(A_{h,k}^N w_0 + (I - A_{h,k}^N)\widehat{w}_h^k, \Theta_h^k)$$

where

$$\Theta_h^k = \text{Cov}[\sqrt{2\eta\gamma} \sum_{l=0}^{N-1} A_{h,k}^l \epsilon_{N-l}] = 2\eta\gamma \cdot \text{Cov}[\sum_{l=0}^{N-1} A_{h,k}^l \epsilon_{N-l}]$$

$$= 2\eta\gamma \cdot \sum_{l=0}^{N-1} A_{h,k}^{2l} = 2\eta\gamma(I - A_{h,k}^{2N})(I - A_{h,k}^2)^{-1}$$

$$= \gamma(I - A_{h,k}^{2N_k})(\Omega_h^k)^{-1}(I + A_{h,k})^{-1}.$$

Next, due to the choice of $\eta = \frac{1}{4\lambda_{\max}(\Omega_h^k)}$, we have

$$\frac{1}{2}I \prec A_{h,k} = I - 2\eta\Omega_h^k \prec (1 - 2\eta\lambda_{\min}(\Omega_h^k))I$$

$$\Rightarrow \frac{1}{2^{2N}}I \prec A_{h,k}^{2N} \prec (1 - 2\eta\lambda_{\min}(\Omega_h^k))^{2N}I \quad\quad (20)$$

$$\Rightarrow \left(1 - (1 - 2\eta\lambda_{\min}(\Omega_h^k))^{2N}\right)I \prec I - A_{h,k}^{2N} \prec (1 - \frac{1}{2^{2N}})I$$

In addition,

$$\frac{1}{2}I \prec A_{h,k} = I - 2\eta\Omega_h^k \prec (1 - 2\eta\lambda_{\min}(\Omega_h^k))I$$

$$\Rightarrow \frac{3}{2}I \prec I + A_{h,k} \prec (2 - 2\eta\lambda_{\min}(\Omega_h^k))I$$

$$\Rightarrow \frac{1}{2 - 2\eta\lambda_{\min}(\Omega_h^k)}I \prec (I + A_{h,k})^{-1} \prec \frac{2}{3}I$$

The above two implies

$$\gamma\frac{\left(1 - (1 - 2\eta\lambda_{\min}(\Omega_h^k))^{2N}\right)}{2 - 2\eta\lambda_{\min}(\Omega_h^k)}(\Omega_h^k)^{-1} \prec \Theta_h^k \prec \gamma\frac{2}{3}(1 - \frac{1}{2^{2N}})(\Omega_h^k)^{-1}$$

$$\Rightarrow \gamma\frac{\left(1 - (1 - 2\eta\lambda_{\min}(\Omega_h^k))^{2N}\right)}{2}(\Omega_h^k)^{-1} \prec \Theta_h^k \prec \gamma(\Omega_h^k)^{-1}$$

Replacing $N$ with $N_k$ and $w_N$ with $\widetilde{w}_h^{k,m}$ for all $m \in [M]$ completes the proof. $\square$

## C.2 Proofs of optimism for Delayed-LPSVI

**Lemma C.3** (Anti-concentration for Optimism). *Suppose the event*

$$E = \{\left|\widehat{Q}_h^k(s,a) - (r_h^k + \mathbb{P}_h\widetilde{V}_{h+1}^k)(s,a)\right| \le C_{\delta'}\|\phi(s,a)\|_{(\Omega_h^k)^{-1}}, \ \forall s, a, h, k\}$$

*holds. Choose* $N_k \ge \max\{\log(\frac{32H^2(K+\lambda)dk}{\gamma\lambda} + 1)/[2\log(1/(1 - \frac{1}{2\kappa_h}))], \frac{\log 2}{2\log(1/(1 - \frac{1}{2\kappa_h}))}\}$, $\gamma = 16C_{\delta'}^2$, *and* $M_\delta = \log(HK/\delta)/\log(64/63)$. *Then we have with probability* $1 - \delta$,

$$\widetilde{Q}_h^k(s,a) \ge (r_h + \mathbb{P}_h\widetilde{V}_{h+1}^k)(s,a), \ \forall(s,a) \in \mathcal{S} \times \mathcal{A}, h \in [H], k \in [K].$$

*Proof of Lemma C.3.* For the rest of the proof, we condition on the event

$$E = \{\left|\widehat{Q}_h^k(s,a) - (r_h^k + \mathbb{P}_h\widetilde{V}_{h+1}^k)(s,a)\right| \le C_{\delta'}\|\phi(s,a)\|_{(\Omega_h^k)^{-1}}, \ \forall s, a, h, k\}$$

where $\delta'$ will be specified later and $C_\delta$ is defined in the Lemma C.9.

$$\phi(s,a)^{\mathrm{T}}(\widetilde{w}_h^k - (I - A_{h,k}^{N_k})\widehat{w}_h^k) \sim \mathcal{N}(0, \phi(s,a)^{\mathrm{T}}\Theta_h^k\phi(s,a)).$$

Also note

$$\widetilde{Q}_h^{k,m}(s,a) - \phi(s,a)^{\mathrm{T}}(I - A_{h,k}^{N_k})\widehat{w}_h^k \sim \mathcal{N}(0, \phi(s,a)^{\mathrm{T}}\Theta_h^k\phi(s,a))$$

$$\Leftrightarrow \frac{\widetilde{Q}_h^{k,m}(s,a) - \phi(s,a)^{\mathrm{T}}(I - A_{h,k}^{N_k})\widehat{w}_h^k}{\sqrt{\phi(s,a)^{\mathrm{T}}\Theta_h^k\phi(s,a)}} \sim \mathcal{N}(0,1).$$

Therefore,

$$\mathbb{P}\left(\widetilde{Q}_h^{k,m}(s,a) \geq (r_h + \mathbb{P}_h\widetilde{V}_{h+1}^k)(s,a), \forall s,a\right)$$

$$=\mathbb{P}\left(\frac{\widetilde{Q}_h^{k,m}(s,a) - \phi(s,a)^{\mathrm{T}}(I - A_{h,k}^{N_k})\widehat{w}_h^k}{\sqrt{\phi(s,a)^{\mathrm{T}}\Theta_h^k\phi(s,a)}} \geq \frac{(r_h + \mathbb{P}_h\widetilde{V}_{h+1})(s,a) - \phi(s,a)^{\mathrm{T}}(I - A_{h,k}^{N_k})\widehat{w}_h^k}{\sqrt{\phi(s,a)^{\mathrm{T}}\Theta_h^k\phi(s,a)}}, \forall s,a\right)$$

$$=\mathbb{P}\left(\mathcal{N}(0,1) \geq \frac{(r_h + \mathbb{P}_h\widetilde{V}_{h+1})(s,a) - \phi(s,a)^{\mathrm{T}}(I - A_{h,k}^{N_k})\widehat{w}_h^k}{\sqrt{\phi(s,a)^{\mathrm{T}}\Theta_h^k\phi(s,a)}}, \forall s,a\right)$$

$$\geq\mathbb{P}\left(\mathcal{N}(0,1) \geq \frac{(r_h + \mathbb{P}_h\widetilde{V}_{h+1})(s,a) - \phi(s,a)^{\mathrm{T}}(I - A_{h,k}^{N_k})\widehat{w}_h^k}{\sqrt{\frac{\gamma}{2}\left(1 - (1 - \frac{1}{2\kappa_h})^{2N_k}\right)\phi(s,a)^{\mathrm{T}}(\Omega_h^k)^{-1}\phi(s,a)}}, \forall s,a\right)$$

$$\geq\mathbb{P}\left(\mathcal{N}(0,1) \geq 1\right) \geq \frac{1}{2\sqrt{8\pi}}e^{-1/2} \geq \frac{1}{64},$$

where the first two inequalities follow Lemma C.2 and Lemma C.4 respectively and the thrid inequality results from Lemma D.5. Applying Lemma B.6 with $f = r_h + \mathbb{P}_h\widetilde{V}_{h+1}^k$ and without conditioning, for $M_\delta = \log(1/\delta)/\log(64/63)$,

$$\mathbb{P}\left(\widetilde{Q}_h^k(s,a) \geq (r_h + \mathbb{P}_h\widetilde{V}_{h+1}^k)(s,a), \forall s,a\right) \geq 1 - \delta.$$

Apply a union bound for $h,k$, we have for $M_\delta = \log(HK/\delta)/\log(64/63)$, with probability $1 - \delta$,

$$\mathbb{P}\left(\widetilde{Q}_h^k(s,a) \geq (r_h + \mathbb{P}_h\widetilde{V}_{h+1}^k)(s,a), \forall s,a,h,k\right) \geq 1 - \delta.$$

$\square$

**Lemma C.4.** *Suppose the event*

$$E = \left\{\left|\widehat{Q}_h^k(s,a) - (r_h^k + \mathbb{P}_h\widetilde{V}_{h+1}^k)(s,a)\right| \leq C_{\delta'}\|\phi(s,a)\|_{(\Omega_h^k)^{-1}}, \forall s,a,h,k\right\}$$

*holds. Choose $N_k \geq \max\{\log(\frac{32H^2(K+\lambda)dk}{\gamma\lambda} + 1)/[2\log(1/(1 - \frac{1}{2\kappa_h}))], \frac{\log 2}{2\log(1/(1-\frac{1}{2\kappa_h}))}\}$ and $\gamma = 16C_{\delta'}^2$. Then*

$$\frac{|(r_h + \mathbb{P}_h\widetilde{V}_{h+1})(s,a) - \phi(s,a)^{\mathrm{T}}(I - A_{h,k}^{N_k})\widehat{w}_h^k|}{\sqrt{\frac{\gamma}{2}\left(1 - (1 - \frac{1}{2\kappa_h})^{2N_k}\right)\phi(s,a)^{\mathrm{T}}(\Omega_h^k)^{-1}\phi(s,a)}} \leq 1, \quad \forall s,a \in \mathcal{S} \times \mathcal{A}, h \in [H], k \in [K].$$

*Proof of Lemma C.4.* By direct calculation,

$$\frac{|(r_h + \mathbb{P}_h\widetilde{V}_{h+1})(s,a) - \phi(s,a)^{\mathrm{T}}(I - A_{h,k}^{N_k})\widehat{w}_h^k|}{\sqrt{\frac{\gamma}{2}\left(1 - (1 - \frac{1}{2\kappa_h})^{2N_k}\right)\phi(s,a)^{\mathrm{T}}(\Omega_h^k)^{-1}\phi(s,a)}} \leq \frac{|\phi(s,a)^{\mathrm{T}}A_{h,k}^{N_k}\widehat{w}_h^k|}{\sqrt{\frac{\gamma}{2}\left(1 - (1 - \frac{1}{2\kappa_h})^{2N_k}\right)\phi(s,a)^{\mathrm{T}}(\Omega_h^k)^{-1}\phi(s,a)}}$$

$$+ \frac{|(r_h + \mathbb{P}_h\widetilde{V}_{h+1})(s,a) - \phi(s,a)^{\mathrm{T}}\widehat{w}_h^k|}{\sqrt{\frac{\gamma}{2}\left(1 - (1 - \frac{1}{2\kappa_h})^{2N_k}\right)\phi(s,a)^{\mathrm{T}}(\Omega_h^k)^{-1}\phi(s,a)}}$$

For the first term above, by CS inequality we have

$$|\phi(s,a)^{\mathrm{T}} A_{h,k}^{N_k} \widehat{w}_h^k| \leq \sqrt{\phi(s,a)^{\mathrm{T}}(\Omega_h^k)^{-1}\phi(s,a)} \cdot \left\|(\Omega_h^k)^{1/2} A_{h,k}^{N_k} \widehat{w}_h^k\right\|$$

$$\leq \sqrt{\phi(s,a)^{\mathrm{T}}(\Omega_h^k)^{-1}\phi(s,a)} \cdot \left\|(\Omega_h^k)^{1/2}\right\| \|A_{h,k}\|^{N_k} \cdot 2H\sqrt{\frac{dk}{\lambda}}$$

$$\leq \sqrt{\phi(s,a)^{\mathrm{T}}(\Omega_h^k)^{-1}\phi(s,a)} \cdot \sqrt{k+\lambda} \cdot (1-\frac{1}{2\kappa_h})^{N_k} \cdot 2H\sqrt{\frac{dk}{\lambda}}$$

and this indicates

$$\frac{|\phi(s,a)^{\mathrm{T}} A_{h,k}^{N_k} \widehat{w}_h^k|}{\sqrt{\frac{\gamma}{2}\left(1-(1-\frac{1}{2\kappa_h})^{2N_k}\right)\phi(s,a)^{\mathrm{T}}(\Omega_h^k)^{-1}\phi(s,a)}} \leq \frac{\sqrt{k+\lambda}\cdot(1-\frac{1}{2\kappa_h})^{N_k}\cdot 2H\sqrt{\frac{dk}{\lambda}}}{\sqrt{\frac{\gamma}{2}\left(1-(1-\frac{1}{2\kappa_h})^{2N_k}\right)}} \leq \frac{1}{2}$$

where the last inequality is by $N_k \geq \log(\frac{32H^2(K+\lambda)dk}{\gamma\lambda}+1)/[2\log(1/(1-\frac{1}{2\kappa_h}))]$.

For the second term above,

$$\frac{|(r_h+\mathbb{P}_h\widetilde{V}_{h+1})(s,a)-\phi(s,a)^{\mathrm{T}}\widehat{w}_h^k|}{\sqrt{\frac{\gamma}{2}\left(1-(1-\frac{1}{2\kappa_h})^{2N_k}\right)\phi(s,a)^{\mathrm{T}}(\Omega_h^k)^{-1}\phi(s,a)}} \leq \frac{C_{\delta'}}{\sqrt{\frac{\gamma}{2}\left(1-(1-\frac{1}{2\kappa_h})^{2N_k}\right)}} \leq \frac{C_{\delta'}}{\sqrt{\frac{\gamma}{2}(1-\frac{1}{2})}} = \frac{1}{2}.$$

Here the second inequality uses $N_k \geq \frac{\log 2}{2\log(1/(1-\frac{1}{2\kappa_h}))}$ and the last equal sign comes from $\gamma = 16C_{\delta'}^2$. $\qquad\square$

**Lemma C.5** (Optimism for Langevin Posterior Sampling). *For any $0 \leq \delta < 1$, we set the input in [Algorithm 2](#) as $N_k \geq \max\{\log(\frac{32H^2(K+\lambda)dk}{\gamma\lambda}+1)/[2\log(1/(1-\frac{1}{2\kappa_h}))], \frac{\log 2}{2\log(1/(1-\frac{1}{2\kappa_h}))}\}$, $\gamma = 16C_{\delta/4}^2$ and $M_\delta = \log(4HK/\delta)/\log(64/63)$, then with probability $1-\delta/2$, we have*

$$\widetilde{Q}_h^k(s,a) \geq Q_h^*(s,a), \ \widetilde{V}_h^k(s) \geq V_h^*(s) \quad \forall s,a \in \mathcal{S}\times\mathcal{A}, \forall h \in [H], k \in [K].$$

*Here $C_\delta$ is defined in [Lemma C.9](#).*

*Proof of [Lemma C.5](#).* **Step1:** Suppose the event

$$E = \left\{\left|\widehat{Q}_h^k(s,a)-(r_h^k+\mathbb{P}_h\widetilde{V}_{h+1}^k)(s,a)\right| \leq C_{\delta'}\|\phi(s,a)\|_{(\Omega_h^k)^{-1}}, \ \forall s,a,h,k\right\}$$

holds. Choose $N_k \geq \max\{\log(\frac{32H^2(K+\lambda)dk}{\gamma\lambda}+1)/[2\log(1/(1-\frac{1}{2\kappa_h}))], \frac{\log 2}{2\log(1/(1-\frac{1}{2\kappa_h}))}\}$, $\gamma = 16C_{\delta'}^2$ and $M_\delta = \log(4HK/\delta)/\log(64/63)$. Then we show, for any $h \in [H]$, with probability $1-\delta/4$, $\widetilde{Q}_h^k(s,a) \geq Q_h^*(s,a)$, $\widetilde{V}_h^k(s) \geq V_h^*(s)$ for all $(s,a) \in \mathcal{S}\times\mathcal{A}, h \in [H], k \in [K]$.

First, due to our choice of $M_\delta = \log(4HK/\delta)/\log(64/63)$, by [Lemma C.3](#), with probability $1-\delta/4$,

$$\widetilde{Q}_h^k(s,a) \geq (r_h+\mathbb{P}_h\widetilde{V}_{h+1}^k)(s,a), \ \forall(s,a) \in \mathcal{S}\times\mathcal{A}, h \in [H], k \in [K],$$

which we condition on.

Next, we finish the proof by backward induction. Base case: for $h = H+1$, the value functions are zero, and thus $\widetilde{Q}_{H+1}^k \geq Q_{H+1}^*$ holds trivially, which also implies $\widetilde{V}_{H+1}^k \geq V_{H+1}^*$. Suppose the conclusion holds true for $h+1$. Then for time step $h$ and any $k \in [K]$,

$$\widetilde{Q}_h^k - Q_h^* = \widetilde{Q}_h^k - (r_h+\mathbb{P}_h\widetilde{V}_{h+1}^k) + (r_h+\mathbb{P}_h\widetilde{V}_{h+1}^k) - Q_h^*$$

$$\geq \widetilde{Q}_h^k - (r_h+\mathbb{P}_h\widetilde{V}_{h+1}^k) + (r_h+\mathbb{P}_H V_{h+1}^*) - Q_h^*$$

$$= \widetilde{Q}_h^k - (r_h+\mathbb{P}_h\widetilde{V}_{h+1}^k) \geq 0$$

where the first inequality uses the induction hypothesis and the second inequality uses the condition. Lastly, $\widetilde{V}_h^k(\cdot) = \max_a \min\{\widetilde{Q}_h^k(\cdot, a), H - h + 1\} \leq \max_a \min\{Q_h^*(\cdot, a), H - h + 1\} = \max_a Q_h^*(\cdot, a) = V_h^*(\cdot)$. By induction, this finishes the Step1.

**Step2:** By Lemma C.9, with probability $1 - \delta/4$, for all $k \in [K], h \in [H], s \in \mathcal{S}, a \in \mathcal{A}$, it holds

$$\left|\widehat{Q}_h^k(s,a) - (r_h^k + \mathbb{P}_h \widetilde{V}_{h+1}^k)(s,a)\right| \leq C_{\delta/4} \|\phi(s,a)\|_{(\Omega_h^k)^{-1}}.$$

Therefore, in Step1, choose $\delta' = \delta/4$, and a union bound we obtain: for the choice $N_k \geq \max\{\log(\frac{32H^2(K+\lambda)dk}{\gamma\lambda} + 1)/[2\log(1/(1 - \frac{1}{2\kappa_h}))], \frac{\log 2}{2\log(1/(1-\frac{1}{2\kappa_h}))}\}$, $\gamma = 16C_{\delta/4}^2$ and $M_\delta = \log(4HK/\delta)/\log(64/63)$, then with probability $1 - \delta/2$, we have

$$\widetilde{Q}_h^k(s,a) \geq Q_h^*(s,a), \; \widetilde{V}_h^k(s) \geq V_h^*(s) \; \forall (s,a) \in \mathcal{S} \times \mathcal{A}, \; h \in [H], \; k \in [K].$$

$\square$

## C.3 Proofs of Concentration for Delayed-LPSVI

**Lemma C.6** (Pointwise Concentration for Langevin Posterior Sampling). *Choose* $N_k \geq \log(\frac{4HK^3}{\sqrt{\lambda/dK}})/\log(1/(1 - \frac{1}{2\kappa_h}))$. *Algorithm 2 guarantees that* $\forall k \in [K], h \in [H], s \in \mathcal{S}, a \in \mathcal{A}$, *the following holds with probability* $1 - \delta$,

$$\left|| \min\{\widetilde{Q}_h^k(s,a), H - h + 1\} - (r_h^k + \mathbb{P}_h \widetilde{V}_{h+1}^k)(s,a)\right| \leq \beta \|\phi(s,a)\|_{(\Omega_h^k)^{-1}} + \frac{1}{K^3}. \quad (21)$$

*where* $\beta := \sqrt{2\gamma \log(4C_d HMK/\delta)} + \sqrt{8H^2\left[\frac{d}{2}\log\left(\frac{k+\lambda}{\lambda}\right) + dM\log(1 + \frac{2\sqrt{8k^3}C_{H,d,k,M,\delta/2}}{H\sqrt{\lambda}}) + \log\frac{4}{\delta}\right]}$ $+2\sqrt{\lambda}\sqrt{d}H$. *In particular, here* $\log C_d = d\log(1 + (16\sqrt{2\gamma \log(2/\delta)/\lambda} + 16H\sqrt{d})K^3)$ *and* $C_{H,d,k,M,\delta} = 2H\sqrt{\frac{dk}{\lambda}} + \frac{\sqrt{2d\gamma} + \sqrt{2\gamma \log(M/\delta)}}{\sqrt{\lambda}}$.

*Proof of Lemma C.6.* Recall that $|r_h^k + \mathbb{P}_h \widetilde{V}_{h+1}^k| \leq H - h + 1$, therefore $r_h^k + \mathbb{P}_h \widetilde{V}_{h+1}^k = \min\{r_h^k + \mathbb{P}_h \widetilde{V}_{h+1}^k, H - h + 1\}$. This implies $|\min\{\widetilde{Q}_h^k(s,a), H - h + 1\} - r_h^k - [\mathbb{P}_h \widetilde{V}_{h+1}^k](s,a)| = |\min\{\widetilde{Q}_h^k(s,a), H - h + 1\} - \min\{r_h^k + [\mathbb{P}_h \widetilde{V}_{h+1}^k](s,a), H - h + 1\}| \leq |\widetilde{Q}_h^k(s,a) - r_h^k - [\mathbb{P}_h \widetilde{V}_{h+1}^k](s,a)|$. Hence

$$\left|\min\{\widetilde{Q}_h^k(s,a), H - h + 1\} - r_h^k - [\mathbb{P}_h \widetilde{V}_{h+1}^k](s,a)\right| \leq \left|\widetilde{Q}_h^k(s,a) - r_h^k - [\mathbb{P}_h \widetilde{V}_{h+1}^k](s,a)\right|$$

$$= \left|\widetilde{Q}_h^k(s,a) - \widehat{Q}_h^k(s,a) + \widehat{Q}_h^k(s,a) - r_h^k - [\mathbb{P}_h \widetilde{V}_{h+1}^k](s,a)\right|$$

$$\leq \underbrace{\left|\widetilde{Q}_h^k(s,a) - \widehat{Q}_h^k(s,a)\right|}_{R_1} + \underbrace{\left|\widehat{Q}_h^k(s,a) - r_h^k - [\mathbb{P}_h \widetilde{V}_{h+1}^k](s,a)\right|}_{R_2}.$$

The proof then directly follows Lemma C.7 and Lemma C.9 to bound $R_1$ and $R_2$ respectively (together with a union bound). $\square$

**Lemma C.7** (Concentration of $R_1$ with Langevin Posterior Sampling). *Suppose* $N_k \geq \log(\frac{4HK^3}{\sqrt{\lambda/dK}})/\log(1/(1 - \frac{1}{2\kappa_h}))$. *For any* $0 < \delta < 1$, *define the event* $\widetilde{E}$ *as*

$$\widetilde{E} = \left\{\left|\widetilde{Q}_h^k(s,a) - \phi(s,a)^{\mathrm{T}}\widehat{w}_h^k\right| \leq \sqrt{2\gamma \log(2C_d HMK/\delta)} \|\phi(s,a)\|_{(\Omega_h^k)^{-1}} + \frac{1}{K^3},\right.$$

$$\left.\forall k \in [K], h \in [H], s \in \mathcal{S}, a \in \mathcal{A}\right\}, \quad (22)$$

*then* $\widetilde{E}$ *happens w.p.* $1 - \delta$. *Here* $\log C_d = d\log(1 + (16\sqrt{2\gamma \log(2/\delta)/\lambda} + 16H\sqrt{d})K^3)$.

*Proof of Lemma C.7.* In the Step1 and Step2, we abuse $\widetilde{w}_h^k$ to denote $\widetilde{w}_h^{k,m}$ for arbitrary $m$ to avoid notation redundancy.

In **Step1**: We first show for any $k \in [K], h \in [H], (s,a) \in \mathcal{S} \times \mathcal{A}$, with probability $1 - \delta$,

$$\left| \phi(s,a)^{\mathrm{T}}(\widetilde{w}_h^k - \widehat{w}_h^k) \right| \leq \sqrt{2\gamma \log(2/\delta)} \, \|\phi(s,a)\|_{(\Omega_h^k)^{-1}} + \frac{1}{2K^3}.$$

Indeed, by Lemma C.2 we have $(\widetilde{w}_h^k - (I - A_{h,k}^{N_k})\widehat{w}_h^k) \sim \mathcal{N}(0, \Theta_h^k)$, which gives,

$$\phi(s,a)^{\mathrm{T}}(\widetilde{w}_h^k - (I - A_{h,k}^{N_k})\widehat{w}_h^k) \sim \mathcal{N}(0, \phi(s,a)^{\mathrm{T}}\Theta_h^k \phi(s,a)).$$

Therefore, $\phi(s,a)^{\mathrm{T}}(\widetilde{w}_h^k - (I - A_{h,k}^{N_k})\widehat{w}_h^k)$ is $\phi(s,a)^{\mathrm{T}}\Theta_h^k \phi(s,a)$-sub-Gaussian. By concentration of sub-Gaussian random variables, we have

$$\mathbb{P}\left( \left| \phi(s,a)^{\mathrm{T}}(\widetilde{w}_h^k - (I - A_{h,k}^{N_k})\widehat{w}_h^k) \right| \geq t \right) \leq 2\exp\left( -\frac{t^2}{2\phi(s,a)^{\mathrm{T}}\Theta_h^k \phi(s,a)} \right) := \delta$$

Solving for $\delta$ gives with probability $1 - \delta$,

$$\left| \phi(s,a)^{\mathrm{T}}(\widetilde{w}_h^k - (I - A_{h,k}^{N_k})\widehat{w}_h^k) \right| \leq \sqrt{2\log(2/\delta)} \, \|\phi(s,a)\|_{\Theta_h^k} \leq \sqrt{2\gamma \log(2/\delta)} \, \|\phi(s,a)\|_{(\Omega_h^k)^{-1}},$$

where the last inequality is by Lemma C.2, and by Lemma C.8, the above further implies

$$\left| \phi(s,a)^{\mathrm{T}}(\widetilde{w}_h^k - \widehat{w}_h^k) \right| \leq \sqrt{2\gamma \log(2/\delta)} \, \|\phi(s,a)\|_{(\Omega_h^k)^{-1}} + \frac{1}{2K^3}.$$

**Step2:** we prove that for any $0 < \delta < 1$, define the event $\widetilde{E}$ as

$$\widetilde{E} = \left\{ \left| \phi(s,a)^{\mathrm{T}}\widetilde{w}_h^k - \phi(s,a)^{\mathrm{T}}\widehat{w}_h^k \right| \leq \sqrt{2\gamma \log(2C_d HK/\delta)} \, \|\phi(s,a)\|_{(\Omega_h^k)^{-1}} + \frac{1}{K^3}, \right.$$
$$\left. \forall k \in [K], h \in [H], s \in \mathcal{S}, a \in \mathcal{A} \right\}, \tag{23}$$

then $\widetilde{E}$ happens w.p. $1 - \delta$. Here $\log C_d = d\log(1 + (16\sqrt{2\gamma \log(2/\delta)/\lambda} + 16H\sqrt{d})K^3)$.

In Lemma D.12, set $\theta = \widetilde{w}_h^k - \widehat{w}_h^k$ and $A = (\Omega_h^k)^{-1}$ and $B = 1/\lambda$, and let $\mathcal{V}$ be the $\frac{1}{4K^3}$-epsilon net for the class of values $\{|\langle \phi, \widetilde{w}_h^k - \widehat{w}_h^k\rangle| - C\sqrt{\phi^{\top}(\Omega_h^k)^{-1}\phi} - \frac{1}{2K^3} : \|\phi\| \leq 1\}$ (where $C = \sqrt{2\gamma \log(2/\delta)}$), then it must also be the $\frac{1}{4K^3}$-epsilon net for the class of values $\mathcal{F} = \{|\langle \phi(s,a), \widetilde{w}_h^k - \widehat{w}_h^k\rangle| - C\sqrt{\phi(s,a)^{\top}(\Omega_h^k)^{-1}\phi(s,a)} - \frac{1}{2K^3} : (s,a) \in \mathcal{S} \times \mathcal{A}\}$, let $\bar{\mathcal{V}}$ is the smallest subset of $\mathcal{V}$ such that it is $\frac{1}{2K^3}$-epsilon net for the class of values $\mathcal{F}$. Then we can select $\mathcal{V}_{\mathcal{S} \times \mathcal{A}}$ to be the set of state-action pairs such that for any $f_\phi := |\langle \phi, \widetilde{w}_h^k - \widehat{w}_h^k\rangle| - C\sqrt{\phi^{\top}(\Omega_h^k)^{-1}\phi} \in \bar{\mathcal{V}} - \frac{1}{2K^3}$, there exists $(s,a) \in \mathcal{V}_{\mathcal{S} \times \mathcal{A}}$ satisfies $\left| |\langle \phi(s,a), \widetilde{w}_h^k - \widehat{w}_h^k\rangle| - C\sqrt{\phi(s,a)^{\top}(\Omega_h^k)^{-1}\phi(s,a)|} - \frac{1}{2K^3} - f_\phi \right| \leq 1/4K^3$, then we have $\mathcal{V}_{\mathcal{S} \times \mathcal{A}}$ is a $1/(2K^3)$-epsilon net of $\mathcal{F}$ and $|\mathcal{V}_{\mathcal{S} \times \mathcal{A}}| \leq |\bar{\mathcal{V}}| \leq |\mathcal{V}|$. Therefore,

$$\sup_{s,a} \left( |\langle \phi(s,a), \widetilde{w}_h^k - \widehat{w}_h^k\rangle| - C\sqrt{\phi(s,a)^{\top}(\Omega_h^k)^{-1}\phi(s,a)} - \frac{1}{2K^3} \right)$$
$$\leq \sup_{(s,a) \in \mathcal{V}_{\mathcal{S} \times \mathcal{A}}} \left( |\langle \phi(s,a), \widetilde{w}_h^k - \widehat{w}_h^k\rangle| - C\sqrt{\phi(s,a)^{\top}(\Omega_h^k)^{-1}\phi(s,a)} - \frac{1}{2K^3} \right) + 1/(2K^3)$$
$$\leq 1/(2K^3),$$

where the last inequality is from Step1. Then by a union bound over $H$, $K$ and $(1 + (16\sqrt{2\gamma \log(2/\delta)/\lambda} + 16H\sqrt{d})K^3)^d$, we have with probability $1 - \delta$,

$$\sup_{s,a,h,k} \left( |\langle \phi(s,a), \widetilde{w}_h^k - \widehat{w}_h^k\rangle| - \sqrt{2\gamma \log(2C_d HK/\delta)}\sqrt{\phi(s,a)^{\top}(\Omega_h^k)^{-1}\phi(s,a)} \right)$$
$$\leq \frac{1}{2K^3} + \frac{1}{2K^3} = \frac{1}{K^3},$$

where $\log C_d = d\log(1 + (16\sqrt{2\gamma\log(2/\delta)/\lambda} + 16H\sqrt{d})K^3)$.

**Step3:** We finish the proof. Note $\widetilde{Q}_h^k = \max_m \phi^{\mathrm{T}}\widetilde{w}_h^{k,m}$, hence by a union bound over $M$, we have

$$\left|\widetilde{Q}_h^k(s,a) - \phi(s,a)^{\mathrm{T}}\widehat{w}_h^k\right| = |\max_m \phi(s,a)^{\mathrm{T}}\widetilde{w}_h^{k,m} - \phi(s,a)^{\mathrm{T}}\widehat{w}_h^k|$$

$$\leq \max_m |\phi(s,a)^{\mathrm{T}}\widetilde{w}_h^{k,m} - \phi(s,a)^{\mathrm{T}}\widehat{w}_h^k|$$

$$\leq \sqrt{2\gamma\log(2C_d HMK/\delta)}\,\|\phi(s,a)\|_{(\Omega_h^k)^{-1}} + \frac{1}{K^3}$$

for all $k, h, s, a$ with probability $1 - \delta$. Here the last inequality uses Step2. This finishes the proof. $\square$

**Lemma C.8.** *Let* $N_k \geq \log(\frac{4HK^3}{\sqrt{\lambda/dK}})/\log(1/(1 - \frac{1}{2\kappa_h}))$ *and* $\eta = \frac{1}{4\lambda_{\max}(\Omega_h^k)}$, *then*

$$\left\|\phi(s,a)^{\mathrm{T}}A_{h,k}^{N_k}\widehat{w}_h^k\right\| \leq \frac{1}{2K^3}$$

*Proof of Lemma C.8.* By direct calculation,

$$\left\|\phi(s,a)^{\mathrm{T}}A_{h,k}^{N_k}\widehat{w}_h^k\right\| \leq \|\phi(s,a)\|\,\|A_{h,k}\|^{N_k}\,\|\widehat{w}_h^k\| \leq \|A_{h,k}\|^{N_k}\,\|\widehat{w}_h^k\|$$

$$\leq \|A_{h,k}\|^{N_k} 2H\sqrt{\frac{dk}{\lambda}} \leq \left(1 - \frac{1}{2\kappa_h}\right)^{N_k} \cdot 2H\sqrt{\frac{dk}{\lambda}} \leq \frac{1}{2K^3}$$

where the third inequality is by Lemma D.4 and the fourth inequality is by (20). The last inequality is by the choice of $N_k$. $\square$

**Lemma C.9** (Concentration of $R_2$ with Langevin Posterior Sampling). *For any* $0 < \delta < 1$, *with probability* $1 - \delta$, *for all* $k \in [K], h \in [H], s \in \mathcal{S}, a \in \mathcal{A}$, *it holds*

$$\left|\widehat{Q}_h^k(s,a) - (r_h^k + \mathbb{P}_h\widetilde{V}_{h+1}^k)(s,a)\right| \leq C_\delta\,\|\phi(s,a)\|_{(\Omega_h^k)^{-1}}$$

*where* $C_\delta = \sqrt{8H^2\left[\frac{d}{2}\log\left(\frac{k+\lambda}{\lambda}\right) + dM\log(1 + \frac{2\sqrt{8k^3}C_{H,d,k,M,\delta}}{H\sqrt{\lambda}}) + \log\frac{2}{\delta}\right]} + 2\sqrt{\lambda}\sqrt{d}H$ *and the quantity* $C_{H,d,k,M,\delta} = 2H\sqrt{\frac{dk}{\lambda}} + \frac{\sqrt{2d\gamma} + \sqrt{2\gamma\log(M/\delta)}}{\sqrt{\lambda}}$.

*Proof of Lemma C.9.* For any $(k,h) \in [K] \times [H]$ and $(s,a) \in \mathcal{S} \times \mathcal{A}$, denote

$$\phi(s,a)^{\mathrm{T}}w_h^k := (r_h^k + \mathbb{P}_h\widetilde{V}_{h+1}^k)(s,a), \text{ where } w_h^k := \theta_h + \int_{\mathcal{S}}\widetilde{V}_{h+1}^k(s')\mathrm{d}\mu_h(s').$$

Recall $y_h^\tau = \mathbb{1}_{\tau,k-1} \cdot [r_h^\tau(s_h^\tau, a_h^\tau) + \widetilde{V}_{h+1}^k(s_{h+1}^\tau)]$ from Algorithm 1 and denote $\bar{y}_h^\tau := r_h^\tau(s_h^\tau, a_h^\tau) + \widetilde{V}_{h+1}^k(s_{h+1}^\tau)$. Then by definition,

$$\widehat{w}_h^k = (\Omega_h^k)^{-1}\sum_{\tau=1}^{k-1}\mathbb{1}_{\tau,k-1}\cdot\phi(s_h^\tau, a_h^\tau)y_h^\tau = (\Omega_h^k)^{-1}\sum_{\tau=1}^{k-1}\mathbb{1}_{\tau,k-1}\cdot\phi(s_h^\tau, a_h^\tau)\bar{y}_h^\tau.$$

By definition of $\Omega_h^k$, we have $\Phi_h\Phi_h^{\mathrm{T}} = \Omega_h^k - \lambda I$. Plug it into the definition of $\widehat{w}_h^k$, we have

$$\widehat{w}_h^k = (\Omega_h^k)^{-1}\sum_{\tau=1}^{k-1}\mathbb{1}_{\tau,k-1}\cdot\phi(s_h^\tau, a_h^\tau)\left(\bar{y}_h^\tau - \phi(s_h^\tau, a_h^\tau)^{\mathrm{T}}w_h^k + \phi(s_h^\tau, a_h^\tau)^{\mathrm{T}}w_h^k\right)$$

$$= (\Omega_h^k)^{-1}\sum_{\tau=1}^{k-1}\mathbb{1}_{\tau,k-1}\cdot\phi(s_h^\tau, a_h^\tau)\left(\bar{y}_h^\tau - \phi(s_h^\tau, a_h^\tau)^{\mathrm{T}}w_h^k\right) + (\Omega_h^k)^{-1}\left(\Omega_h^k - \lambda I\right)w_h^k.$$

We then proceed to bound $\widehat{w}_h^k - w_h^k$, which gives

$$\widehat{w}_h^k - w_h^k = (\Omega_h^k)^{-1} \sum_{\tau=1}^{k-1} \mathbb{1}_{\tau,k-1} \cdot \phi(s_h^\tau, a_h^\tau) \left( \bar{y}_h^\tau - \phi(s_h^\tau, a_h^\tau)^\mathrm{T} w_h^k \right) - \lambda(\Omega_h^k)^{-1} w_h^k$$

$$= \underbrace{(\Omega_h^k)^{-1} \sum_{\tau=1}^{k-1} \mathbb{1}_{\tau,k-1} \cdot \phi(s_h^\tau, a_h^\tau) \left( \widetilde{V}_{h+1}^k(s_{h+1}^\tau) - \mathbb{P}_h \widetilde{V}_{h+1}^k(s_h^\tau, a_h^\tau) \right)}_{\text{(i)}} - \underbrace{\lambda(\Omega_h^k)^{-1} w_h^k}_{\text{(ii)}}.$$

**Term (i).** Since $\Omega_h^k$ is positive definite, multiplying the first term $(i)$ with $\phi(s,a)$ and by Cauchy-Schwartz inequality, we obtain,

$$\left| \phi(s,a)^\mathrm{T}(\mathrm{i}) \right| \le \|\phi(s,a)\|_{(\Omega_h^k)^{-1}} \left\| \sum_{\tau=1}^{k-1} \mathbb{1}_{\tau,k-1} \cdot \phi(s_h^\tau, a_h^\tau) \left( \widetilde{V}_{h+1}^k(s_{h+1}^\tau) - \mathbb{P}_h \widetilde{V}_{h+1}^k(s_h^\tau, a_h^\tau) \right) \right\|_{(\Omega_h^k)^{-1}}.$$

Apply Lemma C.10, we have with probability at least $1 - \delta$, for any $(k,h) \in [K] \times [H]$, and $(s,a) \in \mathcal{S} \times \mathcal{A}$,

$$\left| \phi(s,a)^\mathrm{T}(\mathrm{i}) \right| \le C_1 \|\phi(s,a)\|_{(\Omega_h^k)^{-1}}, \tag{24}$$

where $C_1 = \sqrt{8H^2 \left[ \frac{d}{2} \log\left(\frac{k+\lambda}{\lambda}\right) + dM \log(1 + \frac{2\sqrt{8k^3}C_{H,d,k,M,\delta}}{H\sqrt{\lambda}}) + \log\frac{2}{\delta} \right]}$.

**Term (ii).** By Lemma B.12, $\forall (s,a) \in \mathcal{S} \times \mathcal{A}$, and $(k,h) \in [K] \times [H]$, $\left| \phi(s,a)^\mathrm{T}(\mathrm{ii}) \right|$ can be bounded as

$$\left| \phi(s,a)^\mathrm{T}(\mathrm{ii}) \right| = \lambda \left| \phi(s,a)^\mathrm{T}(\Omega_h^k)^{-1} w_h^k \right| \le 2\sqrt{\lambda}\sqrt{d}H \|\phi(s,a)\|_{(\Omega_h^k)^{-1}}. \tag{25}$$

Combining (24), (25), we have with probability $1 - \delta$, for any $(k,h) \in [K] \times [H]$ and $(s,a) \in \mathcal{S} \times \mathcal{A}$,

$$\left| \widehat{Q}_h^k(s,a) - (r_h^k + \mathbb{P}_h \widetilde{V}_{h+1}^k)(s,a) \right| = \left| \phi(s,a)^\mathrm{T}(\widehat{w}_h^k - w_h^k) \right| \le \left| \phi(s,a)^\mathrm{T}(\mathrm{i}) \right| + \left| \phi(s,a)^\mathrm{T}(\mathrm{ii}) \right|$$

$$\le (C_1 + 2\sqrt{\lambda}\sqrt{d}H) \|\phi(s,a)\|_{(\Omega_h^k)^{-1}},$$

This concludes the proof. $\qquad \square$

**Lemma C.10.** *For any $0 < \delta < 1$, with probability $1 - \delta$, we have $\forall (k,h) \in [K] \times [H]$,*

$$\left\| \sum_{\tau=1}^{k-1} \mathbb{1}_{\tau,k-1} \cdot \phi(s_h^\tau, a_h^\tau) \left( \widetilde{V}_{h+1}^k(s_{h+1}^\tau) - \mathbb{P}_h \widetilde{V}_{h+1}^k(s_h^\tau, a_h^\tau) \right) \right\|_{(\Omega_h^k)^{-1}}^2$$

$$\le 8H^2 \left[ \frac{d}{2} \log\left(\frac{k+\lambda}{\lambda}\right) + dM \log(1 + \frac{2\sqrt{8k^3}C_{H,d,k,M,\delta}}{H\sqrt{\lambda}}) + \log\frac{2}{\delta} \right],$$

*here $C_{H,d,k,M,\delta} = 2H\sqrt{\frac{dk}{\lambda}} + \frac{\sqrt{2d\gamma} + \sqrt{2\gamma \log(M/\delta)}}{\sqrt{\lambda}}.$* [9]

*Proof of Lemma C.10.* First note that

$$\widetilde{V}_h^k(\cdot) := \max_a \min\{\widehat{Q}_h^k(\cdot,a), (H-h+1)\} = \max_a \min \max_m \{\widetilde{Q}_h^{k,m}, (H-h+1)\}$$

$$= \max_a \min\{\max_m \phi(\cdot,a)^\mathrm{T} \widetilde{w}_h^{k,m}, (H-h+1)\}.$$

Choosing $w_0 = 0$, then by Lemma C.2 and $(\Theta_h^k)^{-1/2}(\widetilde{w}_h^{k,m} - (I - A_{h,k}^{N_k})\widehat{w}_h^k) \sim \mathcal{N}(0, I_d)$, and by Lemma D.7, with probability $1 - \delta/2$, we have

$$\frac{\sqrt{\lambda}}{\sqrt{\gamma}} \left\| \widetilde{w}_h^{k,m} - (I - A_{h,k}^{N_k})\widehat{w}_h^k \right\| \le \left\| (\Theta_h^k)^{-1/2}(\widetilde{w}_h^{k,m} - (I - A_{h,k}^{N_k})\widehat{w}_h^k) \right\| \le \sqrt{2d} + \sqrt{2\log(1/\delta)},$$

---

[9] We will choose $\gamma$ to be $\mathrm{Poly}(H,d,K)$ and this will not affect the overall dependence of the guarantee since $C_{H,d,k,M,\delta}$ is inside the log term.

where the first inequality uses Lemma C.2 again. Apply the union bound over all $m$, then above implies with probability $1 - \delta/2$, $\forall m \in [M]$

$$\left\|\widetilde{w}_h^{k,m}\right\| \leq \left\|\widehat{w}_h^k\right\| + \frac{\sqrt{2d\gamma} + \sqrt{2\gamma \log(M/\delta)}}{\sqrt{\lambda}} \leq 2H\sqrt{\frac{dk}{\lambda}} + \frac{\sqrt{2d\gamma} + \sqrt{2\gamma \log(M/\delta)}}{\sqrt{\lambda}} := C_{H,d,k,M,\delta}.$$

(26)

(where we used $\left\|(I - A_{h,k}^{N_k})\widehat{w}_h^k\right\| \leq \left\|(I - A_{h,k}^{N_k})\right\| \left\|\widehat{w}_h^k\right\| \leq \left\|\widehat{w}_h^k\right\|$). Now consider the function class $\bar{\mathcal{V}} := \{\max_a \max_m \phi(\cdot,a)^{\mathrm{T}} w^m : \|w^m\| \leq C_{H,d,k,M,\delta}\}$, so by Lemma D.13 the $\epsilon$-log covering number for $\bar{\mathcal{V}}$ is $dM \log(1 + \frac{2C_{H,d,k,M,\delta}}{\epsilon})$. Since $\min\{\cdot,\cdot\}$ is a non-expansive operator, the $\epsilon$-log covering number for the function class $\mathcal{V} := \{\max_a \min\{\max_m \phi(\cdot,a)^{\mathrm{T}} w^m, (H-h+1)\} : \|w^m\| \leq C_{H,d,k,M,\delta}\}$, is at most $dM \log(1 + \frac{2C_{H,d,k,M,\delta}}{\epsilon})$. Hence, for any $V \in \mathcal{V}$, there exists $V'$ in the $\epsilon$-covering such that $V = V' + \Delta_V$ with $\|\Delta_V\|_\infty \leq \epsilon$. Then with probability $1 - \delta/2$,

$$\left\|\sum_{\tau=1}^{k-1} \mathbb{1}_{\tau,k-1} \phi(s_h^\tau, a_h^\tau)\left(V(s_{h+1}^\tau) - \mathbb{P}_h V(s_h^\tau, a_h^\tau)\right)\right\|_{(\Omega_h^k)^{-1}}^2$$

$$\leq 2\left\|\sum_{\tau=1}^{k-1} \mathbb{1}_{\tau,k-1} \phi(s_h^\tau, a_h^\tau)\left(V'(s_{h+1}^\tau) - \mathbb{P}_h V'(s_h^\tau, a_h^\tau)\right)\right\|_{(\Omega_h^k)^{-1}}^2$$

$$+ 2\left\|\sum_{\tau=1}^{k-1} \mathbb{1}_{\tau,k-1} \phi(s_h^\tau, a_h^\tau)\left(\Delta_V(s_{h+1}^\tau) - \mathbb{P}_h \Delta_V(s_h^\tau, a_h^\tau)\right)\right\|_{(\Omega_h^k)^{-1}}^2$$

(27)

$$\leq 2\left\|\sum_{\tau=1}^{k-1} \mathbb{1}_{\tau,k-1} \phi(s_h^\tau, a_h^\tau)\left(V'(s_{h+1}^\tau) - \mathbb{P}_h V'(s_h^\tau, a_h^\tau)\right)\right\|_{(\Omega_h^k)^{-1}}^2 + \frac{8k^2\epsilon^2}{\lambda}$$

$$\leq 4H^2\left[\frac{d}{2}\log\left(\frac{k+\lambda}{\lambda}\right) + dM \log(1 + \frac{2C_{H,d,k,M,\delta}}{\epsilon}) + \log\frac{2}{\delta}\right] + \frac{8k^2\epsilon^2}{\lambda}$$

where the second inequality can be conducted using a direct calculation and the third inequality uses Lemma D.9 and a union bound over the covering number. Now by (26) and (27) and a union bound, we have for any $\epsilon > 0$, with probability $1 - \delta$,

$$\left\|\sum_{\tau=1}^{k-1} \phi(s_h^\tau, a_h^\tau)\left(\widetilde{V}_{h+1}^k(s_{h+1}^\tau) - \mathbb{P}_h \widetilde{V}_{h+1}^k(s_h^\tau, a_h^\tau)\right)\right\|_{(\Omega_h^k)^{-1}}^2$$

$$\leq 4H^2\left[\frac{d}{2}\log\left(\frac{k+\lambda}{\lambda}\right) + dM \log(1 + \frac{2C_{H,d,k,M,\delta}}{\epsilon}) + \log\frac{2}{\delta}\right] + \frac{8k^2\epsilon^2}{\lambda}$$

$$\leq 8H^2\left[\frac{d}{2}\log\left(\frac{k+\lambda}{\lambda}\right) + dM \log(1 + \frac{2\sqrt{8k^3}C_{H,d,k,M,\delta}}{H\sqrt{\lambda}}) + \log\frac{2}{\delta}\right],$$

where the last step choose $\epsilon^2 = H^2\lambda/8k^2$ so $\frac{8k^2\epsilon^2}{\lambda} \leq 4H^2$. Lastly, apply the union bound over $H, K$ to obtain the stated result. $\square$

# D    Auxiliary lemmas

## D.1    Useful Norm Inequalities

**Lemma D.1.** *Suppose* $v \in \mathbb{R}^d$, *and* $A$ *is some positive definite matrix whose eigenvalues satisfy* $\lambda_{\max}(A) \geq \cdots \geq \lambda_{\min}(A) > 0$. *It can be shown that*

$$\sqrt{\lambda_{\min}(A)}\,\|v\| \leq \|v\|_A \leq \sqrt{\lambda_{\max}(A)}\,\|v\|.$$

*Proof of Lemma D.1.* Consider the eigenvalue decomposition of $A$, which gives $A = U\Lambda U^{\mathrm{T}}$, where $\Lambda = \mathrm{diag}(\lambda_{\max}(A), \ldots, \lambda_{\min}(A))$. Then

$$\|v\|_A = \sqrt{\sum_{i=1}^d \lambda_i(A)(u_i^{\mathrm{T}} v)^2} \leq \sqrt{\lambda_{\max}(A)\left\|u_i^{\mathrm{T}} v\right\|^2} = \sqrt{\lambda_{\max}(A)}\,\|v\|.$$

Similar argument shows $\|v\|_A \geq \sqrt{\lambda_{\min}(A)}$. $\qquad\square$

**Lemma D.2** (Lemma D.1 of [35]). *Let $\Omega_h^k$ be the precision matrix of the posterior distribution of $w_h^k$ at step $h$ of episode $k$, where $\Omega_h^k := \sigma^{-2}\Phi_h\Phi_h^{\mathrm{T}} + \Sigma^{-1}$ with $\Sigma^{-1} = \lambda I_d$ and $\sigma^2 = 1$. Then*

$$\sum_{\tau=1}^{k-1} \|\phi(s_h^\tau, a_h^\tau)\|_{(\Omega_h^k)^{-1}}^2 \leq d.$$

**Lemma D.3** (Bound on Weights of Q-function). *Suppose the linear MDP assumption and at each step $h \in [H]$, rewards $r_h$ are bounded between $[0,1]$, then the norm of the true parameter $w_h^\pi$ under fixed policy $\pi$ satisfies*

$$\forall h \in [H], \quad \|w_h^\pi\| \leq 2H\sqrt{d}.$$

*In addition, for any $(s,a) \in \mathcal{S} \times \mathcal{A}$, let $\phi(s,a)^{\mathrm{T}}w_h^k := (r_h + \mathbb{P}_h \widetilde{V}_{h+1}^k)(s,a)$, we also have*

$$\forall h \in [H], k \in [K], \quad \|w_h^k\| \leq 2H\sqrt{d}.$$

*Proof of Lemma D.3.* By definition in Lemma A.1, the true parameter $w_h$ at time step $h$ is

$$w_h^\pi := \theta_h + \mathbb{E}_{s' \sim \mu_h}[V_{h+1}^\pi(s')].$$

With bounded rewards $r_h \in [0,1]$, we have $V_{h+1}^\pi(s) \leq H$, $\forall s \in \mathcal{S}$. Since $\|\theta_h\| \leq \sqrt{d}$, and $\left\|\mathbb{E}_{\mu_h}[V_{h+1}^\pi(s')]\right\| \leq \left\|\int_{\mathcal{S}} H \mathrm{d}\mu_h(s')\right\| \leq H\sqrt{d}$.

Similarly, by definition of the constructed weights $w_h^k$,

$$w_h^k := \theta_h + \int_{\mathcal{S}} \widetilde{V}_{h+1}^k(s') \mathrm{d}\mu_h(s').$$

From Line 15 of Algorithm 1, for any $h \in [H]$ and $s \in \mathcal{S}$, $\widetilde{V}_h^k(s) = \max_a \min\{\widetilde{Q}_h^k(\cdot, a), H - h + 1\} \leq H$. Applying triangle inequality, we have

$$\begin{aligned}
\left\|w_h^k\right\| &\leq \|\theta_h\| + \left\|\int_{\mathcal{S}} \widetilde{V}_{h+1}^k(s') \mathrm{d}\mu_h(s')\right\| \\
&\leq \sqrt{d} + \left\|\int_{\mathcal{S}} H \mathrm{d}\mu_h(s')\right\| \\
&\leq 2H\sqrt{d}.
\end{aligned}$$

$\qquad\square$

**Lemma D.4** (Bound on Estimated Weights of Algorithm 1). *For any step $h \in [H]$ and episode $k \in [K]$, the weight $\widehat{w}_h^k$ output by Algorithm 1 satisfies,*

$$\left\|\widehat{w}_h^k\right\| \leq 2H\sqrt{\frac{dk}{\lambda}}.$$

*Proof of Lemma D.4.* For any vector $\mathbf{v} \in \mathbb{R}^d$, it holds

$$\begin{aligned}
\left|\mathbf{v}^\top \widehat{w}_h^k\right| &= \left|\mathbf{v}^\top \left(\Omega_h^k\right)^{-1} \sum_{\tau=1}^{k-1} \phi_h^\tau \left[r\left(s_h^\tau, a_h^\tau\right) + \widetilde{V}_h^k(s_{h+1}^\tau)\right]\right| \\
&\leq \sum_{\tau=1}^{k-1} \left|\mathbf{v}^\top \left(\Omega_h^k\right)^{-1} \phi_h^\tau\right| \cdot 2H \leq \sqrt{\left[\sum_{\tau=1}^{k-1} \mathbf{v}^\top \left(\Omega_h^k\right)^{-1} \mathbf{v}\right] \cdot \left[\sum_{\tau=1}^{k-1} \left(\phi_h^\tau\right)^\top \left(\Omega_h^k\right)^{-1} \phi_h^\tau\right]} \cdot 2H \\
&\leq 2H\|\mathbf{v}\|\sqrt{dk/\lambda},
\end{aligned}$$

where the last step is by Lemma D.2. The above directly imply the stated result by the definition of $l_2$ norm. $\qquad\square$

## D.2 Concentration Inequalities

**Lemma D.5** ([3]). *Suppose $Z$ is a random variable following a Gaussian distribution $\mathcal{N}(\mu, \sigma^2)$, where $\sigma > 0$. The following concentration and anti-concentration inequalities hold for any $z \geq 1$:*

$$\frac{1}{2\sqrt{\pi}z}e^{-z^2/2} \leq \mathbb{P}\left(|Z - \mu| > z\sigma\right) \leq \frac{1}{\sqrt{\pi}z}e^{-z^2/2}.$$

*And for $0 \leq z \leq 1$, we have,*

$$\mathbb{P}\left(|Z - \mu| > z\sigma\right) \geq \frac{1}{\sqrt{8\pi}}e^{-z^2/2}.$$

**Lemma D.6** (Sub-exponential tail bound). *Suppose $\{\tau_k\}_{k=1}^{\infty}$ are $(v, b)$-sub-exponential random variables. denote $D_{\tau,K,\delta} := \min\left\{\sqrt{2v^2 \log\left(\frac{3K}{2\delta}\right)}, 2b\log\left(\frac{3K}{2\delta}\right)\right\}$. Then with probability $1 - \delta$,*

$$\max_{k \in [K]} \tau_k \leq \mathbb{E}[\tau] + D_{\tau,K,\delta}.$$

**Lemma D.7** (Multivariate Gaussian Concentration). *Suppose $X \sim \mathcal{N}(0, I_d)$. Then with probability $1 - \delta$,*

$$\|X\| \leq \sqrt{2d} + \sqrt{2\log(1/\delta)}.$$

*Proof.* Apply Proposition 1 of [30], choose $A = I_d$, then $\Sigma = I_d$ and $Tr(\Sigma) = d$, $\|\Sigma\| = 1$. Then

$$P\left[\|X\|^2 \geq d + 2\sqrt{dt} + 2t\right] \leq e^{-t} \Rightarrow P[\|X\|^2 \geq 2(\sqrt{d} + \sqrt{t})^2] \leq e^{-t} := \delta$$

which implies with probability $1 - \delta$, $\|X\| \leq \sqrt{2d} + \sqrt{2\log(1/\delta)}$. $\qquad\qquad\square$

**Lemma D.8** (Elliptical Potential Lemma [1]). *Suppose $\{\phi_t\}_{t=1}^{\infty}$ is an $\mathbb{R}^d$-valued sequence, $\Omega_0 \in \mathbb{R}^{d \times d}$ is positive definite, and $\Omega_t = \Omega_0 + \sum_{\tau=1}^{t-1} \phi_\tau \phi_\tau^{\mathrm{T}}$. If $\lambda_{\min}(\Omega_0) \geq 1$, and $\|\phi_\tau\|_2 \leq 1$ for all $\tau \in \mathbb{Z}_+$, then for any $t \in \mathbb{Z}_+$,*

$$\log\left(\frac{\det(\Omega_{t+1})}{\det(\Omega_1)}\right) \leq \sum_{\tau=1}^{t} \phi_\tau^{\mathrm{T}}(\Omega_\tau)^{-1}\phi_\tau \leq 2\log\left(\frac{\det(\Omega_{t+1})}{\det(\Omega_1)}\right).$$

**Lemma D.9** (Self-normalized process [1]). *Let $\{\mathcal{F}_t\}_{t=0}^{\infty}$ be a filtration, and $\{\eta_t\}_{t=1}^{\infty}$ be a real-valued stochastic process such that $\eta_t$ is $\mathcal{F}_t$-measurable and $\eta_t|\mathcal{F}_{t-1}$ is zero-mean (i.e. $\mathbb{E}[\eta_t|\mathcal{F}_{t-1}] = 0$). Assume that conditioning on $\mathcal{F}_t$, $\eta_t$ is $C$-sub-Gaussian. Let $\{\phi_t\}_{t=1}^{\infty}$ be an $\mathbb{R}^d$ real-valued stochastic process such that $\phi_t$ is $\mathcal{F}_t$-measurable. Let $\Omega_0 \in \mathbb{R}^{d \times d}$ be a positive definite matrix and $\Omega_t = \Omega_0 + \sigma^{-2} \sum_{\tau=1}^{t} \phi_\tau \phi_\tau^{T}$. Then for $\delta > 0$, with probability at least $1 - \delta$, for all $t \geq 0$,*

$$\left\|\sum_{\tau=1}^{t} \phi_\tau \eta_\tau\right\|_{\Omega_t^{-1}}^2 \leq 2C^2 \log\left(\frac{\det(\Omega_t)^{1/2}\det(\Omega_0)^{-1/2}}{\delta}\right).$$

**Lemma D.10.** *Suppose $\Omega_0 := \lambda I_d$ is a positive definite matrix in $\mathbb{R}^{d \times d}$ and $\Omega_t = \Omega_0 + \sigma^{-2} \sum_{\tau=1}^{t-1} \phi_\tau \phi_\tau^{T}$.*

$$\frac{\det(\Omega_{t+1})}{\det(\Omega_1)} \leq \left(\frac{\lambda + \sigma^{-2}t}{\lambda}\right)^d.$$

*Proof of Lemma D.10.* By definition, $\det(\Omega_1) = \det(\lambda I) = \lambda^d$. For any $\tau \in \mathbb{Z}_+$ and $\phi_\tau \in R^d$, notice that $\phi_\tau \phi_\tau^{\mathrm{T}}$ is a rank-1 matrix with eigenvalues $\|\phi_\tau\|$ and $0$. By Definition 1 and triangle inequality,

$$\left\|\sum_{\tau=1}^{t} \phi_\tau \phi_\tau^{\mathrm{T}}\right\| \leq \sum_{\tau=1}^{t} \|\phi_\tau \phi_\tau^{\mathrm{T}}\| \leq t.$$

Consider the eigenvalue decomposition for $\sum_{\tau=1}^{t-1} \phi_\tau \phi_\tau^{\mathrm{T}}$:

$$\sum_{\tau=1}^{t-1} \phi_\tau \phi_\tau^{\mathrm{T}} = U\mathrm{diag}(\lambda_1, \ldots, \lambda_d)U^{\mathrm{T}},$$

which suggests

$$\det(\Omega_{t+1}) = \det(\lambda I + \sigma^{-2}\sum_{\tau=1}^{t-1}\phi_\tau\phi_\tau^T) = \prod_{i=1}^{d}(\sigma^{-2}\lambda_i + \lambda) \leq (\sigma^{-2}\max_i|\lambda_i| + \lambda)^d \leq (\lambda + \sigma^{-2}t)^d.$$

$\square$

### D.3 Covering Argument

**Lemma D.11** (Covering number of Euclidean Ball). *Consider an Euclidean ball $B_R$ equipped with the Euclidean metric, whose radius is $R > 0$. The $\epsilon$-covering number of $B_R$ satisfies,*

$$\mathcal{N}_\epsilon(B_R) \leq \left(1 + \frac{2R}{\epsilon}\right)^d.$$

**Lemma D.12.** *Define $\mathcal{V}$ to be a class of values with the parametric form*

$$f_\phi := |\langle\phi,\theta\rangle| - C\sqrt{\phi^\top A \cdot \phi}$$

*where the feature space is $\{\phi : \|\phi\|_2 \leq 1\}$ and $\|A\|_2 \leq B$, $\|\theta\| \leq 2H\sqrt{d}$. Let $\mathcal{N}_\epsilon^{\mathcal{V}}$ be the covering number of $\epsilon$-net with respect to the absolute value distance, then we have*

$$\log\mathcal{N}_\epsilon^{\mathcal{V}} \leq d\log(1 + \frac{4C\sqrt{B} + 4H\sqrt{d}}{\epsilon}).$$

*Proof of Lemma D.12.*

$$|f_{\phi_1} - f_{\phi_2}| \leq \left||\langle\phi_1,\theta\rangle| - C\sqrt{\phi_1^\top A \cdot \phi_1} - (|\langle\phi_2,\theta\rangle| - C\sqrt{\phi_2^\top A \cdot \phi_2})\right|$$

$$\leq \|\phi_1 - \phi_2\| \cdot \|\theta\| + C\sqrt{|\phi_1^\top A \cdot \phi_1 - \phi_2^\top A \cdot \phi_2|}$$

$$\leq \|\phi_1 - \phi_2\| \cdot 2H\sqrt{d} + C\sqrt{\|\phi_1\|\|A\|\|\phi_1 - \phi_2\|} + C\sqrt{\|\phi_1 - \phi_2\|\|A\|\|\phi_2\|}$$

$$\leq \|\phi_1 - \phi_2\| \cdot 2H\sqrt{d} + 2C\sqrt{B\|\phi_1 - \phi_2\|} \leq (2C\sqrt{B} + 2H\sqrt{d\|\phi_1 - \phi_2\|}) \cdot \|\phi_1 - \phi_2\|$$

$$\leq 2C\sqrt{B\|\phi_1 - \phi_2\|} \leq (2C\sqrt{B} + 2H\sqrt{d}) \cdot \|\phi_1 - \phi_2\|$$

Let $\mathcal{C}_\phi$ be the $\frac{\epsilon}{2C\sqrt{B}+2H\sqrt{d}}$-net of space $\{\phi : \|\phi\|_2 \leq 1\}$, then by Lemma D.11,

$$|\mathcal{C}_\phi| \leq (1 + \frac{4C\sqrt{B} + 4H\sqrt{d}}{\epsilon})^d$$

Therefore, the covering number of space $\mathcal{V}$ satisfies

$$\log\mathcal{N}_\epsilon^{\mathcal{V}} \leq d\log(1 + \frac{4C\sqrt{B} + 4H\sqrt{d}}{\epsilon}).$$

$\square$

**Lemma D.13.** *Let $\mathcal{V}$ denote the function class from $\mathcal{S}$ to $\mathbb{R}$*

$$V(\cdot) := \max_a \max_m \phi(\cdot,a)^\mathrm{T}w^m, where \|w^m\| \leq C_{H,d,k,M,\delta}, \forall m \in [M]$$

*let $\mathcal{N}_\epsilon$ be the $\epsilon$-covering number of $\mathcal{V}$ with respect to the distance $\mathbf{dist}(V,V') = \sup_s|V(s) - V'(s)|$. Then*

$$\log\mathcal{N}_\epsilon \leq dM\log(1 + \frac{2C_{H,d,k,M,\delta}}{\epsilon}).$$

*Here $C_{H,d,k,M,\delta} = 2H\sqrt{\frac{dk}{\lambda}} + \frac{\sqrt{2d}+\sqrt{2\log(M/\delta)}}{\sqrt{\lambda}}$.*

*Proof.* Let $V_1 = \max_a \max_m \phi(\cdot, a)^{\mathrm{T}} w_1^m$ and $V_2 = \max_a \max_m \phi(\cdot, a)^{\mathrm{T}} w_2^m$. Then

$$\mathbf{dist}(V_1, V_2) = \max_s |\max_a \max_m \phi(\cdot, a)^{\mathrm{T}} w_1^m - \max_a \max_m \phi(\cdot, a)^{\mathrm{T}} w_2^m|$$
$$\leq \max_{s,a,m} \|\phi(s,a)\| \cdot \|w_1^m - w_2^m\| \leq \max_{s,a,m} \|w_1^m - w_2^m\|,$$

For any $m \in [M]$, let $\mathcal{C}^m$ be the $\epsilon$-net for $\{w^m : \|w^m\| \leq C_{H,d,k,M,\delta}\}$, then by Lemma D.11, $|\mathcal{N}_\epsilon^m| \leq (1 + \frac{2C_{H,d,k,M,\delta}}{\epsilon})^d$, implies the total log covering number

$$\log |\mathcal{N}_\epsilon| \leq \log \Pi_{m=1}^M |\mathcal{N}_\epsilon^m| \leq dM \log(1 + \frac{2C_{H,d,k,M,\delta}}{\epsilon}).$$

$\square$

## D.4 Delayed Feedback

**Lemma D.14** (Lemma 9 of [28]). *Let $A, B \in \mathbb{R}^{d \times d}$ be two symmetric positive semi-definite matrices. Then, $A^{\frac{1}{2}} B A^{\frac{1}{2}}$ and $AB$ share the same set of eigenvalues. Further, these eigenvalues are all non-negative.*

**Lemma D.15.** *Let $\Sigma_h^k, \Omega_h^k, \Lambda_h^k$ be the full design, delayed, and complement matrix respectively. Then $(1 + \frac{U_k}{\lambda})(\Sigma_h^k)^{-1} \succeq (\Omega_h^k)^{-1}$. In addition, with probability $1 - \delta$,*

$$\max_{k \in [K]} U_k \leq \mathbb{E}[\tau] + 2\sqrt{2\mathbb{E}[\tau] \log(3K/2\delta)} + \frac{4}{3} \log(3K/2\delta).$$

*Proof.* The proof follows from Lemma 11 of [28] with $\frac{U_k}{\lambda}(\Sigma_h^k)^{-1} \succeq (\Sigma_h^k)^{-1} \Lambda_h^k (\Omega_h^k)^{-1}$, and then apply Lemma B.2 that $(\Omega_h^k)^{-1} = (\Sigma_h^k)^{-1} + (\Sigma_h^k)^{-1} \Lambda_h^k (\Omega_h^k)^{-1}$. The second part comes from Lemma 4 of [28]. $\square$

# E Experimental Details

In this section, we provide the experimental details of both simulated environments (synthetic linear MDP and RiverSwim) and discuss their results respectively.

## E.1 Delayed-UCBVI

As shown in Table 1 and Section 2, there is no prior UCB method that concerns exactly the same delayed linear MDP setting without resorting to specific policy-switching schemes. To benchmark our posterior sampling algorithms, we modify the existing LSVI-UCB method to accommodate the delayed feedback, which is referred to as the Delayed-UCBVI. Below we include the algorithm of delayed-UCBVI for completeness.

---

**Algorithm 4:** Delayed Value Iteration with UCB (Delayed-UCBVI)

**Input:** bonus parameter $\beta$, regularization $\lambda$.
1 **Initialization:** $\forall k, h, \widetilde{Q}_{H+1}^k(\cdot, \cdot), \widetilde{V}_{H+1}(\cdot, \cdot), \widetilde{V}_h(\cdot, \cdot) \leftarrow 0, \mathcal{D}_h \leftarrow \emptyset$.
2 **for** *episode* $k = 1, \ldots, K$ **do**
3     Sample initial state $s_1^k$
4     **for** *time step* $h = H, \ldots, 1$ **do**
5         $\boldsymbol{y_h} \leftarrow [y_h^1, \ldots, y_h^{k-1}]$, with $y_h^\tau \leftarrow \mathbb{1}_{\tau,k-1} \cdot [r_h^\tau + \widetilde{V}_{h+1}(s_{h+1}^\tau)]$
6         $\Phi_h \leftarrow [\phi^1, \phi^2, \ldots, \phi^{k-1}]$ with $\phi^\tau = \mathbb{1}_{\tau,k-1} \cdot \phi(s_h^\tau, a_h^\tau)$
7         $\Omega_h^k \leftarrow \Phi_h \Phi_h^{\mathrm{T}} + \lambda I$
8         $w_h^k \leftarrow (\Omega_h^k)^{-1} \Phi_h \boldsymbol{y_h}^{\mathrm{T}}$
9         $Q_h^k(\cdot, \cdot) \leftarrow \phi(\cdot, \cdot)^{\mathrm{T}} w_h^k + \beta \sqrt{\phi(\cdot, \cdot)^{\mathrm{T}} (\Omega_h^k)^{-1} \phi(\cdot, \cdot)}$
10         $V_h(\cdot, \cdot) \leftarrow \max_a \min\{Q_h^k(\cdot, a), H - h + 1\}$
11         Update $\pi_h^k(\cdot) \leftarrow \operatorname{argmax}_{a \in \mathcal{A}} \min\{Q_h^k(\cdot, a), H - h + 1\}$
12     **for** *time step* $h = 1, \ldots, H$ **do**
13         Choose action $a_h^k \sim \pi_h^k(s_h^k)$
14         Collect transitions $\mathcal{D}_h \leftarrow \mathcal{D}_h \cup \{(s_h^k, a_h^k, r_h^k, s_{h+1}^k)\}$
        /* Feedback generated in episode $k$ cannot be immediately observed in the presence of delay */

---

## E.2 Synthetic Linear MDP Environment

In this section, we describe the further details in Section 5.1.

**Environment Details.** Following [44, 46, 70], we construct a set of synthetic linear MDP environments with $|\mathcal{S}| = 2$, feature dimension $d = 10$, planning horizon $H = 20$, and varying action space $|\mathcal{A}| \in \{20, 50, 100\}$. Each action $a \in \mathcal{A} \subseteq \{0, 1\}^d$ is encoded with its 8-bit binary representation and represented by a vector $\boldsymbol{b}_a \in \mathbb{R}^8$. The feature map $\phi(\cdot, \cdot)$ can then be defined as

$$\phi(s, a) = [\boldsymbol{b}_a^{\mathrm{T}}, \delta(s, a), 1 - \delta(s, a)]^{\mathrm{T}} \in \mathbb{R}^{10}, \quad \forall (s, a) \in \mathcal{S} \times \mathcal{A},$$

where

$$\delta(s, a) = \begin{cases} 1 & \text{if } \mathbb{1}(s = 0) = \mathbb{1}(a = 0), \\ 0 & \text{otherwise.} \end{cases}$$

In addition, let $\theta_h$ that induces the reward functions $r$ be

$$\theta_h = [0, \ldots, 0, r, 1 - r]^{\mathrm{T}} \in \mathbb{R}^{10},$$

with the choice of $r = 0.99$, and further define the measures $\mu_h$ that govern the transition dynamics $\mathbb{P}$ as

$$\mu_h(s) = [0, \ldots, 0, (1 - s) \oplus \alpha_h, s \oplus \alpha_h],$$

where $\{\alpha_h\}_{h \in [H]} \in \{0, 1\}^H$ is a sequence of integers taking values 0 or 1, $\oplus$ is the XOR operator. By design, the set of environments with identical $d$ and $H$ has the same optimal value $V_1^*(s_1)$.

**Further Results and Discussions.** Figure 2 depicts the empirical distributions of delays considered in section 5.1. Additionally, the average return achieved by each method upon convergence is reported in Table 2, corresponding to the results shown in Figure 1. Our empirical findings indicate that posterior sampling methods excel UCB-based methods in terms of both statistical accuracy and computational efficiency. More specifically, under different types of delays, both Delayed-PSVI and Delayed-LPSVI achieve higher return (lower regret) and exhibit faster convergence compared to Delayed-UCBVI.

While delays following multinomial distribution and Poisson distributions decay exponentially fast, Pareto delays are heavy-tailed. When computational budget is limited or when episodes are finite, feedback is only partially observable under long-tailed delays and is not guaranteed to be revealed to the agent. This setup captures the practical scenarios when small time windows are considered for decision-making or in online recommender systems, where only positive feedback (e.g. click, make a purchase) are often observed. As shown in Table 2, performance of Delayed-UCBVI can dramatically deteriorate in the presence of long-tailed delays.

Furthermore, our results presented in Table 4 and Table 3 illustrate the consistent behavior of posterior sampling in environments with delayed feedback, considering both statistical and computational aspects. When employing feature mapping, performance of the algorithms is much less dependent on the sizes of state and action space in contrast to tabular settings. It is worth noting that in large state and action space, the neighborhoods of a substantial number of state-action pairs may remain unvisited, leading to increased uncertainty in estimation. In such cases, adjusting the scale of exploration by decreasing the noise scaling factor $\sigma$ for Delayed-PSVI can yield faster convergence. Finally, as shown in Table 3, Delayed-LPSVI achieves appealing performance as Delayed-PSVI while reducing computation through the use of approximate sampling with Langevin dynamics.

|  | Multinomial Delay (10, 20, 30) | Poisson Delay ($\mathbb{E}[\tau] = 50$) | Pareto Delay (Shape 1.0, Scale 500) |
|---|---|---|---|
| Delayed-PSVI ($\sigma = 0.1$) | $11.53 \pm 0.76$ | $11.48 \pm 0.81$ | $11.53 \pm 0.74$ |
| Delayed-LPSVI ($c_\eta = 0.5$) | $11.56 \pm 0.48$ | $11.37 \pm 0.48$ | $10.98 \pm 0.40$ |
| Delayed-UCBVI ($c_\beta = 0.1$) | $10.61 \pm 0.76$ | $10.54 \pm 0.81$ | $7.20 \pm 0.38$ |

Table 2: Average return achieved by Delayed-PSVI, Delayed-LPSVI and Delayed-UCBVI upon convergence under different delays. Environment setup: $|\mathcal{S}| = 2$, $|\mathcal{A}| = 20$, $d = 10$, $H = 20$. Optimal average return is $V_1^*(s_1) = 11.96$. Results are obtained over 10 experiments.

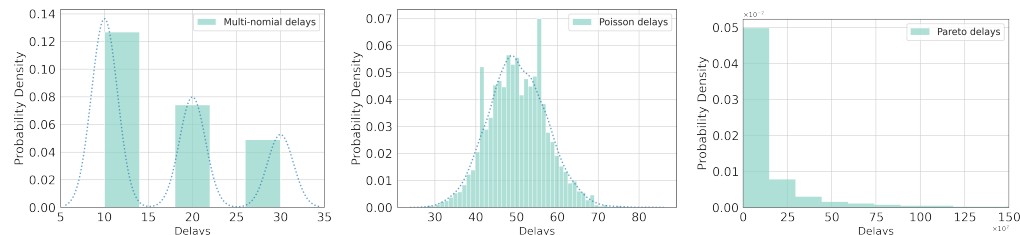

Figure 2: Empirical distributions of three types of delays. (a) Multinomial delays with delay categories $\{10, 20, 30\}$. (b) Poisson delays with rate $\mathbb{E}[\tau] = 50$. (c) Long-tail Pareto delays with shape 1.0, scale 500. The first two types of delays are well-behaved and decay exponentially fast, while pareto delays are heavy-tailed.

|  | $|\mathcal{S}||\mathcal{A}| = 20$ | $|\mathcal{S}||\mathcal{A}| = 40$ | $|\mathcal{S}||\mathcal{A}| = 100$ | $|\mathcal{S}||\mathcal{A}| = 200$ |
|---|---|---|---|---|
| Delayed-PSVI ($\sigma = 0.3$) | 1418 | 1290 | 1669 | 2633 |
| Delayed-PSVI ($\sigma = 0.2$) | 531 | 1114 | 1323 | 826 |
| Delayed-PSVI ($\sigma = 0.1$) | 391 | 571 | 650 | 709 |
| Delayed-LPSVI ($c_\eta = 0.5$) | 293 | 246 | 517 | 566 |
| Delayed-UCBVI ($c_\beta = 0.1$) | 3205 | 2713 | 3351 | 3694 |

Table 3: Number of episodes for each method to achieve its highest expected return. Different synthetic environments are examined with varied $|\mathcal{S}|$ and $|\mathcal{A}|$. Optimal average return is $V_1^*(s_1) = 11.96$ for all environments ($d = 10$, $H = 20$). Results are obtained over 10 experiments with Poisson delays ($\mathbb{E}[\tau] = 50$).

|  | $|\mathcal{S}||\mathcal{A}| = 20$ | $|\mathcal{S}||\mathcal{A}| = 40$ | $|\mathcal{S}||\mathcal{A}| = 100$ | $|\mathcal{S}||\mathcal{A}| = 200$ |
|---|---|---|---|---|
| Delayed-PSVI ($\sigma = 0.3$) | $11.23 \pm 1.00$ | $11.07 \pm 1.05$ | $10.93 \pm 1.11$ | $10.80 \pm 1.13$ |
| Delayed-PSVI ($\sigma = 0.2$) | $11.39 \pm 0.91$ | $11.28 \pm 0.94$ | $11.16 \pm 1.02$ | $11.11 \pm 1.03$ |
| Delayed-PSVI ($\sigma = 0.1$) | $11.57 \pm 0.74$ | $11.48 \pm 0.81$ | $11.39 \pm 0.86$ | $11.33 \pm 0.92$ |
| Delayed-LPSVI ($c_\eta = 0.5$) | $11.31 \pm 0.46$ | $11.37 \pm 0.48$ | $11.57 \pm 0.48$ | $11.57 \pm 0.78$ |
| Delayed-UCBVI ($c_\beta = 0.1$) | $10.98 \pm 1.78$ | $10.54 \pm 0.81$ | $9.67 \pm 0.54$ | $10.01 \pm 0.16$ |

Table 4: Average return achieved by Delayed-PSVI, Delayed-LPSVI and Delayed-UCBVI upon convergence in different linear MDP environments with varied $|\mathcal{S}|$ and $|\mathcal{A}|$. Optimal average return is $V_1^*(s_1) = 11.96$ for all environments ($d = 10$, $H = 20$). Results are obtained over 10 experiments with Poisson delays ($\mathbb{E}[\tau] = 50$).

### E.3 RiverSwim

RiverSwim environment is known to be a difficult exploration problem for least-squares value iteration with $\epsilon$-greedy exploration due to the sparse reward setting. It models an agent swimming in the river who can either swim towards the right (against the current) or towards the left (with the current). While trying to move rightwards may fail with some probability, moving leftwards always yield successful transition. We consider the environment with linear feature maps where $|\mathcal{S}| = 5$, $d = 10$, $H = 20$, and Poisson delays. Accordingly, the tabular environment can be recovered with canonical basis in $\mathbb{R}^d$ as its feature mapping:

$$\phi(s, a) = \boldsymbol{e}_{s,a} \in \mathbb{R}^{10}, \qquad (s, a) \in \mathcal{S} \times \mathcal{A}.$$

Define $\theta_h$ as

$$\theta_h(s, a) = [0.005, 0, \ldots, 0, 1.0]^{\mathrm{T}} \in \mathbb{R}^{10},$$

then reward functions induced by $\theta_h$ are given by:

$$r_h(s, a) = \begin{cases} 0.005 & \text{if } s = 0, a = \text{left;} \\ 1.0 & \text{if } s = 4, a = \text{right;} \\ 0.0 & \text{otherwise.} \end{cases}$$

In this environment, We warm start LMC for Delayed-LPSVI by reusing the previous sample for initialization, and set $M = 2$, $N = 40$, $\eta = c_\eta / \lambda_{\max}(\Omega_h^k)$, $\gamma = c_\gamma^2 dMH^2$. We set parameters $M = 2$, $\nu = 1.0$ for Delayed-PSVI, and the bonus coefficient in Delayed-UCBVI as $\beta_h^k = c_\beta / 2 \cdot$

$d\sqrt{k}(H - h)$. Optimal hyperparameters are determined by gridsearch and we fix $c_\beta = 0.04$, $c_\eta = 0.5$, $c_\gamma = 0.005$, $\sigma = 1.13$. Experiments are repeated with 5 different random seeds. Cumulative regrets are then depicted in Figure 3.

**Results and Discussions.** Compared to the previous synthetic environment where dense rewards are available, posterior sampling methods are shown to be robust with spare rewards even in the presence of delays. Figure 3 shows that both Delayed-PSVI and Delayed-LPSVI outperform Delayed-UCBVI in delayed-feedback settings with linear function approximation. In particular, LMC (Algorithm 3) provides strong concentration such that Delayed-LPSVI is able to maintain the order-optimal regret as Delayed-PSVI when exploring the value-function space.

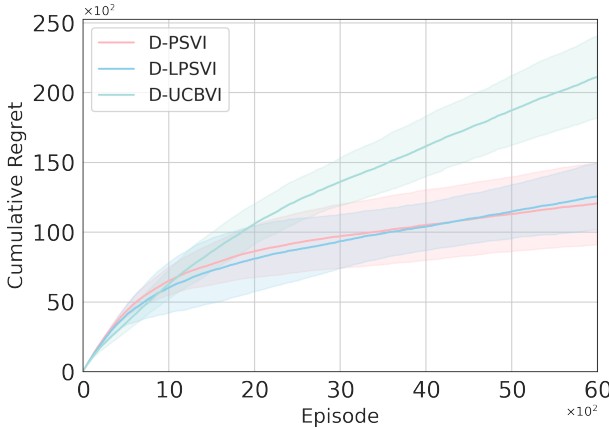

Figure 3: Delayed-PSVI and Delayed-LPSVI outperform Delayed-UCBVI in sparse-reward setting with Poisson delays ($\mathbb{E}[\tau] = 5$). Results are reported over 5 experiments.

