# OpenReview forum: "Posterior Sampling with Delayed Feedback for Reinforcement Learning with Linear Function Approximation"
_NeurIPS.cc/2023/Conference — NeurIPS 2023 poster_

### Official Review · Reviewer_TvPe · 2023-07-02

**Soundness:** 3 good
**Presentation:** 1 poor
**Contribution:** 2 fair
**Rating:** 5
**Confidence:** 3

**Summary:**

This paper introduces and analyzes two "posterior sampling" algorithms for learning optimal policies in linear MDPs in a setting where feedback from the environment is delayed (by stochastic amounts of time). The algorithms, called Delayed-PSVI and Delayed-LPSVI (namely Delayed (Langevin) Posterior Sampling Value Iteration) explore by randomly sampling value functions from some notion of a posterior, which concentrates on the optimal value function. The Delayed-LPSVI algorithm circumvents some computational burdens in the inference and sampling procedure of Delayed-PSVI, and instead performs the "posterior sampling" with noisy gradient descent updates (akin to SGLD). It is established that the Delayed-[L]PSVI algorithms achieve sublinear regret with respect to the time horizon, which grows linearly with the mean feedback delay. Comparing to a simple modification to an existing optimism-based algorithm for linear MDPs to handle the delayed feedback, the authors demonstrate empirically in some simple environments that their posterior sampling algorithms tend to earn high return more quickly (and it is implied therefore that regret grows more slowly).

**Strengths:**

Delayed feedback is an interesting, relevant, and under-explored topic in RL research. This paper provides thorough analysis for posterior sampling algorithms with delayed feedback, with detailed analysis and with empirical evaluation. The algorithms appear to be reasonable and simple to implement.

**Weaknesses:**

Firstly, while I believe the problem of delayed feedback is an important one to study, I do not particularly understand the motivation and the logistics of the particular setting of delayed feedback that is studied in the paper. In this work, to my understanding, all data from an entire trajectory is kept hidden from the agent until a random future time. As I elaborate on further in the Questions section, I do not understand how hiding this information even allows any feedback control, and I cannot think of a real scenario that this delay models. There is very little discussion about the motivation for this setting.

Moreover, it is not entirely clear to me how the delayed feedback makes the analysis of Delayed-[L]PSVI substantially different from the algorithms they're based on. Suppose that instead of the proposed algorithm, we employ [L]PSVI but only update the value function when new data becomes available. Intuitively, I would expect the regret to grow by a factor of $\mathbb{E}[\tau]$ roughly. It would be nice to have an intuitive understanding of how Delayed-[L]PSVI circumvents this multiplicative increase in regret.

Furthermore, perhaps this is due to the fact that the paper proposes a new form of delayed feedback, but the baseline algorithm (to my understanding) is not actually rigorously justified. Thus, it is not clear that it is in fact the posterior sampling that explains the superior performance of Delayed-[L]PSVI.

**Questions:**

I am confused about how exactly the delayed feedback works. For example, in Algorithms 1, 2, 4, during the data collection phase, there is a comment saying "Environment rollouts cannot be observed if there is a delay". What does this mean? If the rollouts are not observed, which states are being fed to the policy to generate rollouts (i.e., how is $a_h^k\sim\pi_h^k(s_h^k)$ computed if $s_h^k$ is not observed)?

The only way I can see this working is if the agent can observe its trajectory online, but then cannot use the data it collects for updates until a random time in the future. But what is the motivation for such a model?

**Limitations:**

As mentioned above, it is unclear what types of real problems are modeled by the particular delayed-feedback setting that is proposed.

---

> ### Author Rebuttal · Authors · 2023-08-10
>
> We sincerely appreciate Reviewer TvPe's thoughtful review of our work. We are grateful for the detailed feedback, which provides valuable insights.
>
>     1. Common ground of delayed feedback definition with existing literature
> By Definition 2, we call each trajectory (i.e. rollouts) $[s^t_h,a^t_h,r^t_h,s^t_{h+1}]_{h\in[H]}$ as feedback at episode k.
>
> In our algorithms, by saying "Environment rollouts cannot be observed if there is a delay", we mean that at episode $k$, if a non-negative delay $\tau_k$ takes place, the feedback $[s^t_h,a^t_h,r^t_h,s^t_{h+1}]_{h\in[H]}$ will not be observed until the end of episode $k+\tau_k$. The formal definition of delayed feedback is provided at the end of Section 2 (Line 130 - 137), and the way to encode the delayed feedback in our algorithm is discussed in Line 179 - 183.
>
> Essentially, the introduction of delays causes the feedback in an episode to return at a future time $k+\tau_k$.  As a result, at the beginning of episode $k+1$, the algorithm can only update posteriors using the observed feedback it has observed (feedback associated with episodes $i$ s.t. $i + \tau_i < k + 1$),  but not feedback from all previous episodes $[i]_{i=1}^{k}$.
>
> Note that this setting is a counterpart of delayed feedback in bandit literature such as [Vernade et al 2023], where a censoring variable is defined for delayed feedback to indicate whether or not the reward in round $s$ is revealed by round $t$.  This setting is also similar to concurrent works for RL with delayed feedback, e.g. [Howson, et al 2023].
>
> This setting is motivated by various practical scenarios. For instance, in recommender systems, feedback from users (e.g. clicks, purchases) often comes in a batched manner and may be observed only after a certain time window (i.e., delays occur by hours, days or even weeks depending on the types of feedback).  In clinical trials, the effectiveness of treatments can only be determined at a deferred time frame. Our setting is to model such sequential decision-making problems with delayed feedback.
>
> We will further clarify the idea in our revised manuscript.
>
>     2. Contributions and novelties compared to the existing literature in delayed feedback
>
> (a) Existing bandit literature typically studies delays either in rewards or in the selection of arms independently. No prior work jointly studies the effect of delays that take place simultaneously in different aspects. However, in RL, delays can be observed not only in rewards, but also in state transitions, actions which in turn will also affect the transition. Therefore, in our work, we study a more challenging setting by considering delays with respect to the whole episodic trajectory that simultaneously includes transitions, actions, and rewards.
>
> (b) We study the more practical setting of stochastic delays, which is unknown and random at each episode, instead of making the simpler assumption that delays are fixed and constant.
>
>     3. Techincal clarification in obtaining sub-linear regret by controlling the delayed error
>
> In our regret analysis, we decompose the regret into two parts, which capture the error from optimism and estimation respectively. To obtain a sublinear regret, we need to strike a balance between both terms. The presence of delays prohibits the direct application of Elliptical Potential Lemma to bound the term $\sum_{k=1}^K||\phi(s^k, a^k)||_{(\Omega^k)^{-1}}$ in regret decomposition as in the non-delayed case. To address this issue, we decompose the covariance matrix of posteriors into two parts: a full design matrix and a complement matrix. Note that this design is dedicated to delayed settings and requires more careful analysis to bound the delayed error incurred. Details can be found in Appendix B.1, where Lemma B.3 and Lemma B.4 are the key lemmas in addressing it.
>
> Once again, thank you for dedicating time to reviewing our work and for providing valuable insights that will contribute to its improvement.
>
>     Reference
>
> Vernade, Claire, et al. "Linear bandits with stochastic delayed feedback." International Conference on Machine Learning. PMLR, 2020.
>
> Howson, Benjamin, et al. "Optimism and delays in episodic reinforcement learning." International Conference on Artificial Intelligence and Statistics. PMLR, 2023.

---

> > ### Comment · Reviewer_TvPe · 2023-08-11
> > **Clarifications about delay**
> >
> > Thanks to the authors for the response. I am still not clear on the delay model. There is a fundamental difference between the motivating examples that are presented (i.e., recommender systems, clinical trials, etc) and the general RL problem, as I understand it. Taking the recommender system problem as an example, for instance, you are probably only observing some state/context at timestep 0, and you can subsequently make decisions thereafter without conditioning on a changing state. In RL, the policy is a mapping from states to actions. Given that your delay model (to my understanding) prohibits the agent from observing its current state, how is it executing a policy? Are your algorithms learning 'open-loop policies' (i.e., full sequences of actions that are not influenced by state feedback)? This would have some major consequences, especially if the transition dynamics are stochastic.
> >
> > My instinct is that there is a mistake in Definition 2 (or at least my understanding of it is wrong) -- I think it would make sense if only the rewards are delayed. In this case, the agent can still execute a feedback policy, and it just cannot update its value function until after the delay.

---

> > > ### Author Response · Authors · 2023-08-11
> > > **Clarifications about delay**
> > >
> > > Thank you for your further questions! We greatly appreciate the opportunity to provide additional clarity and address your concerns.
> > >
> > > Here, **delays are introduced at the episode level (i.e. across different episodes)** and are considered exclusively for posterior updates, which **do not prohibit the agent from playing, executing policy, or choosing actions within each episode**. Instead, delays affect the timing of when the current information can be used in the future during the learning/update process. At the beginning of each episode, the agent updates the policy based on the estimated value functions, which are determined by some parameter $[w_h^k]_{h=1}^H$. Once these parameters are determined (through posterior sampling), value estimations and policy are fixed, then the agent plays and executes the policy as in the usual no-delay case. It is only when the posteriors need to be updated in the next episode that delays come into play. Thus, delay $\tau_k$ influences subsequent episodes only during posterior (parameters) updates, but does not affect episode $k$ itself.
> > >
> > > This characteristic results from the episodic setting that we study, which differs from the infinite time-horizon setting. In episodic MDP setting, upon the rollout completion at step $H$, episode $k$ terminates, then the state of the agent is reset to $s_{1}^{k+1}$ (according to the initial state distribution) for the next episode $k+1$ to begin a new rollout. Consequently, the Markovian dependencies are not carried over across episodes. Posteriors only need to be updated once at the start of each episode, and we only need to consider delays at this point to determine the applicable data.
> > >
> > > Roughly speaking, in **episodic setting**, our algorithms maintain $H$ posterior distributions (one for each step $h \in [H]$) to distinguish the behavior at each step. For every step $h \in [H]$, the corresponding posterior is updated conditioning on the available feedback **at the same step $h$ from previous episodes** (Line 8 and Line 5 in Algorithm 1).
> > >
> > > In contrast, **infinite time-horizon setting** (commonly employed in control and continuous tasks) lacks state reset or episodes. Posteriors need to be updated conditioning on available data from **all previous time steps**.  Updates are necessary at each time step unless low-switching schemes are used. In this context, considering delays only in relation to rewards, like in bandits, is indeed a more natural (and simpler) choice. This setting however differs from our setting.
> > >
> > > Thank you for bringing it up! We deeply value your time and commitment to seeking a clear understanding of our delayed feedback model, and we are committed to addressing this confusion comprehensively in our revised manuscript.
> > >
> > > We appreciate your continued engagement and valuable feedback.

---

> > > > ### Comment · Reviewer_TvPe · 2023-08-14
> > > >
> > > > Thanks for the clarification. My remaining question, then, is about what types of real problems are captured by this delay model. If an agent is able to observe certain data to make transitions, why can it not store the data and use it for updates at the beginning of the next episode?
> > > >
> > > > What I've gathered is that, to my understanding, rewards are not observed until the random delay (this makes sense). My question then boils down to what the difference is between the following settings:
> > > >
> > > > 1. Standard episodic RL setting where only the rewards are hidden until a random future time
> > > > 2. The setting in this paper, where additionally the state transitions cannot be used for updates until the reward is revealed.
> > > >
> > > > Is setting (1) easier than setting (2)? I would assume so, since the transition information should theoretically permit faster concentration of the posterior over MDPs. If that's the case, then it seems like the proposed algorithms could be improved using information that it is available to them (namely, the state transitions), and I do not understand the motivation for setting (2) relative to setting (1).

---

> > > > > ### Author Response · Authors · 2023-08-15
> > > > >
> > > > > Thank you for your follow-up question!
> > > > >
> > > > > **Q1:  Why not store the data and use it for updates at the beginning of the next episode?**
> > > > >
> > > > > **A1:** This is exactly the difference between no-delay setting and the delayed setting, as well as the difference between delays in bandits and delays in RL. More specifically,
> > > > > - If data can be used immediately to update the posteriors in the next episode, it means $\tau_k = 0$, which corresponds to no-delay setting.
> > > > > - In episodic RL, an episode has $H$ steps. An agent needs to complete a whole rollout (which involves $H$ steps) using the current posterior till the end of the episode, then store the feedback captured in this rollout. Whether this new feedback can be used to update the posterior in the next episode depends on whether $\tau_k = 0$.
> > > > > - The linear bandit setting can be seen as a special case of linear MDP where $H=1$. That is why after each step (instead of after each episode), delays need to be considered to see whether it affects the next decision.
> > > > > - Note that this setting is consistent with concurrent work [Howson et al 2023], which studies delays in episodic tabular RL. In their work, they consider delays $\tau_k$ at the episodic level, which affects the observation of visiting counts of state-action pairs $(s, a)$. They do not assume delays take place in rewards, and only study UCB-based methods for tabular settings. More details can be found in Section 3 of [Howson et al 2023].
> > > > >
> > > > > **Q2: Difference between settings where delays take place in rewards v.s. delays take place in both rewards and transition**
> > > > >
> > > > > **A2:** The reason why we care about delays in both state transitions and rewards mainly lies in the fact that we study linear MDP setting and the fact that posterior distributions are posited in the space of value functions.
> > > > > - We aim to learn posterior distributions for value functions,  which encode the information from both transitions and rewards, considering delays in both state transitions and rewards is a natural choice.
> > > > > -  By definition of linear MDP (Definition 1), $Q$ function is linear in feature map for some parameter $w^*$ (see Lemma A.1), where $w^*$ governs both transitions and rewards. To learn its posteriors at each step, feedback information in both observed transitions and rewards is required. If delays are considered for rewards only in linear MDP case,
> > > > >     - from the algorithmic perspective, one may need to maintain two kinds of posterior distributions, one for reward functions (where delays take place), the other is for transitions (where feedback is instant at the end of each episode), and combine two kinds posteriors together in order to learn the posteriors for value functions, which could be highly inefficient in high-dimensional space. More importantly, the combined posterior for value function could potentially be a biased estimate.
> > > > >     - from the theory perspective, in linear MDP,  in order to analyze the effect of delays, we need to decouple the analysis for transitions and reward functions to isolate the parts where delays come into play.
> > > > >
> > > > > **Q3: real problems captured by the delayed setting**
> > > > >
> > > > > A3: Delays in both state transitions and rewards can be commonly seen in healthcare treatment and recommender systems.
> > > > > - In recommender system, states can include information of the user's current preferences, historical purchase behavior, and interactions with the platform, e.g. the user's browsing history, items added to the shopping cart, and items previously purchased. The actions correspond to product recommendations. The rewards indicate user's satisfaction or dissatisfaction with the recommended products over time. Both state transitions (user interactions, preferences) and rewards (user satisfaction) are subject to delays. When a user interacts with a recommended product, the impact of that interaction and user satisfaction may only be known after a certain time window.
> > > > > - In a medical setting where physicians are making treatment decisions for patients with chronic conditions, states include information about the patient's health status, including blood glucose levels, medication dosage, dietary habits, and exercise routines. The actions correspond to treatment decisions, such as adjusting medication dosages, recommending dietary changes, or prescribing exercise regimens. The rewards depend on the long-term impact of treatment decisions on the patient's health.  Both state transitions (patient health changes) and rewards (health outcomes) are subject to delays, as the effects of treatment decisions unfold over time.
> > > > >
> > > > > We hope that this helps further clarify the setting and address your concerns. Once again, thank you for your valuable comments, we are committed to incorporating the clarification in our revised version.
> > > > >
> > > > > ****
> > > > >      Reference:
> > > > >      Howson, Benjamin, et al. "Optimism and delays in episodic reinforcement learning." International Conference on Artificial Intelligence and Statistics. PMLR, 2023.

---

> > > > > > ### Comment · Reviewer_TvPe · 2023-08-15
> > > > > >
> > > > > > > A1: This is exactly the difference between no-delay setting and the delayed setting, as well as the difference between delays in bandits and delays in RL.
> > > > > >
> > > > > > I don't really agree... I'm struggling to understand a setting where the agent can observe state transitions, but be forced to wait a random time until they can use that information for updates.
> > > > > >
> > > > > > Having said that, upon reading your reply, I think this is a moot point -- for the purpose of this paper, as I understand it from your reply, there is no clear way to update the posteriors without reward information (since you're modeling a posterior over value functions as opposed to a posterior over MDPs), so you might as well wait the random delay for reward data before using the observed state transitions anyway. I suppose this is fair. Personally, I still think it makes more sense to not impose this 'hidden state transition' constraint -- even if we do not yet know how to incorporate the state transition data to improve the posterior over value functions, perhaps this is an interesting avenue for future work.
> > > > > >
> > > > > > Regarding your examples listed under "Delays in both state transitions and rewards can be commonly seen in healthcare treatment and recommender systems" in the previous comment, it sounds to me like what is being described is not an issue of delayed state feedback (because again, you cannot delay state feedback unless you are using an open loop policy), but rather non-Markovian rewards/transitions and/or partially observability, both of which are not modeled by the linear MDP setting in the paper.
> > > > > >
> > > > > > Anyway, the previous comment did clarify my main concern to some extent, and I raised my score. Please let me know if I am still missing something.

---

> > > > > > > ### Author Response · Authors · 2023-08-17
> > > > > > >
> > > > > > > We are grateful for your efforts in engaging with our work and your support in raising your score. Your assessment has been invaluable in refining our work and clarifying the key aspects of our research.
> > > > > > >
> > > > > > > We acknowledge that various modeling choices are available, and the decision to adopt a POMDP or a linear MDP framework with delayed feedback is indeed a consideration of how best to represent and analyze the problem at hand. While POMDP assumes that certain information is always not directly observable and needs to be inferred based on available observations, delayed feedback setting emphasizes the uncertain temporal delays that hinder timely updates in the learning process, in which feedback can still be eventually observed and utilized. We aim to provide a structured lens to study the performance of sequential decision-making in the presence of delays from the theoretical perspective. In concurrent theoretical studies ([Yang, et al 2023] and [Howson et al 2023]), akin to ours, the focus lies in studying how delays affect the update/learning process, which only occurs at the beginning of each episode. After that, the posterior of value functions and the corresponding policy remain fixed, and do not require further update/learning within the episode. This setting, while may not encompass all practical elements, allows us to develop an understanding of the interplay between delays, posterior sampling, and RL with function approximation. The exploration of this topic is still in its nascent stages, and we hope that this initial work will inspire more future studies in this compelling direction.
> > > > > > >
> > > > > > > Once again, we sincerely thank you for your time, thoughtful feedback, willingness to engage, your support in raising your score, and we look forward to incorporating these discussions into our revised manuscript.

---

### Official Review · Reviewer_7RMs · 2023-07-06

**Soundness:** 3 good
**Presentation:** 2 fair
**Contribution:** 2 fair
**Rating:** 5
**Confidence:** 2

**Summary:**

The paper discusses the challenges of delayed feedback in real-world systems for reinforcement learning and proposes an optimistic algorithm that utilizes noise perturbation via posterior sampling to effectively explore. The algorithm is shown to achieve a worse-case regret that depends on the delay and time horizon, and the paper provides theoretical analysis and empirical results to support the effectiveness of the proposed algorithm.

**Strengths:**

1. The paper addresses a practical problem in reinforcement learning which is the challenge of delayed feedback in real-world systems.
2. The paper provides both theoretical analysis and empirical results to support the effectiveness of the algorithm.

**Weaknesses:**

1. Algorithm-wise, it is not clear what is the core idea to handle the delayed feedback problem, i.e., the key difference between delayed PSVI and normal PSVI (normal PSVI could be adjusted to update only when the feedback is received to handle the delayed feedback case) (See Q1).
2. Theory-wise, it is not clear what is the improvement from previous work also designed to handle delayed feedback (see Q2).
3. Experimental-wise, it is good to see a comparison with UCB, but LSVI-UCB is not designed to handle delayed feedback. So it might not be a fair comparison (see Q2).

**Questions:**

1. The proposed methods (encode value iteration, multi-round sampling) do not seem to be specifically designed for the delayed feedback case, could you explain why these techniques could improve the results with delayed feedback? Also, a straightforward approach could be just to apply PSRL with value iteration and only update the model when the feedback is received. How is that different from the current approach? If there are key differences, it would be a natural baseline missing from the current experimental section.
2. In Table 1, there is no comparison with other algorithms that handle the delayed feedback problem. Is it possible to compare with other SOTA methods that also address this problem? Also, the same question applies to the experimental section.

**Limitations:**

The paper could benefit from a comprehensive discussion about its limitations like the linear assumption, synthetic experiments, etc.

---

> ### Author Rebuttal · Authors · 2023-08-10
>
> We appreciate Reviewer 7RMs' thorough evaluation of our work. We thank the reviewer for providing valuable feedback that will contribute to the refinement of our work.
>
>     1. Clarification of technical novelty (theory)
> Theory-wise, our main technical contributions and novelties are as follows:
>
> (a) Concentration: compared to the standard setting without delays, the posterior concentration is weaker when delays take place. An agent needs to take actions based on limited observed history and existing posteriors, which can cause distribution shift. The key is to show that even with weaker posterior concentration, our algorithm is able to achieve the desired concentration and anti-concentration criteria that guarantee the sub-linear regret. In particular, for algorithm 2, we show that the convergence guarantee provided by LMC is strong enough to yield approximate posteriors with high accuracy even with fewer available samples due to delays (Lemma C.2).
>
> (b) Delay errors: The presence of delays prohibits the direct application of Elliptical Potential Lemma to bound the term $\sum_{k=1}^K||\phi(s^k, a^k)||_{(\Omega^k)^{-1}}$ in regret decomposition as in the non-delayed case. To address this issue, we decompose the covariance matrix of posteriors into two parts: a full design matrix and a complement matrix. Note that this design is dedicated to delayed settings and requires more careful analysis to bound the delayed error incurred. Details can be found in Appendix B.1, where Lemma B.3 and Lemma B.4 are the key lemmas in addressing it.
>
>     2. Clarification of technical novelty (algorithm)
> From the algorithmic perspective, we would like to highlight that,
>
> (a) We provably incorporate approximate sampling scheme by resorting to Langevin Monte Carlo (LMC) in algorithm 2, which is an important extension from the standard posterior sampling algorithms. By utilizing LMC, algorithm 2 is proved to enjoy computational efficiency while maintaining the same regret guarantee (Thm 2). Besides, owing to the ability of LMC, algorithm 2 (Delayed-LPSVI) is able to generate samples from arbitrarily complex posteriors in complex problem domains, where posterior distributions can be intractable without closed-form. When standard posterior sampling fails to capture the noise model accurately, algorithm 2 (Delayed-LPSVI) is expected to come into play. To the best of our knowledge, this is the first analysis that provably incorporates LMC in linear MDPs and jointly considers the effect of stochastic delays.
>
> (b) The general workflow of our algorithms follows a similar philosophy as posterior sampling in standard no-delay RL settings, but with multi-round parameter, the scaling factor of the posterior variance, and the resulting changes in Bayesian linear models that are adapted to account for the delays. This is expected, one can find that most algorithms that study delay feedback (e.g. [Vernade et al 2020, Yang et al 2023]) also result in similar algorithmic procedures as in no-delay setting, but require the computation of new confidence interval / exploration bonus / posterior variance due to the weaker concentration arising from delays. In general, this is challenging, as
> (i) we need to balance between concentration and anti-concentration to make sure the resulting posterior sampling algorithm provides optimistic estimates that do not deviate too much from the true values even when delays take place; (2) it is important to guarantee that the number of runs for sampling (i.e. multi-round parameter) has proper scaling in time horizon. We show that in Lemma B.7, our choice of M has order $Polylog(H, K, d, \delta)$ and thus does not increase the complexity dependence.
>
>     3. Comparison with other algorithms under delayed feedback
> Theoretical understanding of delayed feedback is mainly studied in bandit settings, and is less underexplored in RL until very recently.  We summarize and compare our work with all existing theoretical works in RL with delayed feedback that we are aware of:
>
> (a) Different RL settings: while [Lancewicki et al. 2022] and [Jin et al. 2022] consider adversarial MDPs, concurrent works [Howson et al 2023] and [Mondal and Aggarwal 2023] study episodic tabular MDPs, our work focuses on linear MDPs.
>
> (b) In [Yang et al 2023], they propose a reduced framework for different sequential decision making problems. However, their framework relies on policy-switching schemes, which differ from our settings with fixed episodic length.
>
> Therefore, there is no prior theoretical work that studies exactly the same setting as ours. As mentioned above, the general algorithmic workflows in no-delay and delayed settings are similar, one can easily adapt the UCBVI algorithm to fit the delayed linear MDP setting and test the performance against it. For a fair comparison, we grid search to choose the optimal parameters for Dealyed-UCBVI in experiments.
>
> Once again, thank you for dedicating time to reviewing our work and for providing valuable insights that will contribute to its improvement.
>
>     Reference:
> Vernade, Claire, et al. "Linear bandits with stochastic delayed feedback." International Conference on Machine Learning. PMLR, 2020.
>
> Yang, Yunchang, et al. "A Reduction-based Framework for Sequential Decision Making with Delayed Feedback." arXiv preprint arXiv:2302.01477 (2023).
>
> Lancewicki, Tal, et al. "Learning adversarial markov decision processes with delayed feedback." Proceedings of the AAAI Conference on Artificial Intelligence.  2022.
>
> Jin, Tiancheng, et al. "Near-optimal regret for adversarial mdp with delayed bandit feedback." Advances in Neural Information Processing Systems 35, 2022.
>
> Howson, Benjamin, et al. "Optimism and delays in episodic reinforcement learning." International Conference on Artificial Intelligence and Statistics. PMLR, 2023.
>
> Mondal, W. U., & Aggarwal, V. Reinforcement Learning with Delayed, Composite, and Partially Anonymous Reward. arXiv preprint arXiv:2305.02527, 2023.

---

> > ### Comment · Reviewer_7RMs · 2023-08-17
> >
> > Thank you for the reply and it resolves some of my concerns. I've updated my score.

---

> > > ### Author Response · Authors · 2023-08-17
> > >
> > > We are pleased to learn that our clarifications have helped address your main concerns. We thank you for your support in raising your score and assessment. We are committed to incorporating the clarifications comprehensively in our revised manuscript.
> > >
> > > Thank you once again for your time and consideration.

---

### Official Review · Reviewer_U3pE · 2023-07-10

**Soundness:** 2 fair
**Presentation:** 1 poor
**Contribution:** 3 good
**Rating:** 5
**Confidence:** 3

**Summary:**

The authors study the frequentist regret of TS under stochastic delayed feedback with linear function approximation. They further proposed using LMC as a posterior sampling strategy to improve computational efficiency.

**Strengths:**

1. As far as I know, this is little work that develops frequentist regret bound with delayed feedback
2. The paper provides a theoretical understanding of using LMC for posterior sampling

**Weaknesses:**

1. Overall, the theories look sound to me (I read Appendix B closely and skimmed Appendix C as it is very similar to Appendix B). However, the math is quite messy (such as missing/inconsistent notations, missing assumptions, incorrect references, incorrect constants, etc), which makes the proof hard to follow. Some examples (not exhaustive) are listed below but the manuscript should be proofread carefully
    1. Definition 1 should include $r_h\in[0,1]$ (given the value function is upper bounded by H-h+1)
    2.  $\tilde{V}$,  $\tilde{Q}$ should be introduced or referred to algorithm 1 before using. in line 517, $\tilde{V_h}^k$ is defined different from line 13 in algorithm 1.
    3. In algorithm 1 line 7, $\hat{w}$ should not be divided by $\sigma^{-2}$
    4. In theorem 2, $\kappa_h$ is defined in appendix instead of the main manuscript. $\eta_k$ is not used in theorem 2
    5. In appendix section B, all Lemma C.5 should be B.8
    6. Every lemma/theorem should use the phrase "with probability **at least**"
    7. In Appendix line 648, please show that $\theta=\tilde{w}-\hat{w}<=2H\sqrt{d}$ as stated in the assumption of Lemma D.12  under event $\tilde{E}$
    8. In Lemma D.7, $d+2\sqrt{dt}+2t \neq 2(\sqrt{d}+\sqrt{t})^2$. The definition of constant $C_{H,d,k,M,\delta}$ is wrong.
2. The computational efficiency/overall performance of D-LPSVI should be demonstrated in high-dmensional domain
3. Missing reference
    - LMC is studied in general bandit setting (e.g. Mazumdar, Eric, et al. On Thompson Sampling with Langevin Algorithms)
    - In the discussion of arbitrary delayed feedback, please include Wu, Han, and Stefan Wager. Thompson sampling with unrestricted delays.

**Questions:**

1. In Figure 1, why are there non-smooth jumps for average return of D-UCBVI
2. Please provide some intuition on the bound of $N_k$ (how the condition number scales w.r.t K,H)
3. To reduce computational complexity, one can also consider updating the parameters less frequently (the feedbacks come in with batches of episodes). I'm curious how this will scale the regret bound

**Limitations:**

Limitations are addressed as future work in Section 6

---

> ### Author Rebuttal · Authors · 2023-08-10
>
> We appreciate Reviewer U3pE's thorough evaluation of our work. We thank the reviewer for providing valuable feedback that will contribute to the refinement of our work.
>
>     1. Clarification on mathematical definitions and notation issues
> We thank the reviewer for pointing out the unclear definition and missing details. We will address the issues related to notations and definitions in the revised manuscript to ensure consistency and correctness, which is an easy fix from the current version.
>
> We would like to point out that:
>
> (a) We define $r_h: \mathcal{S} \times \mathcal{A} \rightarrow [0, 1]$ in Line 104.
>
> (b) The scaling of $\sigma^{-2}$ comes from the calculation of Bayesian linear regression model, which appears when noise is not from standard Gaussian.
>
> (c) $\eta_k$ is presented in theorem 2 as it is a parameter (learning rate) that is required to run Langevin Monte Carlo for Algorithm 2.
>
>     2. Performance validation of Delayed-LPSVI
> Theoretically, we show in theorem 2 that by adopting Langevin Monte Carlo as an approximate sampling scheme, Delayed-LPSVI enjoys the same regret guarantee as Delayed-PSVI. In addition, it further improves computational efficiency compared to Delayed-LPSVI. Empirically, in experiments, we also measure the computational efficiency in a set of synthetic linear MDPs. As shown Table 3 (Appendix E.2), Delayed-LPSVI requires the least number of episodes to achieve its highest expected return.
>
>     3. Related works
> We have already cited the mentioned paper for LMC. In addition, we do provide a thorough discussion of LMC methods in different sequential decision-making frameworks in section 1.1 (related works). We thank the author for providing further reference for arbitrary delayed feedback. We will include it as well as the concurrent works that study RL with delayed feedback in our revised manuscript.
>
>     4. Clarification of Figure 1
> It is known that in general, posterior sampling algorithms bear greater robustness in the presence of delays compared to UCB-based methods. The jump in Delayed-UCBVI suggests in the testing environments, it is shortsighted and prone to adopt sub-optimal policies with insufficient exploration. In comparison, Delayed-PSVI and Delayed-LPSVI are randomized Bayesian algorithms, and thus are more effective in discovering optimal policies.
>
>     5.  Intuition on the bound of $N_k$
> Condition number $\kappa_h := \frac{\lambda_{\max}(\Omega^k_h)}{\lambda_{\min}(\Omega^k_h)}$. In our case, the number of iterations $N_k$ required for Lagevin Monte Carlo to converge scales with $Polylog(H, K, d)$.
>
>     6. Comment on further reducing computational complexity
> To consider updating the parameters less frequently, one can adopt low-switching schemes in RL, which is a different focus compared to the current work. It should be noted that with low-switching schemes, the benefit is to bring down the communication cost, but will not  further improve the performance. Our current algorithms achieve the optimal dependence on the parameters $d$ and $T$ under the class
> of posterior sampling algorithms. Careful choice of low-switching schemes is required to ensure the distribution shift is not significant in order to maintain the same optimal regret will, otherwise, regret can easily blow up.
>
> Once again, thank you for your dedication to reviewing our submission and your valuable feedback that will contribute to the improvement of our work.

---

> > ### Comment · Reviewer_U3pE · 2023-08-19
> >
> > Thank you for all the clarifications! I'm still not entirely clear on how to interpret the experiment results of D-UCBVI in Figure 1.
> > If UCB-based methods are less robust and converge to sub-optimal policies quickly, I would expect the blue curve to grow slowly most of the time instead of having big jumps. (Unless the sudden increase in average return happens when D-UCBVI receives the delayed observations and updates the parameters accordingly)

---

> > > ### Author Response · Authors · 2023-08-20
> > >
> > > Thank you for your further questions. We are happy to provide more details.
> > >
> > > More specifically, the performance "jump" for D-UCBVI in Figure 1 results in the design of the environment (reward of the optimal action is much greater than that of sub-optimal actions) and the lack of robustness of UCB-based methods. In the setting of Figure 1, we have $|\mathcal{S}| = 2, |\mathcal{A}| = 50, |H| = 20, d = 10$. According to the design (details can be found in Appendix E.2 Line 946 - 955), at each step $h \in [H]$, only one (optimal) action results in the instant reward as $r = 0.99$ for each state $s$, all other 49 actions result in the same small reward as $r = 0.01$. Roughly speaking, when an algorithm chooses an optimal action at one time step, it results in an instant reward of (approximately) $1$. Therefore, the average return can be deemed as the average number of time steps that an algorithm chooses optimal actions in each episode.
> > >
> > > Recall that in D-UCBVI (Algorithm 4 in Appendix E.1), an action is chosen by taking the argmax of the estimated $Q_h$ value function, which is decided by the current estimation of parameter $w_h$ and the confidence intervals of each state-action pair: $Q_{h}^{k}(\cdot, \cdot) \leftarrow \phi(\cdot, \cdot)^T w_{h}^{k} + \beta \sqrt{\phi(\cdot, \cdot)^{T} (\Omega_h^k)^{-1} \phi(\cdot, \cdot)}$.
> > > At each step $h$, in order to choose the optimal action, its Q value needs to be the maximum among all $50$ actions. Similar to other UCB-based algorithms, D-UCBVI favors the actions that result in large rewards and are less explored. Since most of the actions (all sub-optimal ones) result in similar rewards, they all have a similar probability of being chosen due to the similar confidence interval. Therefore, even though the parameters are updated at the beginning of each episode, intuitively, it only eliminates the observed sub-optimal action by decreasing its uncertainty and updating its estimation, but there are still other unexplored sub-optimal actions that have a high probability of being chosen. As a result, the updates of parameters in D-UCBVI (take place at the beginning of each episode) do not necessarily result in the choice of optimal actions. It is only when the overall Q value of the optimal action dominates (compared to all other actions) will it be chosen.
> > >
> > > On the other hand, because the reward magnitude of the optimal action ($r = 0.99$) is significantly greater than that of all other sub-optimal actions ($r = 0.01$), "jumps" can be easily observed if the average number of times of selecting optimal actions differ, and if the algorithm is easily trapped in sub-optima. If the reward difference between the optimal action and sub-optimal ones is small, then the "jumps" will become smooth growing curves instead (e.g., $r = 0.6$ for optimal action, $r$ = 0.4 for all other sub-optimal actions). In comparison, with the same set of delays, the inherent randomness of our posterior sampling algorithms, together with the choice of parameters resulting from our theorems, provides better exploration to escape from local (sub) optima, resulting in fast convergence. Thus, the "jumps" are squeezed as the sharp climbing in the initial stage of D-PSVI and D-LPSVI.
> > >
> > > We hope this help provide a better understanding. Thank you for your time and willingness to engage!

---

> > > > ### Comment · Reviewer_U3pE · 2023-08-21
> > > >
> > > > Thanks for the authors' detailed responses. I raised the rating accordingly

---

> > > > > ### Author Response · Authors · 2023-08-22
> > > > >
> > > > > Thank you for your support! We are pleased to learn that our clarifications have helped address your main concerns. We are committed to incorporating the clarifications comprehensively in our revised manuscript.
> > > > >
> > > > > Thank you once again for your time and assessment.

---

### Official Review · Reviewer_p2ng · 2023-07-12

**Soundness:** 3 good
**Presentation:** 3 good
**Contribution:** 2 fair
**Rating:** 5
**Confidence:** 3

**Summary:**

This paper studies the problem of online reinforcement learning with delayed feedback. To address this problem, the authors proposed two Thompson sampling-based algorithms, Delayed Posterior Sampling Value Iteration (Delayed-PSVI) and Delayed-Langevin Posterior Sampling Value Iteration (Delayed-LPSVI). Delayed-PSVI adopts a Laplace approximation around the mode and thus it is computationally expensive due to the need of computing the covariance matrix. On the other hand, Delayed-LPSVI adopts the Langvein MCMC and thus is more tractable in computation when the dimension $d$ is large.

Theoretically, the authors proved the worst-case regret bound for the algorithms, which attains a regret bound of $O(\sqrt{d^3H^3T} + d^2H^2 E[\tau])$ for linear MDP, and $O(\sqrt{d^3 T})$ for linear bandits. A lower bound is also proved for linear bandits. Empirically, the authors conducted a comparison between Delayed-PSVI, Delayed-LPSVI, and a UCB variant D-UCBVI on a simulated MDP environment, demonstrating the advantages of Delayed-PSVI.

**Strengths:**

- The problem setup is useful for modeling many practical scenarios, where the conventional Thompson sampling framework cannot handle well.

- A lower bound is proved for linear bandits with delayed feedback, showing that the first term in the worst-case regret bound is tight.

- The authors also provided empirical evidence for demonstrating the effectiveness of their algorithm, though the setup is synthetic.

**Weaknesses:**

- The problem is not new, and it has been studied in prior bandit literature, for example, Pike-Burke et al. (2018). The technical contribution might be limited, given the results from bandits and Thompson sampling under the standard setup. It would be better if the authors could emphasize more on the key technical challenges and contrast them with the bandit literature.

- The assumption on the noise (sub-exponential delay) is not well-motivated and is overly strong. From a practical perspective, the distribution of delay should have a long tail, and it is possible that some users may not respond at all.

- Also, the delays should not be iid across different users or different groups of users in online recommendation. Thus the iid assumption is too strong.

-----

Thanks the authors for addressing my concerns, I will increase the score to 5.

**Questions:**

See weakness and strengths.

**Limitations:**

See weakness and strengths.

---

> ### Author Rebuttal · Authors · 2023-08-08
>
> We appreciate Reviewer p2ng's detailed evaluation of our work. We thank the reviewer for recognizing the significance of the delay feedback problem that we address and for providing valuable feedback that will contribute to the refinement of our work.
>
> Below we provide our response to the questions and critical points raised by the reviewer.
>
>     1. Comparison with existing literature
> Existing literature mainly studies delayed feedback in bandit settings, which is summarized in Section 2 (Preliminaries). In comparison, delayed feedback is less understood in the RL setting, and is more challenging in the sense that:
>
> (a) Existing bandit literature typically studies delays either in rewards or in the selection of arms independently. No prior work jointly studies the effect of delays that take place simultaneously in different aspects. However, in RL, delays can be observed not only in rewards, but also in state transitions, actions which in turn will also affect the transition. Therefore, in our work, we study a more challenging setting by considering delays with respect to the whole episodic trajectory that simultaneously includes transitions, actions, and rewards.
>
> (b) In RL, [Lancewicki et al. 2022] and [Jin et al. 2022] consider adversarial MDPs, concurrent works [Howson et al 2023] and [Mondal and Aggarwal 2023] study stochastic tabular MDPs, whereas our work focuses on linear MDPs for large-scale RL. Our regret bounds does not  depend on $|S|$ or $|A|$. In addition, instead of making the simpler assumption that delays are fixed and constant, we study the more practical setting of stochastic delays, which is unknown and random at each time step.
>
> (c) Existing works in bandits and the very recent works that study delays in RL only provide theoretical guarantees for UCB-based algorithms. How to perform posterior sampling under delayed feedback remains unknown. Therefore, we aim to bridge this gap by providing a theoretical understanding for two posterior sampling algorithms in linear MDP, and providing a corollary for linear bandits. To the best of our knowledge, our analysis is the first work that tackles all such challenges together.
>
> We will further clarify and make a more comprehensive comparison with existing literature in our revised manuscript.
>
>     2. Clarification of assumptions on delays
> We would like to highlight that the assumption of sub-exponential delay is mainly required for theoretical analysis:
> (a) For stochastic delays with general / arbitrary / non-sub-exponential distributions, we provide a discussion on the regret bound at the end of Section 4 (Line 277 - 282) and Appendix A.3. Briefly speaking, the regret can be (roughly) bounded by $\widetilde{O}(\frac{1}{q}\sqrt{d^3H^3T}+dH^2 d_\tau(q))$, where $d_\tau(q)$ is the $q$-th quantile of delay $\tau$. This could be achieved by creating a low-switching variant of our Thm 1 and Thm 2. Note that for delays that are not heavy-tailed (which include a wide class of practical phenomena, e.g. Poisson, multinomial, Gaussian, etc.), sharp analysis can be obtained by positing the sub-exponential assumption. Therefore, we focus on the latter case.
>
> (b) Such an assumption is common and standard in prior works with theoretical guarantees under delayed feedback (e.g. [Howson et al 2023, Yang et al 2023]). The sub-exponential / sub-Gaussian assumptions are required in order to obtain proper posterior concentration results to prove sub-linear regret bounds. It allows us to obtain tighter bounds by utilizing nice distribution properties.
>
> (c) Our algorithms exhibit promising empirical performance even when the assumption is not satisfied. As shown in experiments (Section 5.1, figure 1(c)), with long-tail delays, the performance of our algorithms does not deteriorate whereas the UCBVI algorithm is suboptimal.
>
> (d) The statistical property of heavy-tailed delays in bandits is studied very recently (Jul 2023) by [Shi etc. 2023]. Nevertheless, this paper focuses on establishing statistical inference approaches and does not analyze the regret guarantee in this scenario. Still, It is unclear whether the result in this paper help achieve a better regret bound other than the one we discussed above in (a).
>
> (e) The i.i.d. assumption on delays is common in the existing theoretical literature. While the non-i.i.d.case is interesting, it is beyond the scope of the current work, and we thus leave it for future study.
>
> Once again, thank you for dedicating time to reviewing our work and for providing valuable insights that will contribute to its improvement.
>
>     Reference:
>
> Yunchang Yang,  et al. "A reduction-based framework for sequential decision making with delayed feedback." arXiv preprint arXiv:2302.01477, 2023.
>
> Shi, Lei, et al. "Statistical Inference on Multi-armed Bandits with Delayed Feedback." ICML, 2023.
>
> Lancewicki, Tal, et al. "Learning adversarial markov decision processes with delayed feedback." Proceedings of the AAAI Conference on Artificial Intelligence. 2022.
>
> Jin, Tiancheng, et al. "Near-optimal regret for adversarial mdp with delayed bandit feedback." Advances in Neural Information Processing Systems 35, 2022.
>
> Howson, Benjamin, et al. "Optimism and delays in episodic reinforcement learning." International Conference on Artificial Intelligence and Statistics. PMLR, 2023.
>
> Mondal, W. U., & Aggarwal, V. Reinforcement Learning with Delayed, Composite, and Partially Anonymous Reward. arXiv preprint arXiv:2305.02527, 2023.

---

> > ### Author Response · Authors · 2023-08-17
> > **Rebuttal by authors**
> >
> > Concerning the key technical challenges and contributions in RL domain, we would like to highlight the following aspects,
> >
> > **(a) Weaker concentration due to delays**
> >
> > Standard TS in bandits learns posteriors for reward functions without involving state transitions, while PSRL typically adopts posterior sampling to learn transition models. In contrast, we aim to learn posterior distributions for value functions, which encode information from both transitions and rewards. Compared to the standard no-delay setting, delays result in weaker posterior concentration, which can cause distribution shift and hinders timely update in learning posteriors in value space. The key is to show that even with weaker posterior concentration, our posterior sampling algorithms are able to achieve desired concentration and anti-concentration properties for the parameters that govern both transitions and rewards, so as to achieve sub-linear worst-case regrets. We also generalize our theoretical result from linear MDPs to linear Bandits, which is lacking in existing bandit literature for posterior sampling algorithms.
> >
> > **(b) Incorporation of Langevin Monte Carlo under delays**
> >
> > To the best of our knowledge, how to incorporate approximate Bayesian inference under (stochastic) delays remains unknown in both bandit and RL literature (as well as in the settings with function approximation), though it is particularly useful in learning intractable posteriors in practice. We aim to bridge this gap by provably incorporating Langevin Monte Carlo (LMC) in algorithm 2, which is an important extension from the standard posterior sampling algorithms. By utilizing LMC, algorithm 2 is proved to enjoy computational efficiency while maintaining the same regret guarantee (Thm 2). Besides, owing to the ability of LMC, algorithm 2 (Delayed-LPSVI) is able to generate samples from arbitrarily complex posteriors in complex problem domains, where posterior distributions can be intractable without closed-form. When standard posterior sampling fails to capture the noise model accurately, Delayed-LPSVI is expected to come into play. In particular, we show that the convergence guarantee provided by Langevin Monte Carlo is strong enough to yield approximate posteriors with high accuracy even with fewer available samples due to delays (Lemma C.2).
> >
> >
> > **(c) Worst-case regret bounds with delay error**
> >
> > We provide frequentist analyses to prove worst-case regret bounds for our posterior sampling algorithms in RL with linear function approximation, which are more challenging and stronger guarantees compared to expected regret or Bayes regret. To do so, we rely on a multi-run sampling scheme that ensures optimism with high probability. Compared to the existing analysis of PS in no-delay RL, it requires the computation of new multi-round parameter, careful design of the scaling factor of posterior variance, and the resulting changes in Bayesian linear models to account for the weaker concentration arising from delays. In general, this is challenging, as
> >
> > (i) we need to balance between concentration and anti-concentration to make sure the resulting posterior sampling algorithm provides optimistic estimates that do not deviate too much from the true values even when delays take place;
> >
> > (2) it is important to guarantee that the number of runs for sampling (i.e. multi-round parameter) has proper scaling in time horizon. We show that in Lemma B.7, our choice of $M$ has order $Polylog(H, k, d, \delta)$ and thus does not increase the overall complexity dependence.
> >
> > In addition, the presence of delays prohibits the direct application of Elliptical Potential Lemma to upper bound the estimation error $\Delta_{est}^k$ in regret decomposition as in the non-delayed case. To address this issue, we decompose the covariance matrix of posteriors into two parts: a full design matrix and a complement matrix. Note that this design is dedicated to delayed settings and requires more careful analysis to bound the delayed error incurred. Details can be found in Appendix B.1, where Lemma B.3 and Lemma B.4 are the key lemmas in addressing it.
> >
> > ------------
> > We hope the further clarification helps address your concern about the technical challenges and contributions of our work. And we are committed to providing a more clear discussion in our revised version.
> >
> > Thank you once again in providing valuable feedback and taking time in reviewing our work!

---

### Official Review · Reviewer_RbB5 · 2023-07-26

**Soundness:** 3 good
**Presentation:** 3 good
**Contribution:** 2 fair
**Rating:** 6
**Confidence:** 2

**Summary:**

This paper presents a theoretical study of a posterior sampling algorithm for linear MDPs with _delayed rewards_. The authors propose two algorithms, Delayed-PSVI and Delayed-LPSVI, which are based on sampling from a Gaussian posterior distribution over weights induced by a Bellman style likelihood function. The LPSVI variant is similar to PSVI, except the posterior sampling is done using Langevin Monte Carlo instead of sampling from the potentially high-dimensional Gaussian directly (motivation is reduced computational complexity). The authors provide worst-case regret bounds for both algorithms, with optimal dependence on on dimension $d$ and time horizon $T$, and slightly suboptimal dependence on the episode length $H$. The theoretical study is complemented by a toy experiment in which the proposed algorithms perform better than a UCB based baseline.

**Strengths:**

* To my knowledge, this paper provides the first worst-case regret bounds for Thompson sampling in linear MDPs with delays.

* The bounds are tight up to a $\sqrt{H}$ factor (and possibly polylog terms).

**Weaknesses:**

* The experimental setup is somewhat limited, with only two states and 10 feature dimensions.

* Little detail on how LMC was tuned, and evidence it converged in practice.

**Questions:**

* Can you clarify your comment on page 5, where you claim that your analysis is relevant for double-Q learning? IIUC, the point of having two networks in double-Q learning is to alleviate overestimation of Q-values, whereas, in your analysis, taking multiple samples of the Q function is done to ensure that your estimates are higher than the true optimal Q value with high probability? (That sounds quite different.)

* Could you please clarify why does it make sense to condition the posterior model on past estimates of $\tilde{V}$? From a Bayesian perspective, the only observed quantities are rewards (and delays); it seems like conditioning on past estimates of $\tilde{V}$ could lead the algorithm astray, especially if the number of samples required for convergence is large. How does your approach/analysis handle this?

**Limitations:**

This is not my area of expertise. I am not aware of limitations beyond the already raised simplicity of experiments (which is ok for a theoretical paper like this).

---

> ### Author Rebuttal · Authors · 2023-08-08
>
> We appreciate Reviewer RbB5's thorough evaluation and positive assessment of our work. We thank the reviewer for providing valuable feedback that will contribute to the refinement of our work.
>
>     1. Clarification of relationship with double Q-learning
> Thank you for the good question! Overall speaking, Double q learning and our method are designed for different purposes. Double q learning is designed for preventing the overestimation issue in RL (the learned Q function largely overestimates the true optimal q values), while our multi-round sampling scheme is for achieving optimism. Concretely, we utilize the multi-round sampling scheme to ensure optimism ($\widetilde{Q} \geq Q^*$) is achieved with high probability so as to perform worse-case regret analysis for posterior sampling algorithms. The fact that we mention ensampling in existing empirical studies such as double Q-learning is to stress its importance and effectiveness in achieving good empirical performance in various settings. Owing to its empirical popularity, we contribute to explaining its theoretical effectiveness in our work, and justify how it can be adopted to achieve the desired performance guarantee in our scenario.
>
>
>     2. Clarification of posterior models
> The reason why the posterior model is valid based on the past estimates of $\widetilde(V)$ are two fold:
>
> (a) In RL, apart from rewards, an agent also observes state transitions. Therefore, delays could take place in rewards, state transitions, and the impact of actions (which in turn also affects state transitions). Recall that from Line 13 of Algorithm 1, $\widetilde{V}_{h}(\cdot, \cdot) \leftarrow \max_a\min (\widetilde{Q}_h^k(\cdot, a), H-h+1)$.
>
> As a result, $\widetilde{V}_{h}$ is estimated based on the observations of rewards and transitions.
>
> (b) According to the definition of linear MDP (Definition 1), both the transition dynamics and reward functions are linear in feature map. Thus, $Q$ function is also linear in feature map for some parameter $w^*$(see Lemma A.1), where parameter $w^*$ governs both transitions and rewards. Here, our posterior sampling algorithm is to learn the posterior distributions of parameter $w$, and note that $\widetilde{V}_{h}$ (or $\widetilde{Q}$) is essentially a linear transformation on the observed transitions and rewards, it therefore serves as the observed data for the posterior. In short, the parameter of interest ($w$) governs both transitions and rewards, to learn its posteriors, we need to condition on the observations that also encode the information from transitions and rewards.
>
>     3. Clarification of experiments
> In addition, we thank the reviewer's critique regarding the experiments in our paper. Here are a few points that we would like to highlight:
> (a). Our main contribution is to provide a theoretical understanding of posterior sampling algorithms in linear MDP (and linear bandits) under delayed feedback for the very first time. Toward this end, we design two posterior sampling algorithms. One is based on the Bayesian linear regression with Gaussian noise perturbation on value function space, and the other incorporates approximate sampling by resorting to Langevin Monte Carlo (LMC) method). We establish the same worst-case regret bound for both algorithms.
>
> (b). Hence, to corroborate our theoretical findings, we perform experiments in two different sets of environments. Note that in synthetic Linear MDP environments, we vary $|S||A|$ (cardinality of 20, 40, 100, 200) to demonstrate the empirical performance, details can be found in Appendix E.2. Due to the space constraint, we only present results for $|S| = 2, |A|=50$ in the main text. In the main text, even though the state space is small, we do consider a large action space for testing. Essentially, whether with a large number of actions or a large number of states will not affect the empirical results and parameters, as our bounds do not have dependence on $|S|$ and $|A|$.
>
> (c). The parameter setting of LMC closely follows our Theorem 2, which guides us to choose the proper scaling for all parameters, and the actual tuning is to choose the appropriate constant depending on the environments that we are testing upon. For synthetic MDP, parameter details of LMC can be found in Line 303-308; and the counterpart in the RiverSwim environment can be found in Line 989-983. In terms of the convergence of LMC, one can easily discover the convergent behavior from the plots in Figure 1. Upon the convergence of regret performance, we find that further increasing the number of iterations in LMC will not further help improve the overall performance.
>
> Once again, thank you for dedicating time to reviewing our work and for providing valuable insights that will contribute to its improvement.

---

### Official Review · Reviewer_UyNX · 2023-07-26

**Soundness:** 3 good
**Presentation:** 3 good
**Contribution:** 2 fair
**Rating:** 5
**Confidence:** 3

**Summary:**

This paper studies the Thompson sampling methods for reinforcement learning problems with delayed feedbacks. The authors tackle the challenge of delayed feedback under the linear function approximation setting. They proposed two new Thompson sampling algorithms, using the exact sampling mechanism and also the Langevin dynamics for an approximate posterior sampling. The regrets and the computation complexity for both algorithms are then analyzed.  They further conduct experiments to compare the empirical results of the proposed methods and the UCB-based algorithms.

**Strengths:**

1) The authors solve the reinforcement learning (RL) problem with delayed feedback from the angle of posterior sampling rather than the well-explored UCB-based algorithms, which is a new solution to this problem.

2) The second posterior sampling algorithm based on Langevin dynamics outperforms the first algorithm in terms of computation complexity, which improves the efficiency of the posterior sampling methods for the problem of RL with delayed feedback.

3) The authors conduct sufficient empirical comparisons to justify the superiority of the posterior sampling over UCB methods in RL with delayed feedback.

**Weaknesses:**


1) Although this is the first work for RL with delayed feedback by posterior sampling, the technical contribution of this work still seems not significant. In terms of the algorithm design, the methods seem to be combinations of existing posterior sampling methods for linear MDPs and the multi-round sampling scheme in [24]. The theoretical analysis, in my understanding, is also mainly dependent on the analysis for RL without delay and the decomposition of the covariance matrix to show the error caused by the delay. Can the authors further discuss in detail the major technical challenges in this work and the corresponding technical contributions?

2) In order to clearly show the effectiveness of the posterior sampling methods, the authors need to further present a detailed comparison of the regret results in the existing UCB-based approaches and the proposed posterior sampling approach for RL with delayed feedback.

3) Is there any possibility of further improving the regret term associated with the delay? For example, use some techniques to achieve a regret more robust to the delay $\tau$, i.e., having a sublinear dependence on $\tau$, or to improve the dependence on the multiplicative factors $H$ and $d$. I am wondering what the best-known result is for the UCB-based approaches under a similar setting.




**Questions:**

Please see the Weaknesses.

**Limitations:**

The authors have discussed the limitations.

---

> ### Author Rebuttal · Authors · 2023-08-09
>
> We appreciate Reviewer UyNX's detailed evaluation of our work. We thank the reviewer for the thoughtful feedback and for recognizing the soundness of our work.
>
>     1. Clarification of technical contribution
> Regarding the concern about the technical contribution, we would like to point out that:
> (a) While from the algorithmic perspective, Algorithm 1 behaves similarily as standard posterior sampling procedures and utilize multi-run sampling to ensure worse-case regret guarantee, the main challenges lie in the analysis under the delayed feedback setting. Compared to the standard setting with immediate feedback, the posterior concentration is weaker when delays take place. As the agent needs to take actions by utilizing the observed history and existing posteriors only, which can cause distribution shift. The key is to show that even with weaker posterior concentration, our algorithm is able to achieve the desired concentration and anti-concentration criteria that guarantee the sub-linear regret. Besides, we show that in algorithm 2, the convergence guarantee provided by LMC is strong enough to yield approximate posteriors with high accuracy even with fewer available samples due to delays.
>
> (b) In addition, we incorporate approximate sampling scheme by resorting to Langevin Monte Carlo (LMC) in algorithm 2, which is an important extension from the standard posterior sampling algorithms. By utilizing LMC, algorithm 2 is proved to enjoy computational efficiency while maintaining the same regret guarantee (Thm 2). Besides, owing to the ability of LMC, algorithm 2 (Delayed-LPSVI) is able to generate samples from arbitrarily complex posteriors in complex problem domains, where posterior distributions can be intractable without closed-form. When standard posterior sampling fails to capture the noise model accurately, algorithm 2 (Delayed-LPSVI) is expected to come into play. To the best of our knowledge, this is the first analysis that provably incorporates LMC in linear MDPs and jointly consider the effect of stochastic delays.
>
>     2. Comparison with UCB-based methods
>
> Theoretical understanding of delayed feedback is mainly studied in bandit settings, and is less underexplored in RL until recently. We summarize and compare our work with all existing theoretical works in RL with delayed feedback that we are aware of:
>
> (a) Different RL settings: while [Lancewicki et al. 2022] and [Jin et al. 2022] consider adversarial MDPs, concurrent works [Howson et al 2023] and [Mondal and Aggarwal 2023] study episodic tabular MDPs, our work focuses on linear MDPs.
>
> (b) In [Yang et al 2023], they propose a reduced framework for different sequential decision-making problems. However, their framework relies on policy-switching schemes, which differ from our settings with fixed episodic length.
>
> (c) All the above works study UCB-based methods.
>
> Therefore, there is no prior theoretical work that studies exactly the same setting as ours. In our experiments, we thus adapt the UCBVI algorithm to fit the delayed linear MDP setting and test the empirical performance against it. For a fair comparison, we grid search to choose the optimal parameters for Dealyed-UCBVI in experiments.
>
> We will further clarify and make a more comprehensive comparison with UCB approaches in our revised manuscript.
>
>
>     3. Comment on potential improvement of delayed dependence
> The dependence on $\mathbb{E}[\tau]$ in the regret term associated with the delay matches that in the concurrent works ([Howson et al 2023] and [Yang et al 2023]). Therefore, we are not optimistic that this dependence can be further improved. We conjecture that more careful analysis may help bring down the dependence on $H$.
>
> Once again, thank you for dedicating time to reviewing our work and for providing valuable insights to further enhance the quality of our work.
>
>     Reference:
>
> Yang, Yunchang, et al. "A Reduction-based Framework for Sequential Decision Making with Delayed Feedback." arXiv preprint arXiv:2302.01477 (2023).
>
> Lancewicki, Tal, et al. "Learning adversarial markov decision processes with delayed feedback." Proceedings of the AAAI Conference on Artificial Intelligence. 2022.
>
> Jin, Tiancheng, et al. "Near-optimal regret for adversarial mdp with delayed bandit feedback." Advances in Neural Information Processing Systems 35, 2022.
>
> Howson, Benjamin, et al. "Optimism and delays in episodic reinforcement learning." International Conference on Artificial Intelligence and Statistics. PMLR, 2023.
>
> Mondal, W. U., & Aggarwal, V. Reinforcement Learning with Delayed, Composite, and Partially Anonymous Reward. arXiv preprint arXiv:2305.02527, 2023.

---

> > ### Author Response · Authors · 2023-08-10
> >
> > In addition, we would also like to mention that in delayed feedback setting, it is common that algorithms are developed based on existing algorithms, sharing similar algorithmic procedures as in no-delay setting (e.g. [Vernade et al 2020, Howson et al 2023, Yang et al 2023]). However, it requires the computation of new confidence interval / exploration bonus / posterior variance due to the weaker concentration arising from delays. In general, this is challenging, since
> >
> > (a) we need to balance between concentration and anti-concentration to make sure the resulting posterior sampling algorithm achieves the trade-off between optimism and estimation accuracy, so as to provide optimistic estimates that do not deviate too much from the true values even when delays take place;
> >
> > (b) in our case, multi-round parameter, the scaling factor of the posterior variance, and the resulting changes in Bayesian linear models need to be adapted simultaneously to maintain the above trade-offs while accounting for the effect of delays;
> >
> > (c) it is important to guarantee that the number of runs for sampling (i.e. multi-round parameter) has proper scaling in time horizon, which indicates optimistic samples can be obtained with a few reasonable number of runs under delays. We show that in Lemma B.7, our choice of M has order $Polylog(H, K, d, \delta)$ and thus does not increase the complexity dependence;
> >
> > (d) the presence of delays prohibits the direct application of Elliptical Potential Lemma to bound the term $\sum_{k=1}^K||\phi(s^k, a^k)||_{(\Omega^k)^{-1}}$ in regret decomposition as in the non-delayed case. To address this issue, we decompose the covariance matrix of posteriors into two parts: a full design matrix and a complement matrix. Note that this design is dedicated to delayed settings and requires more careful analysis to bound the delayed error incurred. Details can be found in Appendix B.1, where Lemma B.3 and Lemma B.4 are the key lemmas.
> >
> > We will further clarify the technical challenges in delayed feedback setting and our technical contributions. Thanks for bringing it up!

---

> > > ### Comment · Reviewer_UyNX · 2023-08-18
> > >
> > > Thank you very much for your response. I will modify the rating accordingly.

---

> > > > ### Author Response · Authors · 2023-08-18
> > > >
> > > > Thank you for your review and support.  We are pleased to learn that our clarifications have helped address your main concerns. We are committed to incorporating the clarifications comprehensively in our revised manuscript.
> > > >
> > > > Thank you once again for your time and assessment.

---

### Official Review · Reviewer_8fzn · 2023-08-01

**Soundness:** 3 good
**Presentation:** 3 good
**Contribution:** 3 good
**Rating:** 6
**Confidence:** 3

**Summary:**

This paper considers the scenario where the feedback is delayed. The motivation for such a scenario is recommender/advertisement systems and clinical trials. They present a novel algorithm based on Thompson sampling idea: Delayed-PSVI. The authors present theoretical results for worst-case regret. They also show experimental results on synthetic linear MDP.

**Strengths:**

Most of the works in RL are focused on instant feedback. This paper consider a real-time problem where the feedback may not instant. The theoretical results along with the experimental section make solid contribution.

**Weaknesses:**

-

**Questions:**

1. Posterior sample algorithms often are restricted to Gaussian, multinomial distribution due to posterior computation. The proposed algorithm considers normal distribution, which I have commonly seen in posterior sampling algorithms. Is such an assumption restrictive when it comes to real-world application?

**Limitations:**

-

---

> ### Author Rebuttal · Authors · 2023-08-08
>
> We appreciate Reviewer 8fzn's thoughtful review and positive assessment of our work. We are pleased to see that the reviewer acknowledges the significance of addressing delayed feedback scenarios in RL, and finds our proposed algorithm, Delayed-PSVI, to be a valuable contribution to this domain.
>
> The reviewer raises a pertinent question regarding the assumption of restrictive distributions in posterior sampling, considering its potential impact on real-world applications.
>
> We thank the reviewer for highlighting the concern about the real-world applicability of posterior sampling algorithms. We would like to stress that
> 1. When posterior sampling is adopted to learn reward distributions in bandits or to learn transition dynamics (as PSRL) in model-based RL, it typically relies on the conjugacy assumptions in order to make posteriors tractable. However, in our case, posterior sampling is applied to perform Bayesian linear regression for Q-functions. Intuitively, such model implements noise perturbation in the space of value functions. A Gaussian noise model is able to cope with a wide range of practical scenarios (e.g. sub-Gaussian noise, Poisson noise, multinomial noise etc.) even when practical noises do not follow a Gaussian distribution. Therefore, as mentioned in Line 153 - 156, while we adopt Gaussian prior and Gaussian likelihood, we do not assume that the Bayesian model is correct, and our analysis goes through naturally.
>
> 2. More importantly, our algorithm 2 (Delayed-LPSVI) incorporates Langevin Monte Carlo (LMC) as an approximate sampling scheme, which is capable of handling the general cases where posteriors are intractable and are without closed-form. We prove that when using LMC, Delayed-LPSVI is able to achieve the same regret guarantee as Delayed-PSVI. To focus on the delayed feedback, we restrict our discussion to the Gaussian case in both algorithms for consistency. However, algorithm 2 with LMC is powerful to work with non-conjugate posteriors and thus can be used to model complex problem domains in reality when Bayesian linear regression models with Gaussian noise fail to provide desirable performance.
>
> Once again, thank you for dedicating time to reviewing our work and for providing valuable insights that will contribute to its improvement.

---

### Author Rebuttal · Authors · 2023-08-10

We sincerely thank all reviewers for insightful comments on our work. We are grateful that reviewers recognize the significance of addressing delayed feedback in RL; find our proposed algorithms to be a valuable contribution to this domain; and acknowledge that the theoretical analysis is thorough and sound; the setup is practical and useful; empirical comparisons are sufficient.

Our main contribution is to provide a theoretical understanding of posterior sampling algorithms in linear MDP (and linear bandits) under delayed feedback. We provide the first worst-case regret bound for posteror sampling algorithms in this scenario. In addition, we provably incorporate Langevin Monte Carlo (LMC) as an approximate sampling scheme to further improve the computation efficiency and to provide the feasibility in generalizing to more complex problem domains where posterior can be intractable.

We hope the individual responses addressed your questions and clarified our settings, and you may consider increasing the rating to support our work. We are happy to address further questions and feedback.

---

### Decision · Program_Chairs · 2023-09-21

**Decision:**

Accept (poster)

**Comment:**

The concerns around technical novelty and others has been well addressed in the rebuttal. Thus I recommend acceptance.